# Global organization of neuronal activity only requires unstructured local connectivity

**David Dahmen[1]\*[†], Moritz Layer[1,2][†], Lukas Deutz[1,3], Paulina Anna Dąbrowska[1,2], Nicole Voges[1,4], Michael von Papen[1], Thomas Brochier[4], Alexa Riehle[1,4], Markus Diesmann[1,5,6], Sonja Grün[1,7], Moritz Helias[1,5]\***

[1]Institute of Neuroscience and Medicine and Institute for Advanced Simulation and JARA Institut Brain Structure-Function Relationships, Jülich Research Centre, Jülich, Germany; [2]RWTH Aachen University, Aachen, Germany; [3]School of Computing, University of Leeds, Leeds, United Kingdom; [4]Institut de Neurosciences de la Timone, CNRS - Aix-Marseille University, Marseille, France; [5]Department of Physics, Faculty 1, RWTH Aachen University, Aachen, Germany; [6]Department of Psychiatry, Psychotherapy and Psychosomatics, School of Medicine, RWTH Aachen University, Aachen, Germany; [7]Theoretical Systems Neurobiology, RWTH Aachen University, Aachen, Germany

**\*For correspondence:**
d.dahmen@fz-juelich.de (DD);
m.helias@fz-juelich.de (MH)

[†]These authors contributed equally to this work

**Competing interest:** The authors declare that no competing interests exist.

**Abstract** Modern electrophysiological recordings simultaneously capture single-unit spiking activities of hundreds of neurons spread across large cortical distances. Yet, this parallel activity is often confined to relatively low-dimensional manifolds. This implies strong coordination also among neurons that are most likely not even connected. Here, we combine in vivo recordings with network models and theory to characterize the nature of mesoscopic coordination patterns in macaque motor cortex and to expose their origin: We find that heterogeneity in local connectivity supports network states with complex long-range cooperation between neurons that arises from multi-synaptic, short-range connections. Our theory explains the experimentally observed spatial organization of covariances in resting state recordings as well as the behaviorally related modulation of covariance patterns during a reach-to-grasp task. The ubiquity of heterogeneity in local cortical circuits suggests that the brain uses the described mechanism to flexibly adapt neuronal coordination to momentary demands.

## Editor's evaluation

This is a thorough study showing that long-range correlations in the brain can arise without common input drive or long-range anatomical connections. These long-range correlations are modulated by the animal's behavioral state, a surprising finding that suggests a computational role for control of this kind of correlation. The paper details some analytical methods for modeling this behavior in disordered systems. The work will be of broad interest to neuroscientists, computational biologists, and biophysicists.

## Introduction

Complex brain functions require coordination between large numbers of neurons. Unraveling mechanisms of neuronal coordination is therefore a core ingredient towards answering the long-standing question of how neuronal activity represents information. Population coding is one classical paradigm

(*Georgopoulos et al., 1983*) in which entire populations of similarly tuned neurons behave coherently, thus leading to positive correlations among their members. The emergence and dynamical control of such population-averaged correlations has been studied intensely (*Ginzburg and Sompolinsky, 1994*; *Renart et al., 2010*; *Helias et al., 2014*; *Rosenbaum and Doiron, 2014*). More recently, evidence accumulated that neuronal activity often evolves within more complex low-dimensional manifolds, which imply more involved ways of neuronal activity coordination (*Gallego et al., 2017*; *Gallego, 2018*; *Gallego et al., 2020*): A small number of population-wide activity patterns, the neural modes, are thought to explain most variability of neuronal activity. In this case, individual neurons do not necessarily follow a stereotypical activity pattern that is identical across all neurons contributing to a representation. Instead, the coordination among the members is determined by more complex relations. Simulations of recurrent network models indeed indicate that networks trained to perform a realistic task exhibit activity organized in low-dimensional manifolds (*Sussillo et al., 2015*). The dimensionality of such manifolds is determined by the structure of correlations (*Abbott et al., 2011*; *Mazzucato et al., 2016*) and tightly linked to the complexity of the task the network has to perform (*Gao, 2017*) as well as to the dimensionality of the stimulus (*Stringer et al., 2019*). Recent work has started to decipher how neural modes and the dimensionality of activity are shaped by features of network connectivity, such as heterogeneity of connections (*Smith et al., 2018*; *Dahmen et al., 2019*), block structure (*Aljadeff et al., 2015*; *Aljadeff et al., 2016*), and low-rank perturbations (*Mastrogiuseppe and Ostojic, 2018*) of connectivity matrices, as well as connectivity motifs (*Recanatesi et al., 2019*; *Dahmen et al., 2021*; *Hu and Sompolinsky, 2020*). Yet, these works neglected the spatial organization of network connectivity (*Schnepel et al., 2015*) that becomes more and more important with current experimental techniques that allow the simultaneous recording of ever more neurons. How distant neurons that are likely not connected can still be strongly coordinated to participate in the same neural mode is a widely open question.

To answer this question, we combine analyses of parallel spiking data from macaque motor cortex with the analytical investigation of a spatially organized neuronal network model. We here quantify coordination by Pearson correlation coefficients and pairwise covariances, which measure how temporal departures of the neurons' activities away from their mean firing rate are correlated. We show that, even with only unstructured and short-range connections, strong covariances across distances of several millimeters emerge naturally in balanced networks if their dynamical state is close to an instability within a 'critical regime'. This critical regime arises from strong heterogeneity in local network connections that is abundant in brain networks. Intuitively, it arises because activity propagates over a large number of different indirect paths. Heterogeneity, here in the form of sparse random connectivity, is thus essential to provide a rich set of such paths. While mean covariances are readily accessible by mean-field techniques and have been shown to be small in balanced networks (*Renart et al., 2010*; *Tetzlaff et al., 2012*), explaining covariances on the level of individual pairs requires methods from statistical physics of disordered systems. With such a theory, here derived for spatially organized excitatory-inhibitory networks, we show that large individual covariances arise at all distances if the network is close to the critical point. These predictions are confirmed by recordings of macaque motor cortex activity. The long-range coordination found in this study is not merely determined by the anatomical connectivity, but depends substantially on the network state, which is characterized by the individual neurons' mean firing rates. This allows the network to adjust the neuronal coordination pattern in a dynamic fashion, which we demonstrate through simulations and by comparing two behavioral epochs of a reach-to-grasp experiment.

## Results

### Macaque motor cortex shows long-range coordination patterns

We first analyze data from motor cortex of macaques during rest, recorded with $4 \times 4 \, \text{mm}^2$, 100-electrode Utah arrays with 400 μm inter-electrode distance (*Figure 1A*). The resting condition of motor cortex in monkeys is ideal to assess intrinsic coordination between neurons during ongoing activity. In particular, our analyses focus on true resting state data, devoid of movement-related transients in neuronal firing (see Materials and methods). Parallel single-unit spiking activity of $\approx 130$ neurons per recording session, sorted into putative excitatory and inhibitory cells, shows strong spike-count correlations across the entire Utah array, well beyond the typical scale of the underlying

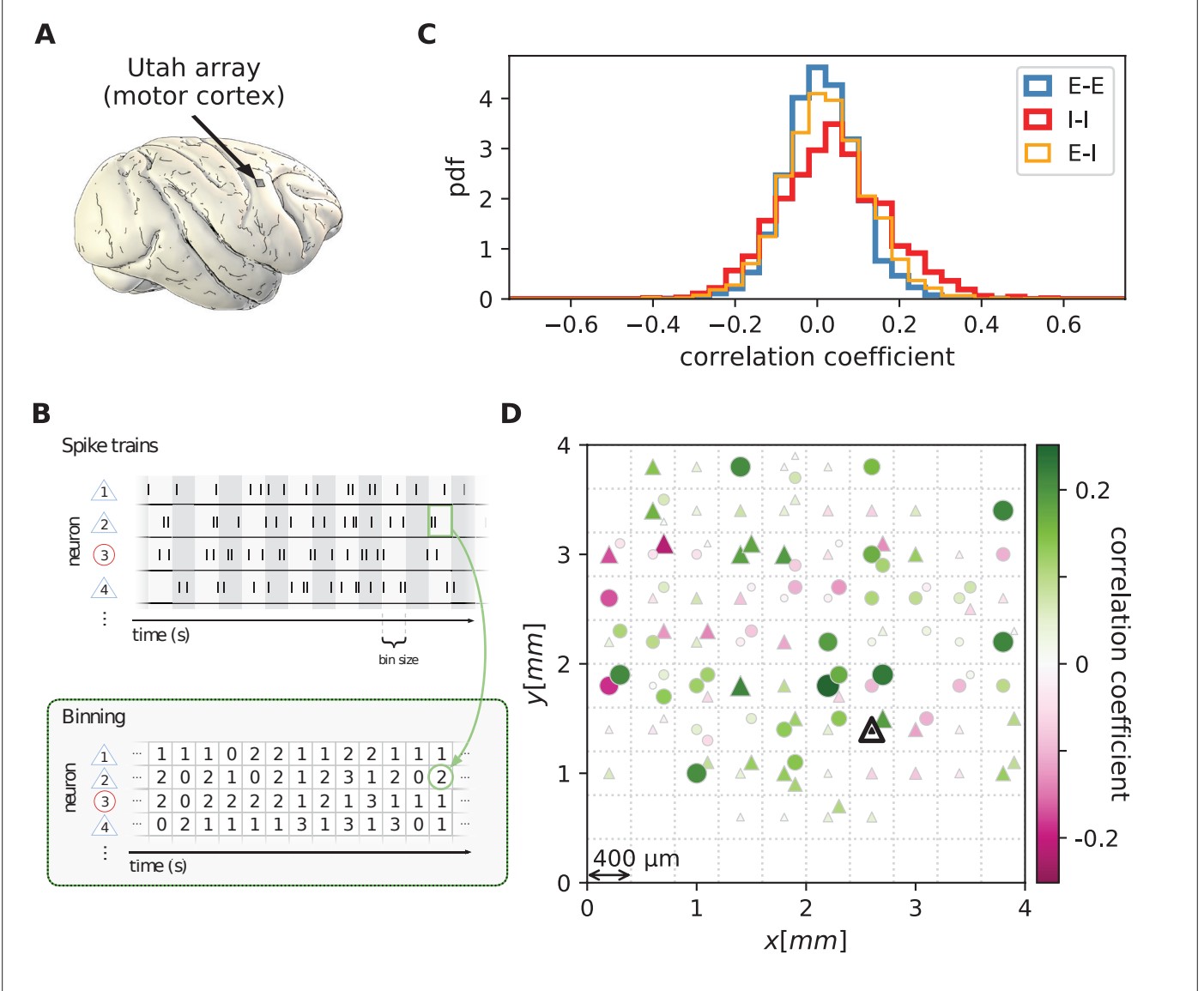

**Figure 1.** Salt-and-pepper structure of covariances in motor cortex. (**A**) Sketch of 10 × 10 Utah electrode array recording in motor cortex of macaque monkey during rest. (**B**) Spikes are sorted into putative excitatory (blue triangles) and inhibitory (red circles) single units according to widths of spike waveforms (see Appendix 1 Section 2). Resulting spike trains are binned in 1 s bins to obtain spike counts. (**C**) Population-resolved distribution of pairwise spike-count Pearson correlation coefficients in session E2 (E-E: excitatory-excitatory, E-I: excitatory-inhibitory, I-I: inhibitory-inhibitory). (**D**) Pairwise spike-count correlation coefficients with respect to the neuron marked by black triangle in one recording (session E2, see Materials and methods). Grid indicates electrodes of a Utah array, triangles and circles correspond to putative excitatory and inhibitory neurons, respectively. Size as well as color of markers represent correlation. Neurons within the same square were recorded on the same electrode.

The online version of this article includes the following source data for figure 1:

**Source data 1.** Code and data.

short-range connectivity profiles (**Figure 1B and D**). Positive and negative correlations form patterns in space that are furthermore seemingly unrelated to the neuron types. All populations show a large dispersion of both positive and negative correlation values (**Figure 1C**).

The classical view on pairwise correlations in balanced networks (**Ginzburg and Sompolinsky, 1994**; **Renart et al., 2010**; **Pernice et al., 2011**; **Pernice et al., 2012**; **Tetzlaff et al., 2012**; **Helias et al., 2014**) focuses on averages across many pairs of cells: average correlations are small if the network dynamics is stabilized by an excess of inhibitory feedback; dynamics known as the 'balanced state' arise (**van Vreeswijk and Sompolinsky, 1996**; **Amit and Brunel, 1997**; **van Vreeswijk and**

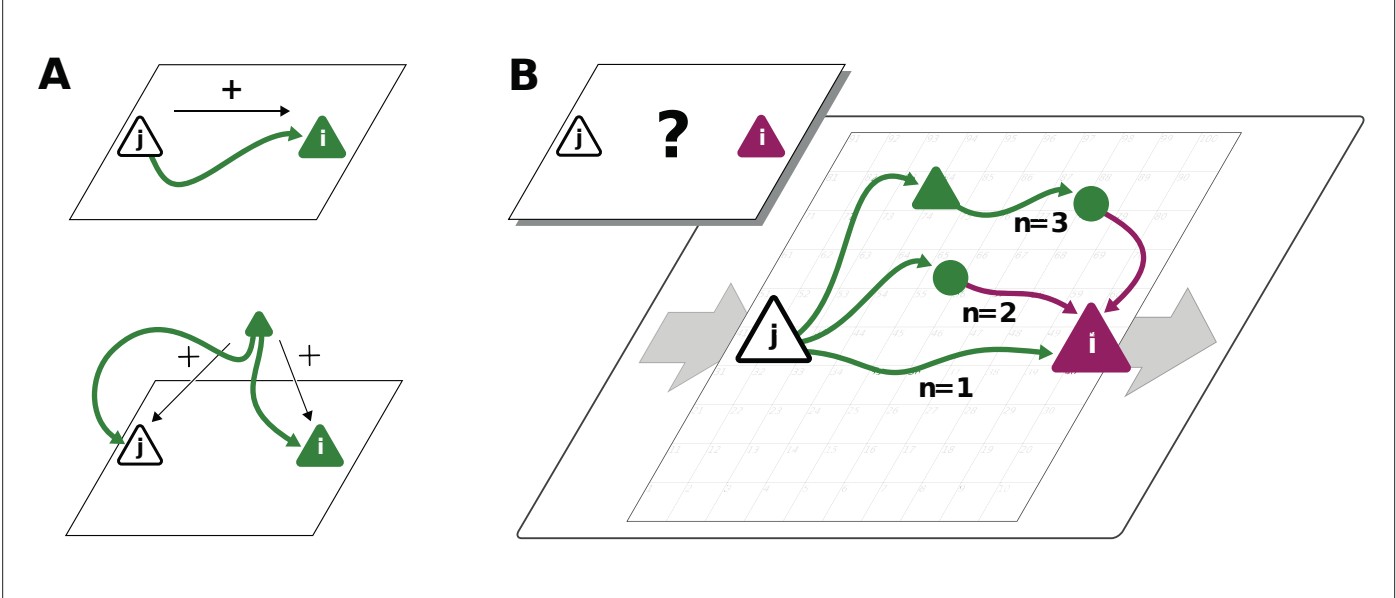

**Figure 2.** Correlations from direct and indirect connections. (**A**) Positive correlation (green neuron *i*) follows from direct excitatory connection (top) or shared input (middle). (**B**) Negative correlation (magenta) between two excitatory neurons cannot be explained by direct connections: Neuronal interactions are not only mediated via direct connections ($n = 1$; sign uniquely determined by presynaptic neuron type) but also via indirect paths of different length $n > 1$. The latter may have any sign (positive: green; negative: purple) due to intermediate neurons of arbitrary type (triangle: excitatory, circle: inhibitory).

*Sompolinsky, 1998*): Negative feedback counteracts any coherent increase or decrease of the population-averaged activity, preventing the neurons from fluctuating in unison (*Tetzlaff et al., 2012*). Breaking this balance in different ways leads to large correlations (*Rosenbaum and Doiron, 2014*; *Darshan et al., 2018*; *Baker et al., 2019*). Can the observation of significant correlations between individual cells across large distances be reconciled with the balanced state? In the following, we provide a mechanistic explanation.

## Multi-synaptic connections determine correlations

Connections mediate interactions between neurons. Many studies therefore directly relate connectivity and correlations (*Pernice et al., 2011*; *Pernice et al., 2012*; *Trousdale et al., 2012*; *Brinkman et al., 2018*; *Kobayashi et al., 2019*). From direct connectivity, one would expect positive correlations between excitatory neurons and negative correlations between inhibitory neurons and a mix of negative and positive correlations only for excitatory-inhibitory pairs. Likewise, a shared input from inside or outside the network only imposes positive correlations between any two neurons (*Figure 2A*). The observations that excitatory neurons may have negative correlations (*Figure 1D*), as well as the broad distribution of correlations covering both positive and negative values (*Figure 1C*), are not compatible with this view. In fact, the sign of correlations appears to be independent of the neuron types. So how do negative correlations between excitatory neurons arise?

The view that equates connectivity with correlation implicitly assumes that the effect of a single synapse on the receiving neuron is weak. This view, however, regards each synapse in isolation. Could there be states in the network where, collectively, many weak synapses cooperate, as perhaps required to form low-dimensional neuronal manifolds? In such a state, interactions may not only be mediated via direct connections but also via indirect paths through the network (*Figure 2B*). Such effective multi-synaptic connections may explain our observation that far apart neurons that are basically unconnected display considerable correlation of arbitrary sign.

Let us here illustrate the ideas first and corroborate them in subsequent sections. Direct connections yield correlations of a predefined sign, leading to correlation distributions with multiple peaks, for example a positive peak for excitatory neurons that are connected and a peak at zero for neurons that are not connected. Multi-synaptic paths, however, involve both excitatory and inhibitory intermediate neurons, which contribute to the interaction with different signs (*Figure 2B*). Hence, a single

indirect path can contribute to the total interaction with arbitrary sign (*Pernice et al., 2011*). If indirect paths dominate the interaction between two neurons, the sign of the resulting correlation becomes independent of their type. Given that the connecting paths in the network are different for any two neurons, the resulting correlations can fall in a wide range of both positive and negative values, giving rise to the broad distributions for all combinations of neuron types in *Figure 1C*. This provides a hypothesis why there may be no qualitative difference between the distribution of correlations for excitatory and inhibitory neurons. In fact, their widths are similar and their mean is close to zero (see Materials and methods for exact values); the latter being the hallmark of the negative feedback that characterizes the balanced state. The subsequent model-based analysis will substantiate this idea and show that it also holds for networks with spatially organized heterogeneous connectivity.

To play this hypothesis further, an important consequence of the dominance of multi-synaptic connections could be that correlations are not restricted to the spatial range of direct connectivity. Through interactions via indirect paths the reach of a single neuron could effectively be increased. But the details of the spatial profile of the correlations in principle could be highly complex as it depends on the interplay of two antagonistic effects: On the one hand, signal propagation becomes weaker with distance, as the signal has to pass several synaptic connections. Along these paths mean firing rates of neurons are typically diverse, and so are their signal transmission properties (*de la Rocha et al., 2007*). On the other hand, the number of contributing indirect paths between any pair of

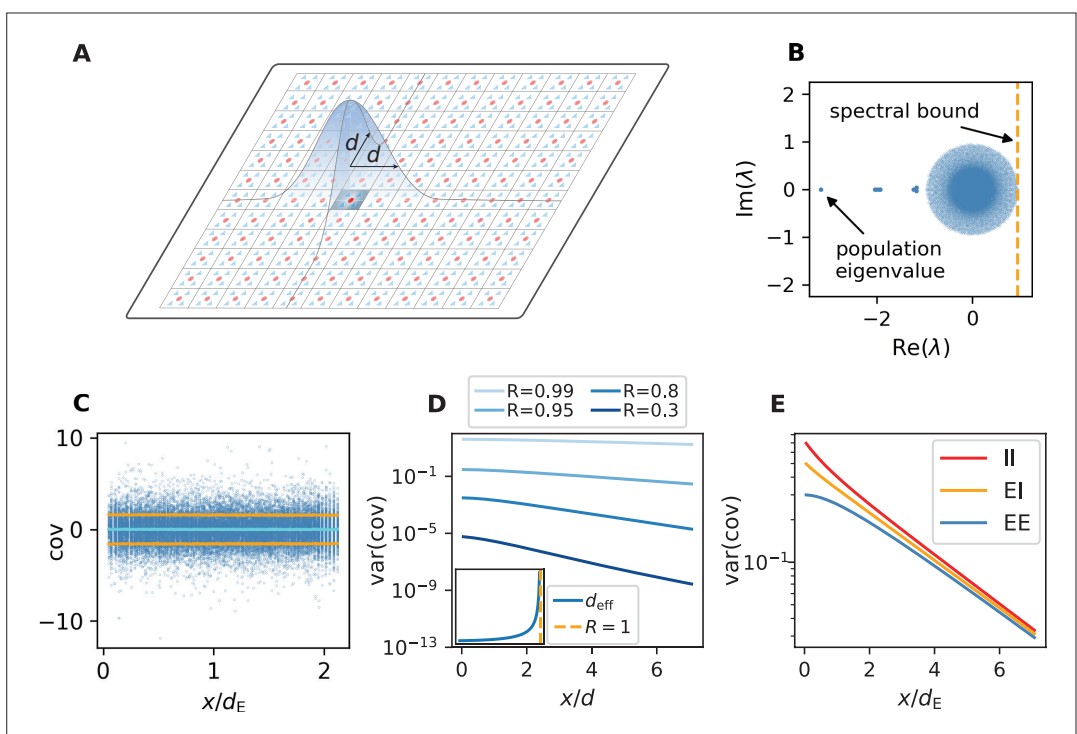

**Figure 3.** Spatially organized E-I network model. (**A**) Network model: space is divided into cells with four excitatory (triangles) and one inhibitory (circle) neuron each. Distance-dependent connection probabilities (shaded areas) are defined with respect to cell locations. (**B**) Eigenvalues $\lambda$ of effective connectivity matrix for network in dynamically balanced critical state. Each dot shows the real part $\mathrm{Re}(\lambda)$ and imaginary part $\mathrm{Im}(\lambda)$ of one complex eigenvalue. The spectral bound (dashed vertical line) denotes the right-most edge of the eigenvalue spectrum. (**C**) Simulation: covariances of excitatory neurons over distance $x$ between cells (blue dots: individual pairs; cyan: mean; orange: standard deviation; sample of 150 covariances at each of 200 chosen distances). (**D**) Theory: variance of covariance distribution as a function of distance $x$ for different spectral bounds of the effective connectivity matrix. *Inset*: effective decay constant of variances diverges as the spectral bound approaches one. (**E**) For large spectral bounds, the variances of EE, EI, and II covariances decay on a similar length scale. Spectral bound $R = 0.95$. Other parameters see *Appendix 1—table 3*.

The online version of this article includes the following source data for figure 3:

**Source data 1.** Code and data.

neurons proliferates with their distance. With single neurons typically projecting to thousands of other neurons in cortex, this leads to involved combinatorics; intuition here ceases to provide a sensible hypothesis on what is the effective spatial profile and range of coordination between neurons. Also it is unclear which parameters these coordination patterns depend on. The model-driven and analytical approach of the next section will provide such a hypothesis.

## Networks close to instability show shallow exponential decay of covariances

We first note that the large magnitude and dispersion of individual correlations in the data and their spatial structure primarily stem from features in the underlying covariances between neuron pairs (*Appendix 1—figure 1*). Given the close relationship between correlations and covariances (*Appendix 1—figure 1D and E*), in the following we analyze covariances, as these are less dependent on single neuron properties and thus analytically simpler to treat. To gain an understanding of the spatial features of intrinsically generated covariances in balanced critical networks, we investigate a network of excitatory and inhibitory neurons on a two-dimensional sheet, where each neuron receives external Gaussian white noise input (*Figure 3A*). We investigate the covariance statistics in this model by help of linear-response theory, which has been shown to approximate spiking neuron models well (*Pernice et al., 2012*; *Trousdale et al., 2012*; *Tetzlaff et al., 2012*; *Helias et al., 2013*; *Grytskyy et al., 2013*; *Dahmen et al., 2019*). To allow for multapses, the connections between two neurons are drawn from a binomial distribution, and the connection probability decays with inter-neuronal distance on a characteristic length scale $d$ (for more details see Materials and methods). Previous studies have used linear-response theory in combination with methods from statistical physics and field theory to gain analytic insights into both mean covariances (*Ginzburg and Sompolinsky, 1994*; *Lindner et al., 2005*; *Pernice et al., 2011*; *Tetzlaff et al., 2012*) and the width of the distribution of covariances (*Dahmen et al., 2019*). Field-theoretic approaches, however, were so far restricted to purely random networks devoid of any network structure and thus not suitable to study spatial features of covariances. To analytically quantify the relation between the spatial ranges of covariances and connections, we therefore here develop a theory for spatially organized random networks with multiple populations. The randomness in our model is based on the sparseness of connections, which is one of the main sources of heterogeneity in cortical networks in that it contributes strongly to the variance of connections (see Appendix 1 Section 15).

A distance-resolved histogram of the covariances in the spatially organized E-I network shows that the mean covariance is close to zero but the width or variance of the covariance distribution stays large, even for large distances (*Figure 3C*). Analytically, we derive that, despite the complexity of the various indirect interactions, both the mean and the variance of covariances follow simple exponential laws in the long-distance limit (see Appendix 1 Section 4 - Section 12). These laws are universal in that they do not depend on details of the spatial profile of connections. Our theory shows that the associated length scales are strikingly different for means and variances of covariances. They each depend on the reach of direct connections and on specific eigenvalues of the effective connectivity matrix. These eigenvalues summarize various aspects of network connectivity and signal transmission into a single number: Each eigenvalue belongs to a 'mode', a combination of neurons that act collaboratively, rather than independently, coordinating neuronal activity within a one-dimensional subspace. To start with, there are as many such subspaces as there are neurons. But if the spectral bound in *Figure 3B* is close to one, only a relatively small fraction of them, namely those close to the spectral bound, dominate the dynamics; the dynamics is then effectively low-dimensional. Additionally, the eigenvalue quantifies how fast a mode decays when transmitted through a network. The eigenvalues of the dominating modes are close to one, which implies a long lifetime. The corresponding fluctuations thus still contribute significantly to the overall signal, even if they passed by many synaptic connections. Therefore, indirect multi-synaptic connections contribute significantly to covariances if the spectral bound is close to one, and in that case we expect to see long-range covariances.

To quantify this idea, for the mean covariance $\bar{c}$ we find that the dominant behavior is an exponential decay $\bar{c} \sim \exp(-x/\bar{d})$ on a length scale $\bar{d}$. This length scale is determined by a particular eigenvalue, the population eigenvalue, corresponding to the mode in which all neurons are excited simultaneously. Its position solely depends on the ratio between excitation and inhibition in the network and becomes more negative in more strongly inhibition-dominated networks (*Figure 3B*). We show in

Appendix 1 Section 9.4 that this leads to a steep decay of mean covariances with distance. The variance of covariances, however, predominantly decays exponentially on a length scale $d_{\text{eff}}$ that is determined by the spectral bound $R$, the largest real part among all eigenvalues (*Figure 3B and D*). In inhibition-dominated networks, $R$ is determined by the heterogeneity of connections. For $R \lesssim 1$ we obtain the effective length scale

$$\frac{d_{\text{eff}}}{d} \sim \sqrt{\frac{R^2}{1-R^2} + \text{const.}} \gg 1. \tag{1}$$

What this means is that precisely at the point where $R$ is close to one, when neural activity occupies a low-dimensional manifold, the length scale $d_{\text{eff}}$ on which covariances decay exceeds the reach of direct connections by a large factor (*Figure 3D*). As the network approaches instability, which corresponds to the spectral bound $R$ going to one, the effective decay constant diverges (*Figure 3D* inset) and so does the range of covariances.

Our population-resolved theoretical analysis, furthermore, shows that the larger the spectral bound the more similar the decay constants between different populations, with only marginal differences for $R \lesssim 1$ (*Figure 3E*). This holds strictly if connection weights only depend on the type of the presynaptic neuron but not on the type of the postsynaptic neuron. Moreover, we find a relation between the squared effective decay constants and the squared anatomical decay constants of the form

$$d_{\text{eff,E}}^2 - d_{\text{eff,I}}^2 = \text{const.} \cdot \left( d_{\text{E}}^2 - d_{\text{I}}^2 \right). \tag{2}$$

This relation is independent of the eigenvalues of the effective connectivity matrix, as the constant of order $\mathcal{O}(1)$ does only depend on the choice of the connectivity profile. For $R \simeq 1$, this means that even though the absolute value of both effective length scales on the left hand side is large, their relative difference is small because it equals the small difference of anatomical length scales on the right hand side.

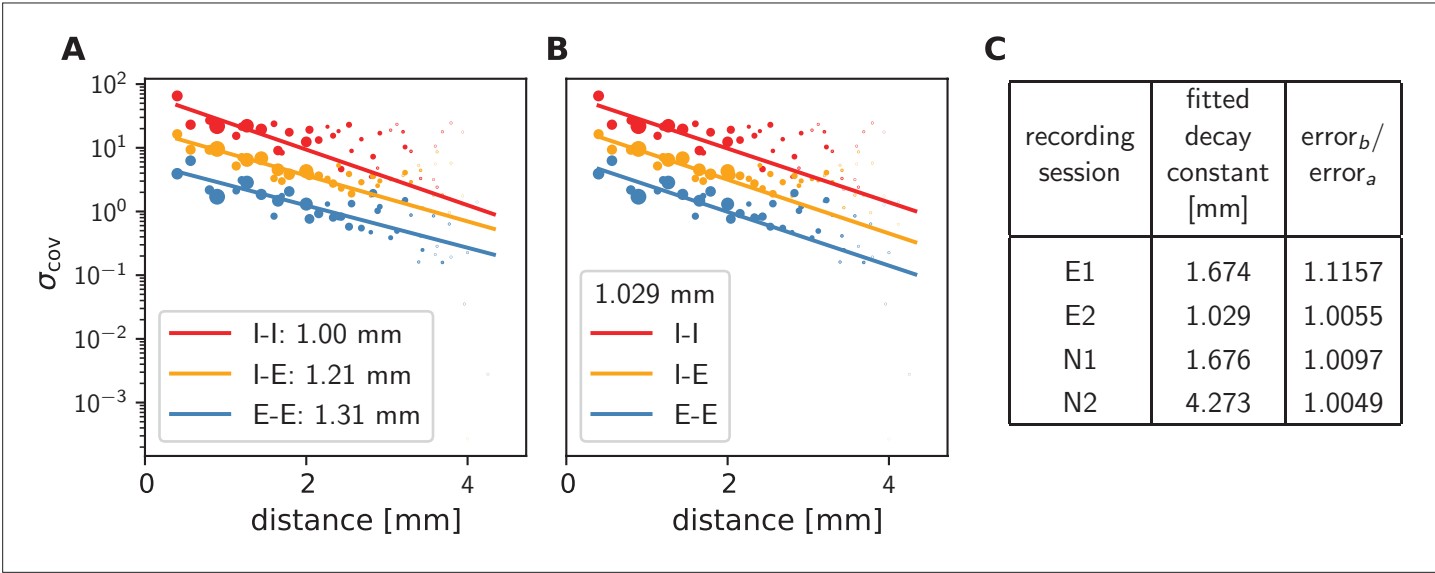

**Figure 4.** Long-range covariances in macaque motor cortex. Variance of covariances as a function of distance. (**A**) Population-specific exponential fits (lines) to variances of covariances (dots) in session E2, with fitted decay constants indicated in the legend (I-I: putative inhibitory neuron pairs, I-E: inhibitory-excitatory, E-E: excitatory pairs). Dots show the empirical estimate of the variance of the covariance distribution for each distance. Size of the dots represents relative count of pairs per distance and was used as weighting factor for the fits to compensate for uncertainty at large distances, where variance estimates are based on fewer samples. Mean squared error 2.918. (**B**) Population-specific exponential fits (lines) analogous to (A), with slopes constrained to be identical. This procedure yields a single fitted decay constant of 1.029 mm. Mean squared error 2.934. (**C**) Table listing decay constants fitted as in (B) for all recording sessions and the ratios between mean squared errors of the fits obtained in procedures *B* and *A*.

The online version of this article includes the following source data for figure 4:

**Source data 1.** Code and data.

## Pairwise covariances in motor cortex decay on a millimeter scale

To check if these predictions are confirmed by the data from macaque motor cortex, we first observe that, indeed, covariances in the resting state show a large dispersion over almost all distances on the Utah array (*Figure 4*). Moreover, the variance of covariances agrees well with the predicted exponential law: Performing an exponential fit reveals length constants above 1 mm. These large length constants have to be compared to the spatial reach of direct connections, which is about an order of magnitude shorter, in the range of 100-400 µm (*Schnepel et al., 2015*), so below the 400 µm inter-electrode distance of the Utah array. The shallow decay of the variance of covariances is, next to the broad distribution of covariances, a second indication that the network is in the dynamically balanced critical regime, in line with the prediction by *Equation (1)*.

The population-resolved fits to the data show a larger length constant for excitatory covariances than for inhibitory ones (*Figure 4A*). This is qualitatively in line with the prediction of *Equation (2)* given the – by tendency – longer reach of excitatory connections compared to inhibitory ones, as derived from morphological constraints (*Reimann et al., 2017*, Fig. S2). In the dynamically balanced critical regime, however, the predicted difference in slope for all three fits is practically negligible. Therefore, we performed a second fit where the slope of the three exponentials is constrained to be identical (*Figure 4B*). The error of this fit is only marginally larger than the ones of fitting individual slopes (*Figure 4C*). This shows that differences in slopes are hardly detectable given the empirical evidence, thus confirming the predictions of the theory given by *Equation (1)* and *Equation (2)*.

## Firing rates alter connectivity-dependent covariance patterns

Since covariances measure the coordination of temporal fluctuations around the individual neurons' mean firing rates, they are determined by how strong a neuron transmits such fluctuations from input to output (*Abeles, 1991*). To leading order this is explained by linear-response theory (*Ginzburg and Sompolinsky, 1994*; *Lindner et al., 2005*; *Pernice et al., 2011*; *Tetzlaff et al., 2012*): How strongly a neuron reacts to a small change in its input depends on its dynamical state, foremost the mean and variance of its total input, called 'working point' in the following. If a neuron receives almost no input, a small perturbation in the input will not be able to make the neuron fire. If the neuron receives a large input, a small perturbation will not change the firing rate either, as the neuron is already saturated. Only in the intermediate regime the neuron is susceptible to small deviations of the input. Mathematically, this behavior is described by the gain of the neuron, which is the derivative of the input-output relation (*Abeles, 1991*). Due to the non-linearity of the input-output relation, the gain is vanishing for very small and very large inputs and non-zero in the intermediate regime. How strongly a perturbation in the input to one neuron affects one of the subsequent neurons therefore not only depends on the synaptic weight $J$ but also on the gain $S$ and thereby the working point. This relation is captured by the effective connectivity $W = S \cdot J$. What is the consequence of the dynamical interaction among neurons depending on the working point? Can it be used to reshape the low-dimensional manifold, the collective coordination between neurons?

The first part of this study finds that long-range coordination can be achieved in a network with short-range random connections if effective connections are sufficiently strong. Alteration of the working point, for example by a different external input level, can affect the covariance structure: The pattern of coordination between individual neurons can change, even though the anatomical connectivity remains the same. In this way, routing of information through the network can be adapted dynamically on a mesoscopic scale. This is a crucial difference of such coordination as opposed to coordination imprinted by complex but static connection patterns.

Here, we first illustrate this concept by simulations of a network of 2000 sparsely connected threshold-linear (ReLU) rate neuron models that receive Gaussian white noise inputs centered around neuron-specific non-zero mean values (see Materials and methods and Appendix 1 Section 14 for more details). The ReLU activation function thereby acts as a simple model for the vanishing gain for neurons with too low input levels. Note that in cortical-like scenarios with low firing rates, neuronal working points are far away from the high-input saturation discussed above, which is therefore neglected by the choice of the ReLU activation function. For independent and stationary external inputs covariances between neurons are solely generated inside the network via the sparse and random recurrent connectivity. External inputs only have an indirect impact on the covariance structure by setting the working point of the neurons.

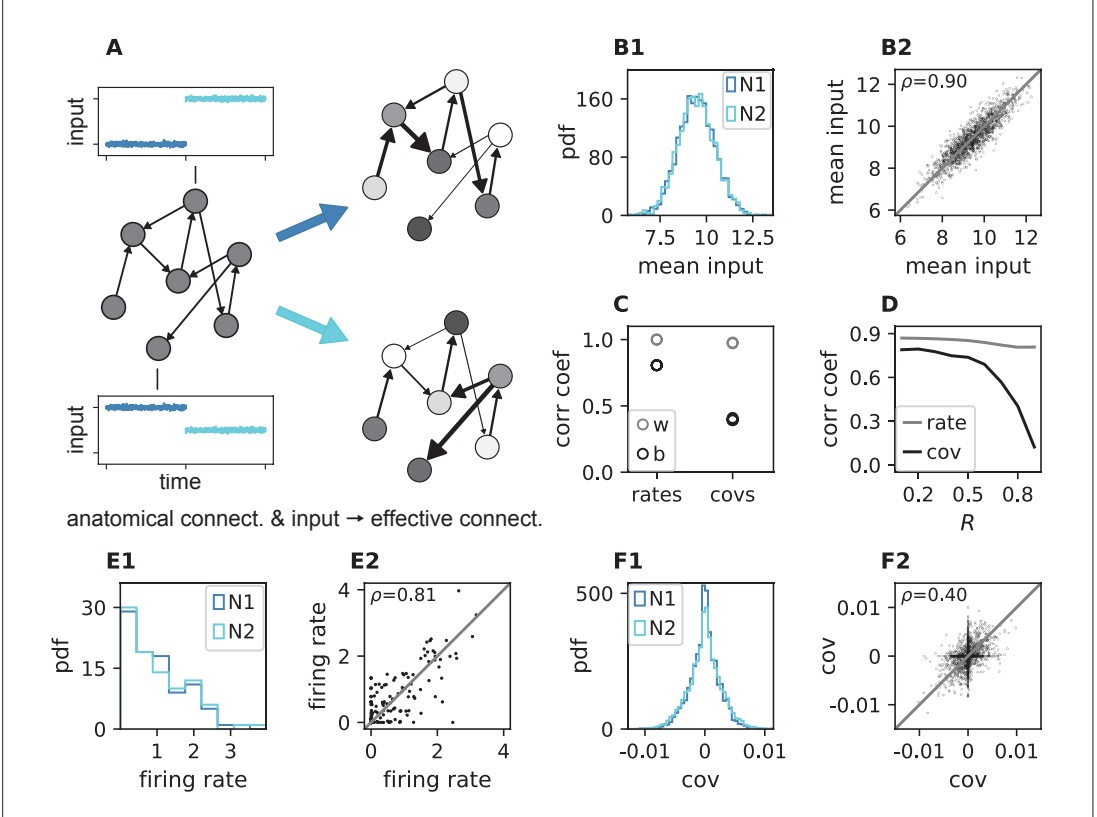

**Figure 5.** Changes in effective connectivity modify coordination patterns. (**A**) Visualization of effective connectivity: A sparse random network with given structural connectivity (left network sketch) is simulated with two different input levels for each neuron (depicted by insets), resulting in different firing rates (grayscale in right network sketches) and therefore different effective connectivities (thickness of connections) in the two simulations. Parameters can be found in *Appendix 1—table 4*. (**B1**) Histogram of input currents across neurons for the two simulations (N1 and N2). (**B2**) Scatter plot of inputs to subset of 1500 corresponding neurons in the first and the second simulation (Pearson correlation coefficient $\rho = 0.90$). (**C**) Correlation coefficients of rates and of covariances between the two simulations (b, black) and within two epochs of the same simulation (w, gray). (**D**) Correlation coefficient of rates (gray) and covariances (black) between the two simulations as a function of the spectral bound $R$. (**E1**) Distribution of rates in the two simulations (excluding silent neurons with $|\text{rate}| < 10^{-3}$). (**E2**) Scatter plot of rates in the first compared to the second simulation (Pearson correlation coefficient $\rho = 0.81$). (**F1**) Distribution of covariances in the two simulation (excluding silent neurons). (**F2**) Scatter plot of sample of 5000 covariances in first compared to the second simulation (Pearson correlation coefficient $\rho = 0.40$). Here silent neurons are included (accumulation of markers on the axes). Other parameters: number of neurons $N = 2000$, connection probability $p = 0.1$, spectral bound for panels $B$, $C$, $E$, $F$ is $R = 0.8$.

The online version of this article includes the following source data for figure 5:

**Source data 1.** Code and data.

We simulate two networks with identical structural connectivity and identical external input fluctuations, but small differences in mean external inputs between corresponding neurons in the two simulations (*Figure 5A*). These small differences in mean external inputs create different gains and firing rates and thereby differences in effective connectivity and covariances. Since mean external inputs are drawn from the same distribution in both simulations (*Figure 5B*), the overall distributions of firing rates and covariances across all neurons are very similar (*Figure 5E1, F1*). But individual neurons' firing rates do differ (*Figure 5E2*). For the simple ReLU activation used here, we in particular observe neurons that switch between non-zero and zero firing rate between the two simulations. This resulting change of working points substantially affects the covariance patterns (*Figure 5F2*): Differences in firing rates and covariances between the two simulations are significantly larger than the differences across two different epochs of the same simulation (*Figure 5C*). The larger the spectral bound, the more sensitive are the intrinsically generated covariances to the changes in firing rates (*Figure 5D*). Thus, a small offset of individual firing rates is an effective parameter to control network-wide coordination among neurons. As the input to the local network can be changed momentarily, we predict that in the dynamically balanced critical regime coordination patterns should be highly dynamic.

## Coordination patterns in motor cortex depend on behavioral context

In order to test the theoretical prediction in experimental data, we analyze parallel spiking activity from macaque motor cortex, recorded during a reach-to-grasp experiment (*Riehle et al., 2013*; *Brochier et al., 2018*). In contrast to the resting state, where the animal was in an idling state, here the animal is involved in a complex task with periods of different cognitive and behavioral conditions (*Figure 6A*). We compare two epochs in which the animal is requested to wait and is sitting still but which differ in cognitive conditions. The first epoch is a starting period (S), where the monkey has self-initiated the behavioral trial and is attentive because it is expecting a cue. The second epoch is a preparatory period (P), where the animal has just received partial information about the upcoming trial and is waiting for the missing information and the GO signal to initiate the movement.

Within each epoch, S or P, the neuronal firing rates are mostly stationary, likely due to the absence of arm movements which create relatively large transient activities in later epochs of the task, which are not analyzed here (see Appendix 1 Section 3). The overall distributions of the firing rates are comparable for epochs S and P, but the firing rates are distributed differently across the individual neurons: *Figure 6C* shows one example session of monkey N, where the changes in firing rates between the two epochs are visible in the spread of markers around the diagonal line in panel *C2*. To assess the extent of these changes, we split each epoch, S and P, into two disjoint sub-periods, S1/S2 and P1/P2 (*Figure 6A*). We compute the correlation coefficient between the firing rate vectors of two sub-periods of different epochs ('between' markers in *Figure 6E*) and compare it to the correlation coefficient between the firing rate vectors of two sub-periods of the same epoch ('within' markers): Firing rate vectors in S1 are almost perfectly correlated with firing rate vectors in S2 ($\rho \approx 1$ for all of the five/eight different recording sessions from different recording days for monkey E/N, similarly for P1 and P2), confirming stationarity investigated in Appendix 1 Section 3. Firing rate vectors in S1 or S2, however, show significantly lower correlation to firing rate vectors in P1 and P2, confirming a significant change in network state between epochs S and P (*Figure 6E*).

The mechanistic model in the previous section shows a qualitatively similar scenario (*Figure 5C and E*). By construction it produces different firing rate patterns in the two simulations. While the model is simplistic and in particular not adapted to quantitatively reproduce the experimentally observed activity statistics, its simulations and our underlying theory make a general prediction: Differences in firing rates impact the effective connectivity between neurons and thereby evoke even larger differences in their coordination if the network is operating in the dynamically balanced critical regime (*Figure 5D*). To check this prediction, we repeat the correlation analysis between the two epochs, which we described above for the firing rates, but this time for the covariance patterns. Despite similar overall distributions of covariances in S and P (*Figure 6D1*), covariances between individual neuron pairs are clearly different between S and P: *Figure 6B* shows the covariance pattern for one representative reference neuron in one example recording session of monkey N. In both epochs, this covariance pattern has a salt-and-pepper structure as for the resting state data in *Figure 1D*. Yet, neurons change their individual coordination: a large number of neuron pairs even changes from positive covariance values to negative ones and vice versa. These neurons fire cooperatively in one epoch of the task while they show antagonistic firing in the other epoch. The covariances of all neuron pairs of that particular recording session are shown in *Figure 6D2*. Markers in the upper left and lower right quadrant show neuron pairs that switch the sign of their coordination (45 % of all neuron pairs). The extent of covariance changes between epochs is again quantified by correlation coefficients between the covariance patterns of two sub-periods (*Figure 6F*). As for the firing rates, we find rather large correlations between covariance patterns in S1 and S2 as well as between covariance patterns in P1 and P2. Note, however, that correlation coefficients are around 0.8 rather than 1, presumably since covariance estimates from 200 ms periods are noisier than firing rate estimates. The covariance patterns in S1 or S2 are, however, significantly more distinct from covariance patterns in P1 and P2, with correlation coefficients around 0.5 (*Figure 6F*). This more pronounced change of covariances compared to firing rates is predicted by a network whose effective connectivity has a large spectral bound, in the dynamically balanced critical state. In particular, the theory provides a mechanistic explanation for the different coordination patterns between neurons on the mesoscopic scale (range of a Utah array), which are observed in the two states S and P (*Figure 6B*). The coordination between neurons is thus considerably reshaped by the behavioral condition.

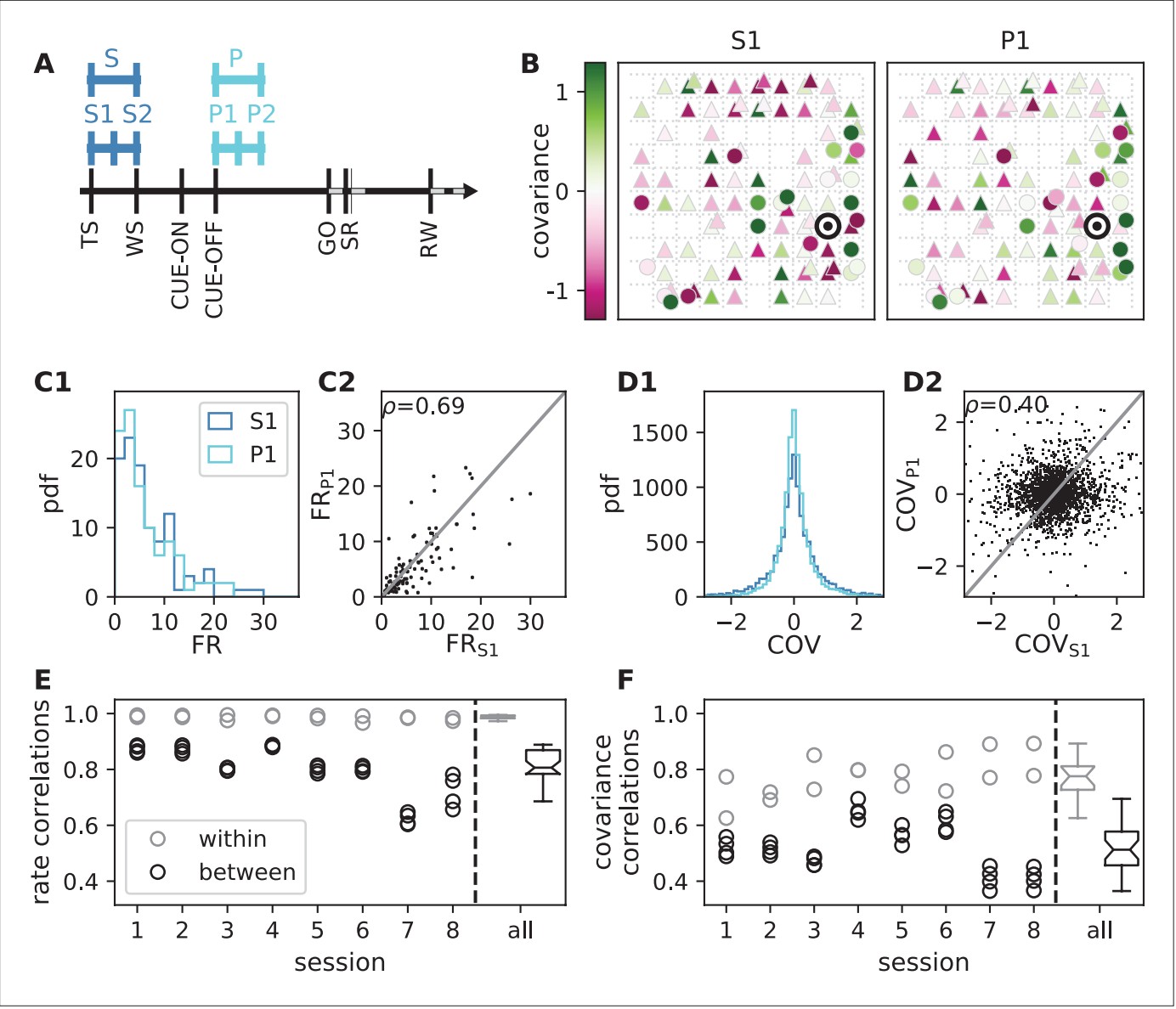

**Figure 6.** Behavioral condition reshapes mesoscopic neuronal coordination. (**A**) Trial structure of the reach-to-grasp experiment (*Brochier et al., 2018*). Blue segments above the time axis indicate data pieces at trial start (dark blue: S (S1+ S2)) and during the preparatory period (light blue: P (P1+ P2)). (**B**) Salt-and-pepper structure of covariance during two different epochs (S1 and P1) of one recording session of monkey N (151 trials, 106 single units, *Figure 1* for recording setup). For some neurons, the covariance completely reverses, while in the others it does not change. Inhibitory reference neuron indicated by black circle. (**C1**) Distributions of firing rates during S1 and P1. (**C2**) Scatter plot comparing firing rates in S1 and P1 (Pearson correlation coefficient $\rho = 0.69$). (**D1**/**D2**) Same as panels *C1*/*C2*, but for covariances (Pearson correlation coefficient $\rho = 0.40$). (**E**) Correlation coefficient of firing rates across neurons in different epochs of a trial for eight recorded sessions. Correlations between sub-periods of the same epoch (S1-S2, P1-P2; within-epoch, gray) and between sub-periods of different epochs (Sx-Py; between-epochs, black). Data shown in panels B-D is from session 8. Box plots to the right of the black dashed line show distributions obtained after pooling across all analyzed recording sessions per monkey. The line in the center of each box represents the median, box's area represents the interquartile range, and the whiskers indicate minimum and maximum of the distribution (outliers excluded). Those distributions differ significantly (Student t-test, two-sided, $p \ll 0.001$). (**F**) Correlation coefficient of covariances, analogous to panel E. The distributions of values pooled across sessions also differ significantly (Student t-test, two-sided, $p \ll 0.001$). For details of the statistical tests, see Materials and methods. Details on number of trials and units in each recording session are provided in *Appendix 1—table 1*.

The online version of this article includes the following source data for figure 6:

**Source data 1.** Code and data.

## Discussion

In this study, we investigate coordination patterns of many neurons across mesoscopic distances in macaque motor cortex. We show that these patterns have a salt-and-pepper structure, which can be explained by a network model with a spatially dependent random connectivity operating in a dynamically balanced critical state. In this state, cross-covariances are shaped by a large number of parallel, multi-synaptic pathways, leading to interactions reaching far beyond the range of direct connections. Strikingly, this coordination on the millimeter scale is only visible if covariances are resolved on the level of individual neurons; the population mean of covariances quickly decays with distance and is overall very small. In contrast, the variance of covariances is large and predominantly decreases exponentially on length scales of up to several millimeters, even though direct connections typically only reach a few hundred micrometers.

Since the observed coordination patterns are determined by the effective connectivity of the network, they are dynamically controllable by the network state; for example, due to modulations of neuronal firing rates. Parallel recordings in macaque motor cortex during resting state and in different epochs of a reach-to-grasp task confirm this prediction. Simulations indeed exhibit a high sensitivity of coordination patterns to weak modulations of the individual neurons' firing rates, providing a plausible mechanism for these dynamic changes.

Models of balanced networks have been investigated before (*van Vreeswijk and Sompolinsky, 1996*; *Brunel, 2000*; *Renart et al., 2010*; *Tetzlaff et al., 2012*) and experimental evidence for cortical networks operating in the balanced state is overwhelming (*Okun and Lampl, 2008*; *Reinhold et al., 2015*; *Dehghani et al., 2016*). Excess of inhibition in such networks yields stable and balanced population-averaged activities as well as low average covariances (*Tetzlaff et al., 2012*). Recently, the notion of balance has been combined with criticality in the dynamically balanced critical state that results from large heterogeneity in the network connectivity (*Dahmen et al., 2019*). Here, we focus on another ubiquitous property of cortical networks, their spatial organization, and study the interplay between balance, criticality, and spatial connectivity in networks of excitatory and inhibitory neurons. We show that in such networks, heterogeneity generates disperse covariance structures between individual neurons on large length-scales with a salt-and-pepper structure.

Spatially organized balanced network models have been investigated before in the limit of infinite network size, as well as under strong and potentially correlated external drive, as is the case, for example, in primary sensory areas of the brain (*Rosenbaum et al., 2017*; *Baker et al., 2019*). In this scenario, intrinsically generated contributions to covariances are much smaller than external ones. Population-averaged covariances then fulfill a linear equation, called the 'balance condition' (*van Vreeswijk and Sompolinsky, 1996*; *Hertz, 2010*; *Renart et al., 2010*; *Rosenbaum and Doiron, 2014*), that predicts a non-monotonous change of population-averaged covariances with distance (*Rosenbaum et al., 2017*). In contrast, we here consider covariances on the level of individual cells in finite-size networks receiving only weak inputs. While we cannot strictly rule out that the observed covariance patterns in motor cortex are a result of very specific external inputs to the recorded local network, we believe that the scenario of weak external drive is more suitable for non-sensory brain areas, such as, for example, the motor cortex in the resting state conditions studied here. Under such conditions, covariances have been shown to be predominantly generated locally rather than from external inputs: *Helias et al., 2014* investigated intrinsic and extrinsic sources of covariances in ongoing activity of balanced networks and found that for realistic sizes of correlated external populations the major contribution to covariances is generated from local network interactions (Figure 7a in *Helias et al., 2014*). *Dahmen et al., 2019* investigated the extreme case, where the correlated external population is of the same size as the local population (Fig. S6 in *Dahmen et al., 2019*). Despite sizable external input correlations projected onto the local circuit via potentially strong afferent connections, the dependence of the statistics of covariances on the spectral bound of the local recurrent connectivity is predicted well by the theory that neglects correlated external inputs (see supplement section 3 in *Dahmen et al., 2019*).

Our analysis of covariances on the single-neuron level goes beyond the balance condition and requires the use of field-theoretical techniques to capture the heterogeneity in the network (*Dahmen et al., 2019*; *Helias and Dahmen, 2020*). It relies on linear-response theory, which has previously been shown to faithfully describe correlations in balanced networks of nonlinear (spiking) units (*Tetzlaff et al., 2012*; *Trousdale et al., 2012*; *Pernice et al., 2012*; *Grytskyy et al., 2013*; *Helias et al.,*

*2013*; *Dahmen et al., 2019*). These studies mainly investigated population-averaged correlations with small spectral bounds of the effective connectivity. Subsequently, *Dahmen et al., 2019* showed the quantitative agreement of this linear-response theory for covariances between individual neurons in networks of spiking neurons for the whole range of spectral bounds, including the dynamically balanced critical regime. The long-range coordination studied in the current manuscript requires the inclusion of spatially non-homogeneous coupling to analyze excitatory-inhibitory random networks on a two-dimensional sheet with spatially decaying connection probabilities. This new theory allows us to derive expressions for the spatial decay of the variance of covariances. We primarily evaluate these expressions in the long-range limit, which agrees well with simulations for distances $x > 2d \sim \mathcal{O}(1\,\text{mm})$, which is fulfilled for most distances on the Utah array (*Figure 3*, *Appendix 1—figure 7*). For these distances, we find that the decay of covariances is dominated by a simple exponential law. Unexpectedly, its decay constant is essentially determined by only two measures, the spectral bound of the effective connectivity, and the length scale of direct connections. The length scale of covariances diverges when approaching the breakdown of linear stability. In this regime, differences in covariances induced by differences in length scales of excitatory and inhibitory connections become negligible. The predicted emergence of a single length scale of covariances is consistent with our data.

This study focuses on local and isotropic connection profiles to show that long-range coordination does not rely on specific connection patterns but can result from the network state alone. Alternative explanations for long-range coordination are based on specifically imprinted network structures: Anisotropic local connection profiles have been studied and shown to create spatio-temporal sequences (*Spreizer et al., 2019*). Likewise, embedded excitatory feed-forward motifs and cell assemblies via excitatory long-range patchy connections (*DeFelipe et al., 1986*) can create positive covariances at long distances (*Diesmann et al., 1999*; *Litwin-Kumar and Doiron, 2012*). Yet, these connections cannot provide an explanation for the large negative covariances between excitatory neurons at long distances (see e.g. *Figure 1D*). Long-range connectivity, for example arising from a salt-and-pepper organization of neuronal selectivity with connections preferentially targeting neurons with equal selectivity (*Ben-Yishai et al., 1995*; *Hansel and Sompolinsky, 1998*; *Roxin et al., 2005*; *Blumenfeld et al., 2006*), would produce salt-and-pepper covariance patterns even in networks with small spectral bounds where interactions are only mediated via direct connections. However, in this scenario, one would expect that neurons which have similar selectivity would throughout show positive covariance due to their mutual excitatory connections and due to the correlated input they receive. Yet, when analyzing two different epochs of the reach-to-grasp task, we find that a large fraction of neuron pairs actually switches from being significantly positively correlated to negatively correlated and vice versa (see *Figure 6D2*, upper left and lower right quadrant). This state-dependence of covariances is in line with the here suggested mechanism of long-range coordination by indirect interactions: Such indirect interactions depend on the effective strengths of various connections and can therefore change considerably with network state. In contrast, correlations due to imprinted network structures are static, so that a change in gain of the neurons will either strengthen or weaken the specific activity propagation, but it will not lead to a change of the sign of covariances that we see in our data. The static impact of these connectivity structures on covariances could nevertheless in principle be included in the presented formalism. Long-range coordination can also be created from short-range connections with random orientations of anisotropic local connection profiles (*Smith et al., 2018*). This finding can be linked to the emergence of tuning maps in the visual cortex. The mechanism is similar to ours in that it uses nearly linearly unstable modes that are determined by spatial connectivity structures and heterogeneity. Given the different source of heterogeneity, the modes and corresponding covariance patterns are different from the ones discussed here: Starting from fully symmetric networks with corresponding symmetric covariance patterns, *Smith et al., 2018* found that increasing heterogeneity (anisotropy) yields more randomized, but still patchy regions of positive and negative covariances that are in line with low-dimensional activity patterns found in visual cortex. In motor cortex we instead find salt-and-pepper patterns that can be explained in terms of heterogeneity through sparsity. We provide the theoretical basis and explicit link between connectivity eigenspectra and covariances and show that heterogeneity through sparsity is sufficient to generate the dynamically balanced critical state as a simple explanation for the broad distribution of covariances in motor cortex, the salt-and-pepper structure of coordination, its long spatial range, and its sensitive dependence on the network state. Note that both mechanisms of long-range coordination, the one

studied in *Smith et al., 2018* and the one presented here, rely on the effective connectivity for the network to reside in the dynamically balanced critical regime. The latter regime is, however, not just one single point in parameter space, but an extended region that can be reached via a multitude of control mechanisms for the effective connectivity, for example by changing neuronal gains (*Salinas and Sejnowski, 2001a*; *Salinas and Sejnowski, 2001b*), synaptic strengths (*Sompolinsky et al., 1988*), and network microcircuitry (*Dahmen et al., 2021*).

What are possible functional implications of the coordination on mesoscopic scales? Recent work demonstrated activity in motor cortex to be organized in low-dimensional manifolds (*Gallego et al., 2017*; *Gallego, 2018*; *Gallego et al., 2020*). Dimensionality reduction techniques, such as PCA or GPFA (*Yu et al., 2009*), employ covariances to expose a dynamical repertoire of motor cortex that is comprised of neuronal modes. Previous work started to analyze the relation between the dimensionality of activity and connectivity (*Aljadeff et al., 2015*; *Aljadeff et al., 2016*; *Mastrogiuseppe and Ostojic, 2018*; *Dahmen et al., 2019*; *Dahmen et al., 2021*; *Hu and Sompolinsky, 2020*), but only in spatially unstructured networks, where each neuron can potentially be connected to any other neuron. The majority of connections within cortical areas, however, stems from local axonal arborizations (*Schnepel et al., 2015*). Here, we add this biological constraint and demonstrate that these networks, too, support a dynamically balanced critical state. This state in particular exhibits neural modes which are spanned by neurons spread across the experimentally observed large distances. In this state a small subset of modes that are close to the point of instability dominates the variability of the network activity and thus spans a low-dimensional neuronal manifold. As opposed to specifically designed connectivity spectra via plasticity mechanisms (*Hennequin et al., 2014*) or low-rank structures embedded into the connectivity (*Mastrogiuseppe and Ostojic, 2018*), the dynamically balanced critical state is a mechanism that only relies on the heterogeneity which is inherent to sparse connectivity and abundant across all brain areas.

While we here focus on covariance patterns in stationary activity periods, the majority of recent works studied transient activity during motor behavior (*Gallego et al., 2017*). How are stationary and transient activities related? During stationary ongoing activity states, covariances are predominantly generated intrinsically (*Helias et al., 2014*). Changes in covariance patterns therefore arise from changes in the effective connectivity via changes in neuronal gains, as demonstrated here in the two periods of the reach-to-grasp experiment and in our simulations for networks close to criticality (*Figure 5D*). During transient activity, on top of gain changes, correlated external inputs may directly drive specific neural modes to create different motor outputs, thereby restricting the dynamics to certain subspaces of the manifold. In fact, *Elsayed et al., 2016* reported that the covariance structures during movement preparation and movement execution are unrelated and corresponding to orthogonal spaces within a larger manifold. Also, *Luczak et al., 2009* studied auditory and somatosensory cortices of awake and anesthetized rats during spontaneous and stimulus-evoked conditions and found that neural modes of stimulus-evoked activity lie in subspaces of the neural manifold spanned by the spontaneous activity. Similarly, visual areas V1 and V2 seem to exploit distinct subspaces for processing and communication (*Semedo et al., 2019*), and motor cortex uses orthogonal subspaces capturing communication with somatosensory cortex or behavior-generating dynamics (*Perich et al., 2021*). *Gallego, 2018* further showed that manifolds are not identical, but to a large extent preserved across different motor tasks due to a number of task-independent modes. This leads to the hypothesis that the here described mechanism for long-range cooperation in the dynamically balanced critical state provides the basis for low-dimensional activity by creating such spatially extended neural modes, whereas transient correlated inputs lead to their differential activation for the respective target outputs. The spatial spread of the neural modes thereby leads to a distributed representation of information that may be beneficial to integrate information into different computations that take place in parallel at various locations. Further investigation of these hypotheses is an exciting endeavor for the years to come.

# Materials and methods
## Experimental design and statistical analysis
Two adult macaque monkeys (monkey E - female, and monkey N - male) are recorded in behavioral experiments of two types: resting state and reach-to-grasp. The recordings of neuronal activity in

motor and pre-motor cortex (hand/arm region) are performed with a chronically implanted $4\text{x}4\,\text{mm}^2$ Utah array (Blackrock Microsystems). Details on surgery, recordings, spike sorting and classification of behavioral states can be found in *Riehle et al., 2013*; *Riehle et al., 2018*; *Brochier et al., 2018*; *Dąbrowska et al., 2020*. All animal procedures were approved by the local ethical committee (C2EA 71; authorization A1/10/12) and conformed to the European and French government regulations.

## Resting state data

During the resting state experiment, the monkey is seated in a primate chair without any task or stimulation. Registration of electrophysiological activity is synchronized with a video recording of the monkey's behavior. Based on this, periods of 'true resting state' (RS), defined as no movements and eyes open, are chosen for the analysis. Eye movements and minor head movements are included. Each monkey is recorded twice, with a session lasting approximately 15 and 20 min for monkeys E (sessions E1 and E2) and N (sessions N1 and N2), respectively, and the behavior is classified by visual inspection with single second precision, resulting in 643 and 652 s of RS data for monkey E and 493 and 502 s of RS data for monkey N.

## Reach-to-grasp data

In the reach-to-grasp experiment, the monkeys are trained to perform an instructed delayed reach-to-grasp task to obtain a reward. Trials are initiated by a monkey closing a switch (TS, trial start). After 400 ms a diode is illuminated (WS, warning signal), followed by a cue after another 400 ms(CUE-ON), which provides partial information about the upcoming trial. The cue lasts 300 ms and its removal (CUE-OFF) initiates a 1 s preparatory period, followed by a second cue, which also serves as GO signal. Two epochs, divided into 200 ms sub-periods, within such defined trials are chosen for analysis: the first 400 ms after TS (starting period, S1 and S2), and the 400 ms directly following CUE-OFF (preparatory period, P1 and P2) (*Figure 6a*). Five selected sessions for monkey E and eight for monkey N provide a total of 510 and 1111 correct trials, respectively. For detailed numbers of trials and single units per recording session see *Appendix 1—table 1*.

## Separation of putative excitatory and inhibitory neurons

Offline spike-sorted single units (SUs) are separated into putative excitatory (broad-spiking) and putative inhibitory (narrow-spiking) based on their spike waveform width (*Barthó et al., 2004*; *Kaufman et al., 2010*; *Kaufman et al., 2013*; *Peyrache, 2012*; *Peyrache and Destexhe, 2019*). The width is defined as the time (number of data samples) between the trough and peak of the waveform. Widths of all average waveforms from all selected sessions (both resting state and reach-to-grasp) per monkey are collected. Thresholds for 'broadness' and 'narrowness' are chosen based on the monkey-specific distribution of widths, such that intermediate values stay unclassified. For monkey E the thresholds are 0.33 ms and 0.34 ms and for monkey N 0.40 ms and 0.41 ms. Next, a two-step classification is performed session by session. Firstly, the thresholds are applied to average SU waveforms. Secondly, the thresholds are applied to SU single waveforms and a percentage of single waveforms pre-classified as the same type as the average waveform is calculated. SU for which this percentage is high enough are marked classified. All remaining SUs are grouped as unclassified. We verify the robustness of our results with respect to changes in the spike sorting procedure in Appendix 1 Section 2.

Synchrofacts, that is, spike-like synchronous events across multiple electrodes at the sampling resolution of the recording system (1/30 ms) (*Torre, 2016*), are removed. In addition, only SUs with a signal-to-noise ratio (*Hatsopoulos et al., 2004*) of at least 2.5 and a minimal average firing rate of 1 Hz are considered for the analysis, to ensure enough and clean data for valid statistics.

## Statistical analysis

All RS periods per resting state recording are concatenated and binned into 1 s bins. Next, pairwise covariances of all pairs of SUs are calculated according to the following formula:

$$\text{COV}(i,j) = \frac{\langle b_i - \mu_i, b_j - \mu_j \rangle}{l-1}\,, \tag{3}$$

with $b_i$, $b_j$ - binned spike trains, $\mu_i$, $\mu_j$ being their mean values, $l$ the number of bins, and $\langle x, y \rangle$ the scalar product of vectors $x$ and $y$. Obtained values are broadly distributed, but low on average in every recorded session: in session E1 E-E pairs: $0.19 \pm 1.10$ (M±SD), E-I: $0.24 \pm 2.31$, I-I: $0.90 \pm 4.19$, in session E2 E-E: $0.060 \pm 1.332$, E-I $0.30 \pm 2.35$, I-I $1.0 \pm 4.5$, in session N1 E-E $0.24 \pm 1.13$, E-I $0.66 \pm 2.26$, I-I $2.4 \pm 4.9$, in session N2 E-E $0.41 \pm 1.47$, E-I $1.0 \pm 3.1$, I-I $3.9 \pm 7.3$.

To explore the dependence of covariance on the distance between the considered neurons, the obtained values are grouped according to distances between electrodes on which the neurons are recorded. For each distance the average and variance of the obtained distribution of cross-covariances is calculated. The variance is additionally corrected for bias due to a finite number of measurements (**Dahmen et al., 2019**). In most of cases, the correction does not exceed 0.01%.

In the following step, exponential functions $y = a\,e^{-\frac{x}{d}}$ are fitted to the obtained distance-resolved variances of cross-covariances ($y$ corresponding to the variance and $x$ to distance between neurons), which yields a pair of values $(a, d)$. The least squares method implemented in the Python scipy.optimize module (SciPy v.1.4.1) is used. Firstly, three independent fits are performed to the data for excitatory-excitatory, excitatory-inhibitory, and inhibitory-inhibitory pairs. Secondly, analogous fits are performed, with the constraint that the decay constant $d$ should be the same for all three curves.

Covariances in the reach-to-grasp data are calculated analogously but with different time resolution. For each chosen sub-period of a trial, data are concatenated and binned into 200 ms bins, meaning that the number of spikes in a single bin corresponds to a single trial. The mean of these counts normalized to the bin width gives the average firing rate per SU and sub-period. The pairwise covariances are calculated according to *Equation (3)*. To assess the similarity of neuronal activity in different periods of a trial, Pearson product-moment correlation coefficients are calculated on vectors of SU-resolved rates and pair-resolved covariances. Correlation coefficients from all recording sessions per monkey are separated into two groups: using sub-periods of the same epoch (*within*-epoch), and using sub-periods of different epochs of a trial (*between*-epochs). These groups are tested for differences with significance level $\alpha = 0.05$. Firstly, to check if the assumptions for parametric tests are met, the normality of each obtained distribution is assessed with a Shapiro-Wilk test, and the equality of variances with an *F*-test. Secondly, a *t*-test is applied to compare within- and between-epochs correlations of rates or covariances. Since there are two *within* and four *between* correlation values per recording session, the number of degrees of freedom equals: $df = (N_{\text{sessions}} \cdot 2 - 1) + (N_{\text{sessions}} \cdot 4 - 1)$, which is 28 for monkey E and 46 for monkey N. To estimate the confidence intervals for obtained differences, the mean difference between groups $m$ and their pooled standard deviation $s$ are calculated for each comparison

$$m = m_{\text{within}} - m_{\text{between}}\,, \qquad s = \sqrt{\frac{(N_{\text{within}} - 1)s_{\text{within}}^2 + (N_{\text{between}} - 1)s_{\text{between}}^2}{N_{\text{within}} + N_{\text{between}} - 2}}\,,$$

with $m_{\text{within}}$ and $m_{\text{between}}$ being the mean, $s_{\text{within}}$ and $s_{\text{between}}$ the standard deviation and $N_{\text{within}}$ and $N_{\text{between}}$ the number of *within*- and *between*-epoch correlation coefficient values, respectively.

This results in 95 % confidence intervals $m \pm t(df) \cdot s$ of $0.192 \pm 0.093$ for rates and $0.32 \pm 0.14$ for covariances in monkey E and $0.19 \pm 0.14$ for rates and $0.26 \pm 0.17$ for covariances in monkey N.

For both monkeys the *within*-epoch rate-correlations distribution does not fulfill the normality assumption of the *t*-test. We therefore perform an additional non-parametric Kolmogorov-Smirnov test for the rate comparison. The differences are again significant; for monkey E $D = 1.00$, $p = 6.66 \cdot 10^{-8}$; for monkey N $D = 1.00$, $p = 8.87 \cdot 10^{-13}$.

For all tests we use the implementations from the Python scipy.stats module (SciPy v.1.4.1).

## Mean and variance of covariances for a two-dimensional network model with excitatory and inhibitory populations

The mean and variance of covariances are calculated for a two-dimensional network consisting of one excitatory and one inhibitory population of neurons. The connectivity profile $p(\boldsymbol{x})$, describing the probability of a neuron having a connection to another neuron at distance $\boldsymbol{x}$, decays with distance. We assume periodic boundary conditions and place the neurons on a regular grid (*Figure 3A*), which

imposes translation and permutation symmetries that enable the derivation of closed-form solutions for the distance-dependent mean and variance of the covariance distribution. These simplifying assumptions are common practice and simulations show that they do not alter the results qualitatively.

Our aim is to find an expression for the mean and variance of covariances as functions of distance between two neurons. While the theory in *Dahmen et al., 2019* is restricted to homogeneous connections, understanding the spatial structure of covariances here requires us to take into account the spatial structure of connectivity. Field-theoretic methods, combined with linear-response theory, allow us to obtain expressions for the mean covariance $\bar{c}$ and variance of covariance $\overline{\delta c^2}$

$$\bar{c} = [\mathbf{1} - M]^{-1} \frac{D}{1 - R^2} [\mathbf{1} - M]^{-\mathrm{T}}, \quad \overline{\delta c^2} = [\mathbf{1} - S]^{-1} \left( \frac{D}{1 - R^2} \right)^2 [\mathbf{1} - S]^{-\mathrm{T}}, \tag{4}$$

with identity matrix $\mathbf{1}$, mean $M$ and variance $S$ of connectivity matrix $W$, input noise strength $D$, and spectral bound $R$. Since $M$ and $S$ have a similar structure, the mean and variance can be derived in the same way, which is why we only consider variances in the following.

To simplify *Equation (4)*, we need to find a basis in which $S$, and therefore also $A = \mathbf{1} - S$, is diagonal. Due to invariance under translation, the translation operators $T$ and the matrix $S$ have common eigenvectors, which can be derived using that translation operators satisfy $T^N = 1$, where $N$ is the number of lattice sites in $x$- or $y$-direction (see Appendix 1). Projecting onto a basis of these eigenvectors shows that the eigenvalues $s_k$ of $S$ are given by a discrete two-dimensional Fourier transform of the connectivity profile $s_k \propto \sum_x p(x) e^{-ikx}$.

Expressing $A^{-1}$ in the eigenvector basis yields $A^{-1}(x) = \mathbf{1} + B(x)$, where $B(x)$ is a discrete inverse Fourier transform of the kernel $s_k/(1 - s_k)$. Assuming a large network with respect to the connectivity profiles allows us to take the continuum limit

$$B(x) = \frac{1}{(2\pi)^2} \int \mathrm{d}^2 k \, \frac{s(k)}{1 - s(k)} e^{ikx} \quad .$$

As we are only interested in the long-range behavior, which corresponds to $|x| \to \infty$, or $|k| \to 0$, respectively, we can approximate the Fourier kernel around $|k| \approx 0$ by a rational function, quadratic in the denominator, using a Padé approximation. This allows us to calculate the integral which yields

$$B(x) \propto K_0(-|x|/d_{\mathrm{eff}}) \quad ,$$

where $K_0(x)$ denotes the modified Bessel function of second kind and zeroth order (*Olver et al., 2010*), and the effective decay constant $d_{\mathrm{eff}}$ is given by *Equation (1)*. In the long-range limit, the modified Bessel function behaves like

$$B(x) \stackrel{|x| \to \infty}{\propto} \frac{\exp(-|x|/d_{\mathrm{eff}})}{\sqrt{|x|}} \quad .$$

Writing *Equation (4)* in terms of $B(x)$ gives

$$\overline{\delta c^2}(x) = \left( \frac{D}{1 - R^2} \right)^2 \left[ \delta(|x|) + B(x) + (B * * B)(x) \right] \quad ,$$

with the double asterisk denoting a two-dimensional convolution. $(B * * B)(x)$ is a function proportional to the modified Bessel function of second kind and first order (*Olver et al., 2010*), which has the long-range limit

$$(B * * B)(x) \stackrel{|x| \to \infty}{\propto} \sqrt{|x|} \exp(-|x|/d_{\mathrm{eff}}) \quad .$$

Hence, the effective decay constant of the variances is given by $d_{\mathrm{eff}}$. Note that further details of the above derivation can be found in the Appendix 1 Section 4 - Section 12.

## Network model simulation

The explanation of the network state dependence of covariance patterns presented in the main text is based on linear-response theory, which has been shown to yield results quantitatively in line with non-linear network models, in particular networks of spiking leaky integrate-and-fire neuron models

(*Tetzlaff et al., 2012*; *Trousdale et al., 2012*; *Pernice et al., 2012*; *Grytskyy et al., 2013*; *Helias et al., 2013*; *Dahmen et al., 2019*). The derived mechanism is thus largely model independent. We here chose to illustrate it with a particularly simple non-linear input-output model, the rectified linear unit (ReLU). In this model, a shift of the network's working point can turn some neurons completely off, while activating others, thereby leading to changes in the effective connectivity of the network. In the following, we describe the details of the network model simulation.

We performed a simulation with the neural simulation tool NEST (*Jordan, 2019*) using the parameters listed in *Appendix 1—table 4*. We simulated a network of $N$ inhibitory neurons (*threshold_lin_rate_ipn*, *Hahne, 2017*), which follow the dynamical equation

$$\tau \frac{\mathrm{d}z_i}{\mathrm{d}t} = -z_i + \sum_j J_{ij}\nu_j + \mu_{\mathrm{ext},i} + \xi_i\sqrt{\tau}\sigma_{\mathrm{noise},i} \quad , \tag{5}$$

where $z_i$ is the input to neuron $i$, $\nu$ the output firing rate with (threshold linear activation function)

$$\nu = \phi(z) = \begin{cases} 0 & \text{for } z \leq 0 \\ z & \text{for } z > 0 \end{cases} \quad ,$$

time constant $\tau$, connectivity matrix $\boldsymbol{J}$, a constant external input $\mu_{\mathrm{ext},i}$, and uncorrelated Gaussian white noise $\langle\xi_i(t)\rangle = 0$, $\langle\xi_i(s)\xi_j(t)\rangle = \delta_{ij}\delta(s-t)$, with noise strength $\sqrt{\tau}\sigma_{\mathrm{noise},i}$. The neurons were connected using the *fixed_indegree* connection rule, with connection probability $p$, indegree $K = p \cdot N$, and delta-synapses (*rate_connection_instantaneous*) of weight $w$.

The constant external input $\mu_{\mathrm{ext},i}$ to each neuron was normally distributed, with mean $\mu_{\mathrm{ext}}$, and standard deviation $\sigma_{\mathrm{ext}}$. It was used to set the firing rates of neurons, which, via the effective connectivity, influence the intrinsically generated covariances in the network. The two parameters $\mu_{\mathrm{ext}}$ and $\sigma_{\mathrm{ext}}$ were chosen such that, in the stationary state, half of the neurons were expected to be above threshold. Which neurons are active depends on the realization of $\mu_{\mathrm{ext},i}$ and is therefore different for different networks.

To assess the distribution of firing rates, we first considered the static variability of the network and studied the stationary solution of the noise-averaged input $\langle z \rangle_{\mathrm{noise}}$, which follows from *Equation (5)* as

$$\langle z_i \rangle_{\mathrm{noise}} = \sum_j J_{ij} \langle \nu_j \rangle_{\mathrm{noise}} + \mu_{\mathrm{ext},i} \quad . \tag{6}$$

Note that $\langle\nu_j\rangle_{\mathrm{noise}} = \langle\phi(z_j)\rangle_{\mathrm{noise}}$, through the nonlinearity $\phi$, in principle depends on fluctuations of the system. This dependence is, however, small for the chosen threshold linear $\phi$, which is only nonlinear in the point $z = 0$.

The derivation of $\mu_{\mathrm{ext}}$ is based on the following mean-field considerations: according to *Equation (6)* the mean input to a neuron in the network is given by the sum of external input and recurrent input

$$\mu = \mu_{\mathrm{ext}} + \mu_{\mathrm{recurrent}} = \mu_{\mathrm{ext}} + Kw\mathrm{Mean}(\nu) \quad .$$

The variance of the input is given by

$$\sigma^2 = \sigma_{\mathrm{ext}}^2 + \sigma_{\mathrm{recurrent}}^2 = \sigma_{\mathrm{ext}}^2 + Kw^2\mathrm{Var}(\nu) \quad .$$

The mean firing rate can be calculated using the diffusion approximation (*Tuckwell, 2009*; *Amit and Tsodyks, 2009*), which is assuming a normal distribution of inputs due to the central-limit theorem, and the fact that a linear threshold neuron only fires if its input is positive

$$\begin{aligned} \mathrm{Mean}(\nu) \quad &= \int_{-\infty}^{\infty} \mathrm{d}\nu\, \mathcal{P}\left(\mu, \sigma^2, \nu\right) \nu \\ &= \int_{-\infty}^{\infty} \mathrm{d}z\, \mathcal{N}\left(\mu, \sigma^2, z\right) \phi(z) \\ &= \int_{0}^{\infty} \mathrm{d}z\, \mathcal{N}\left(\mu, \sigma^2, z\right) z \\ &= \frac{\sigma}{\sqrt{2\pi}} \exp\left(-\frac{\mu^2}{2\sigma^2}\right) + \frac{\mu}{2}\left[1 + \mathrm{erf}\left(\frac{\mu}{\sqrt{2}\sigma}\right)\right] \quad , \end{aligned}$$

where $\mathcal{P}$ denotes the probability density of the firing rate $\nu$. The variance of the firing rates is given by

$$\begin{aligned} \text{Var}(\nu) \quad &= \text{Mean}(\nu^2) - \text{Mean}(\nu)^2 \\ &= \frac{\mu^2}{4}\left[1 - \text{erf}^2\left(\frac{\mu}{\sqrt{2}\sigma}\right)\right] + \frac{\sigma^2}{2}\left[1 - \frac{1}{\pi}\exp\left(-\frac{\mu^2}{\sigma^2}\right) + \text{erf}\left(\frac{\mu}{\sqrt{2}\sigma}\right)\right] + \frac{\mu\sigma}{\sqrt{2\pi}}\text{erf}\left(\frac{\mu}{\sqrt{2}\sigma}\right) \quad . \end{aligned}$$

The number of active neurons is the number of neurons with a positive input, which we set to be equal to $N/2$

$$\frac{N}{2} \overset{!}{=} N\int_0^\infty dz\, \mathcal{N}\left(\mu, \sigma^2, z\right) = \frac{N}{2}\left[1 + \text{erf}\left(\frac{\mu}{\sqrt{2}\sigma}\right)\right] \quad ,$$

which is only fulfilled for $\mu = 0$. Inserting this condition simplifies the equations above and leads to

$$\mu_{\text{ext}} = -\frac{Kw\sigma}{\sqrt{2\pi}} \quad .$$

For the purpose of relating synaptic weight $w$ and spectral bound $R$, we can view the nonlinear network as an effective linear network with half the population size (only the active neurons). In the latter case, we obtain

$$w = -\frac{R}{\sqrt{\frac{N}{2}p(1-p)}} \quad .$$

For a given spectral bound $R$, this relation allows us to derive the value

$$\mu_{\text{ext}} = \sqrt{\frac{Np}{\pi(1-p) - (\pi-1)R^2}}\sqrt{\sigma_{\text{ext}}^2}R \quad , \tag{7}$$

that, for a arbitrarily fixed $\sigma_{\text{ext}}$ (here $\sigma_{\text{ext}} = 1$), makes half of the population being active. We were aiming for an effective connectivity with only weak fluctuations in the stationary state. Therefore, we fixed the noise strength for all neurons to the small value $\sigma_{\text{noise}} = 0.1 \ll \sigma_{\text{ext}}$ compared to the external input, such that the noise fluctuations did not have a large influence on the calculation above that determines which neurons were active.

To show the effect of a change in the effective connectivity on the covariances, we simulated two networks with identical connectivity, but supplied them with slightly different external inputs. This was realized by choosing

$$\mu_{\text{ext},i}^{(\alpha)} = \mu_{\text{ext},i} + \mu_{\text{ext},i}^{(\alpha)} \quad ,$$

with

$$\mu_{\text{ext},i} \sim \mathcal{N}\left(\mu_{\text{ext}}, [1-\epsilon]\sigma_{\text{ext}}^2\right) , \quad \mu_{\text{ext},i}^{(\alpha)} \sim \mathcal{N}\left(0, \epsilon\sigma_{\text{ext}}^2\right) ,$$

$\epsilon \ll 1$, and $\alpha \in \{1, 2\}$ indexing the two networks. The main component $\mu_{\text{ext},i}$ of the external input was the same for both networks. But, the small component $\mu_{\text{ext},i}^{(\alpha)}$ was drawn independently for the two networks. This choice ensures that the two networks have a similar external input distribution (*Figure 5B1*), but with the external inputs distributed differently across the single neurons (*Figure 5B2*). How similar the external inputs are distributed across the single neurons is determined by $\epsilon$.

The two networks have a very similar firing rate distribution (*Figure 5E1*), but, akin to the external inputs, the way the firing rates are distributed across the single neurons differs between the two networks (*Figure 5E2*). As the effective connectivity depends on the firing rates

$$W_{ij} = J_{ij}\phi'(\nu_j) \quad ,$$

this leads to a difference in the effective connectivities of the two networks and therefore to different covariance patterns, as discussed in *Figure 5*.

We performed the simulation for spectral bounds ranging from 0.1 to 0.9 in increments of 0.1. We calculated the correlation coefficient of firing rates and the correlation coefficient of time-lag integrated covariances between $N_{\text{sample}}$ neurons in the two networks (*Figure 5D*) and studied the dependence on the spectral bound.

To check whether the simulation was long enough to yield a reliable estimate of the rates and covariances, we split each simulation into two halves, and calculate the correlation coefficient between the rates and covariances from the first half of the simulation with the rates and covariances from the second half. They were almost perfectly correlated (*Figure 5C*). Then, we calculated the correlation coefficients comparing all halves of the first simulation with all halves of the second simulation, showing that the covariance patterns changed much more than the rate patterns (*Figure 5C*).

## Acknowledgements

This work was partially supported by HGF young investigator's group VH-NG-1028, European Union's Horizon 2020 research and innovation program under Grant agreements No. 785,907 (Human Brain Project SGA2) and No. 945,539 (Human Brain Project SGA3), ANR grant GRASP and partially funded by the Deutsche Forschungsgemeinschaft (DFG, German Research Foundation) - 368482240/GRK2416. We are grateful to our colleagues in the NEST and Elephant developer communities and for continuous collaboration. All network simulations were carried out with NEST 2.20.0 (http://www.nest-simulator.org). All data analyses performed with Elephant (https://neuralensemble.org/elephant/). We thank Sebastian Lehmann for help with the design of the figures.

---

## Additional information

### Funding

| Funder | Grant reference number | Author |
|---|---|---|
| Helmholtz Association | VH-NG-1028 | David Dahmen<br>Moritz Helias |
| European Commission | HBP (785907 945539) | David Dahmen<br>Moritz Layer<br>Lukas Deutz<br>Nicole Voges<br>Michael von Papen<br>Markus Diesmann<br>Sonja Grün<br>Moritz Helias |
| Deutsche Forschungsgemeinschaft | 368482240/GRK2416 | Moritz Layer<br>Markus Diesmann<br>Moritz Helias |
| Agence Nationale de la Recherche | GRASP | Thomas Brochier<br>Alexa Riehle |

The funders had no role in study design, data collection and interpretation, or the decision to submit the work for publication.

### Author contributions

David Dahmen, Conceptualization, Formal analysis, Investigation, Methodology, Software, Supervision, Validation, Visualization, Writing – original draft, Writing – review and editing; Moritz Layer, Data curation, Formal analysis, Investigation, Methodology, Software, Validation, Visualization, Writing – original draft, Writing – review and editing, Conceptualization; Lukas Deutz, Formal analysis, Methodology, Validation, Visualization, Writing – original draft; Paulina Anna Dąbrowska, Data curation, Formal analysis, Investigation, Methodology, Software, Visualization, Writing – original draft, Writing – review and editing; Nicole Voges, Data curation, Formal analysis, Investigation, Methodology, Software, Validation, Visualization, Writing – original draft, Writing – review and editing; Michael von Papen, Formal analysis, Investigation, Methodology, Software, Validation, Visualization, Writing – original draft; Thomas Brochier, Alexa Riehle, Data curation, Funding acquisition, Resources, Writing – original draft, Writing – review and editing; Markus Diesmann, Sonja Grün, Conceptualization, Funding acquisition, Investigation, Resources, Supervision, Writing – original draft, Writing – review and editing; Moritz Helias, Conceptualization, Formal analysis, Funding acquisition, Investigation,

Methodology, Project administration, Supervision, Validation, Visualization, Writing – original draft, Writing – review and editing

### Author ORCIDs
David Dahmen http://orcid.org/0000-0002-7664-916X
Moritz Layer http://orcid.org/0000-0002-7363-2688
Paulina Anna Dąbrowska http://orcid.org/0000-0002-5555-3206
Nicole Voges http://orcid.org/0000-0002-6324-2600
Michael von Papen http://orcid.org/0000-0001-5030-1643
Thomas Brochier http://orcid.org/0000-0001-6948-1234
Alexa Riehle http://orcid.org/0000-0001-5890-3999
Markus Diesmann http://orcid.org/0000-0002-2308-5727
Sonja Grün http://orcid.org/0000-0003-2829-2220
Moritz Helias http://orcid.org/0000-0002-0404-8656

### Ethics
All animal procedures were approved by the local ethical committee (C2EA 71; authorization A1/10/12) and conformed to the European and French government regulations.

### Decision letter and Author response
Decision letter https://doi.org/10.7554/eLife.68422.sa1
Author response https://doi.org/10.7554/eLife.68422.sa2

## Additional files

### Supplementary files
• Transparent reporting form

### Data availability
All code and data required to reproduce the figures are available in a public zenodo repository at https://zenodo.org/record/5524777. Source data/code files are also attached as zip folders to the individual main figures of this submission.

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

# Appendix 1

## 1 Correlations and covariances

A typical measure for the strength of neuronal coordination is the Pearson correlation coefficient, here applied to spike counts in 1 s bins. Correlation coefficients, however, comprise features of both auto- and cross-covariances. From a theoretical point of view, it is simpler to study cross-covariances separately. Indeed, linear-response theory has been shown to faithfully predict cross-covariances in spiking leaky integrate-and-fire networks (*Tetzlaff et al., 2012*; *Pernice et al., 2012*; *Trousdale et al., 2012*; *Helias et al., 2013*; *Dahmen et al., 2019*; *Grytskyy et al., 2013*). *Appendix 1—figure 1* justifies the investigation of cross-covariances instead of correlation coefficients for the purpose of this study. It shows that the spatial organization of correlations closely matches the spatial organization of cross-covariances.

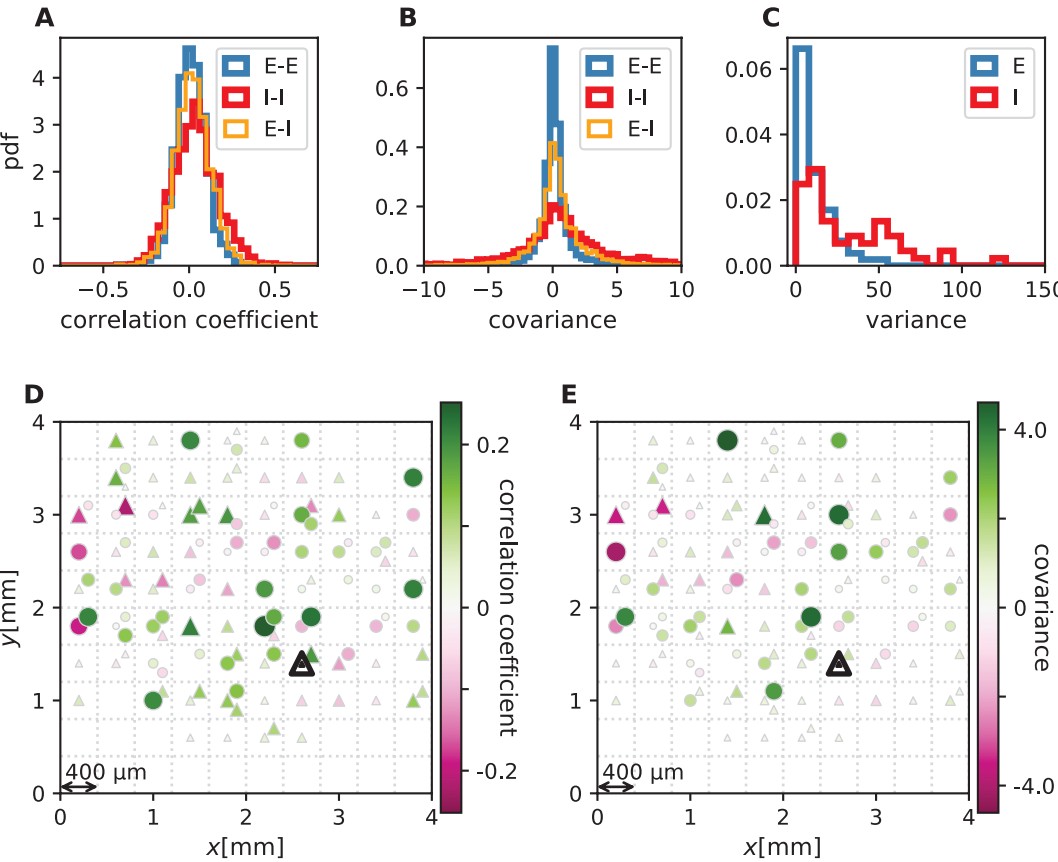

**Appendix 1—figure 1.** Correlations and covariances. The shown data is taken from session E2. (E-E: excitatory-excitatory, E-I: excitatory-inhibitory, I-I: inhibitory-inhibitory). (**A**) Population-resolved distribution of pairwise spike-count Pearson correlation coefficients. Same data as in *Figure 1C*. (**B**) Population-resolved distribution of pairwise spike-count covariances. (**C**) Population-resolved distribution of variances. (**D**) Pairwise spike-count correlation coefficients with respect to the neuron marked by black triangle. Grid indicates electrodes of a Utah array, triangles and circles correspond to putative excitatory and inhibitory neurons, respectively. Size as well as color of markers represent correlation. Neurons within the same square were recorded on the same electrode. Same data as in *Figure 1D*. (**E**) Pairwise spike-count covariances with respect to the neuron marked by black triangle.

## 2 Robustness to E/I separation

The analysis of the experimental data involves a number of preprocessing steps, which may affect the resulting statistics. In our study one such critical step is the separation of putative excitatory and inhibitory units, which is partially based on setting thresholds on the widths of spike waveform,

as described in the Methods section. We tested the robustness of our conclusions with respect to these thresholds.

As mentioned in the Methods, two thresholds for the width of a spike waveform are chosen, based on all SU average waveforms: A width larger than the "broadness" threshold indicates a putative excitatory neuron. A width lower than the "narrowness" threshold indicates a putative inhibitory neuron. Units with intermediate widths are unclassified. Additionally, to increase the reliability of the classification, we perform it in two steps: first on the SU's average waveform, and second on all its single waveforms. We calculate the percentage of single waveforms classified as either type. Finally, only SUs showing a high enough percentage of single waveforms classified the same as the average waveform are sorted as the respective type. The minimal percentage required, referred to as consistency $c$, is initially set to the lowest value which ensures no contradictions between average- and single-waveform thresholding results. While the "broadness" and "narrowness" thresholds are chosen based on all available data for a given monkey, the required consistency is determined separately for each recording session. For monkey N $c$ is set to 0.6 in all but one sessions: In resting state session N1 it is increased to 0.62. For monkey E the values of $c$ equals 0.6 in the resting state recordings and take the following values in five analyzed reach-to-grasp sessions: 0.6, 0.89, 0.65, 0.61, 0.64.

The only step of our analysis for which the separation of putative excitatory and inhibitory neurons is crucial is the fitting of exponentials to the distance-resolved covariances. This step only involves resting state data. To test the robustness of our conclusions, we manipulate the required consistency value for sessions E1, E2, N1, and N2 by setting it to 0.75. *Appendix 1—figure 2* and *Appendix 1—table 1* summarize the resulting fits.

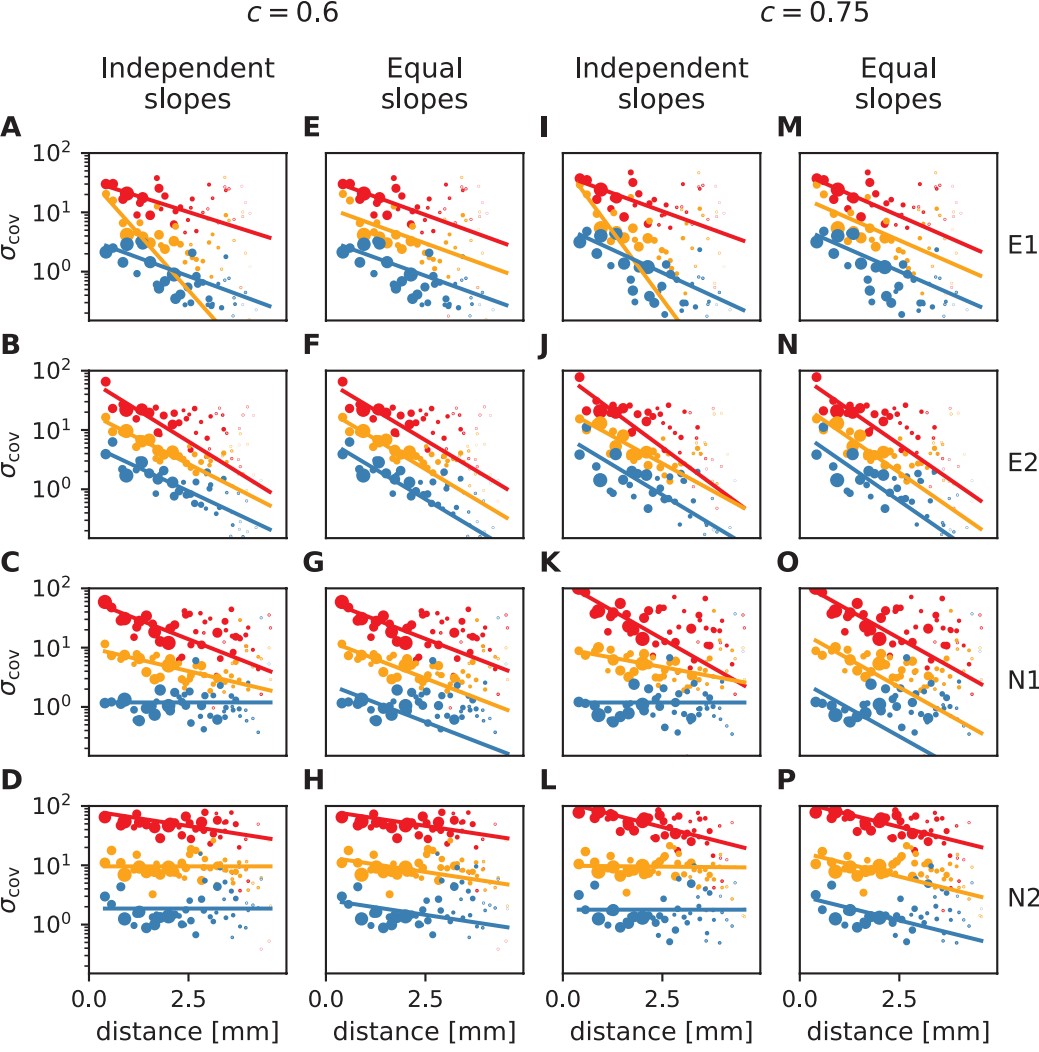

**Appendix 1—figure 2.** Distance-resolved variance of covariance: robustness of decay constant estimation. Exponential fits (lines) to variances of covariances (dots) analogous to *Figure 4A and B* in the main text (columns 1&3 and 2&4, respectively) for all analyzed resting state sessions. The two sets of plots differ in E/I separation consistency values chosen during data preprocessing. Panels **A-H**: default (lowest) required consistency (~0.6), used throughout the main analysis; panels **I-P**: $c = 0.75$ The values of the obtained decay constants are listed in *Appendix 1—table 1*.

It turns out that increasing $c$ to 0.75, which implies disregarding about 20-25 percent of all data, does not have a strong effect on the fitting results. The obtained decay constants are smaller than for a lower $c$ value, but they stay in a range about an order of magnitude larger than the anatomical connectivity. We furthermore see that fitting individual slopes to different populations in some sessions leads to unreliable results (cf. yellow lines in *Appendix 1—figure 2A, I* and blue lines in *Appendix 1—figure 2C,D,K,L*). Therefore, the data is not sufficient to detect differences in decay constants for different neuronal populations. Fitting instead a single decay constant yields trustworthy results (cf. yellow lines in *Appendix 1—figure 2E,M* and blue lines in *Appendix 1—figure 2G,H,O,P*). Our data thus clearly expose that decay constants of covariances are in the millimeter range.

**Appendix 1—table 1.** Summary of exponential fits to distance-resolved variance of covariance. For each value of E/I separation consistency $c$ the numbers of sorted putative neurons and the percentages of unclassified units, and therefore not considered for fitting SUs, are listed per resting state session, along with the resulting fits (*Figure 4* in the main text)

| C | | E1 | E2 | N1 | N2 |
|---|---|---|---|---|---|
| | #exc/#inh | 56/50 | 67/56 | 76/45 | 78/62 |
| | unclassified | 0.078 | 0.075 | 0.069 | 0.091 |
| | relative error | 1.1157 | 1.0055 | 1.0097 | 1.0049 |
| | 1-slope fit | 1.674 | 1.029 | 1.676 | 4.273 |
| | I-I | 1.919 | 0.996 | 1.647 | 4.156 |
| | I-E | 0.537 | 1.206 | 2.738 | 96100.688 |
| 0.6 (default) | E-E | 1.642 | 1.308 | 80308.482 | 94096.871 |
| | #exc/#inh | 45/42 | 47/48 | 70/36 | 74/48 |
| | unclassified | 0.24 | 0.28 | 0.18 | 0.21 |
| | relative error | 1.1778 | 1.0141 | 1.0102 | 1.0090 |
| | 1-slope fit | 1.357 | 0.874 | 1.420 | 2.587 |
| | I-I | 1.794 | 0.809 | 1.394 | 2.550 |
| | I-E | 0.496 | 1.123 | 3.682 | 40.852 |
| 0.75 | E-E | 1.390 | 1.199 | 80548.500 | 10310.780 |

## 3 Stationarity of behavioral data

The linear-response theory, with the aid of which we develop our predictions about the covariance structure in the network, assumes that the processes under examination are stationary in time. However, this assumption is not necessarily met in experimental data, especially in motor cortex during active behavioral tasks. For this reason we analyzed the stationarity of average single unit firing rate and pairwise zero time-lag covariance throughout a reach-to-grasp trial, similarly to *Dahmen et al., 2019*. Although the spiking activity becomes highly non-stationary during the movement, those epochs that are chosen for the analysis in our study (S and P) show only moderate variability in time (*Appendix 1—figure 3*). An analysis on the level of single-unit resolved activity also shows that the majority of neurons has stationary activity statistics within the relevant epochs S and P, especially when comparing to their whole dynamic range that is explored during movement transients towards the end of the task (*Appendix 1—figure 5*). *Appendix 1—figure 6* shows that there are, however, a few exceptions (e.g. units 11, 84 in this session) that show moderate transients also within an epoch. Nevertheless, these transients are small compared to changes between the two epochs S and P.

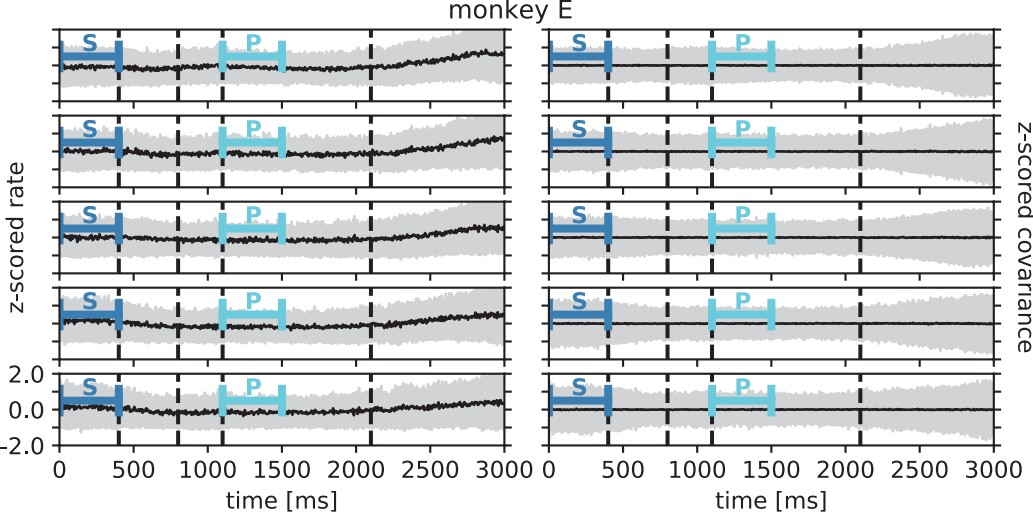

**Appendix 1—figure 3.** Rate and covariance stationarity during a reach-to-grasp trial: monkey E. Black line indicates population mean and gray area +/- 1 population standard deviation of single unit firing rate (left column) and pairwise zero time-lag covariance (right column) during trial of a given session (row). Blue bars indicate starting (S) and preparatory (P) periods used in the analysis (**Figure 6** in the main text). First, second and fourth dashed lines indicate visual signals lighting up and the third dashed line indicates the removal of a visual cue and beginning of a waiting period.

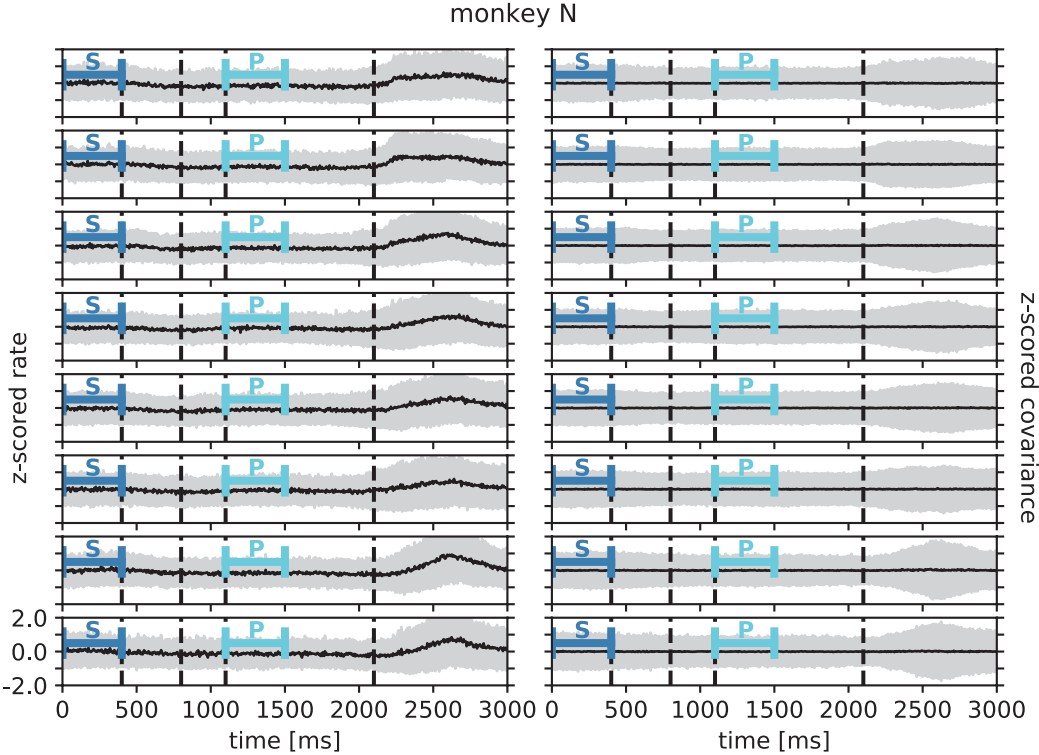

**Appendix 1—figure 4.** Rate and covariance stationarity during a reach-to-grasp trial: monkey N. Analogous to **Appendix 1—figure 3**.

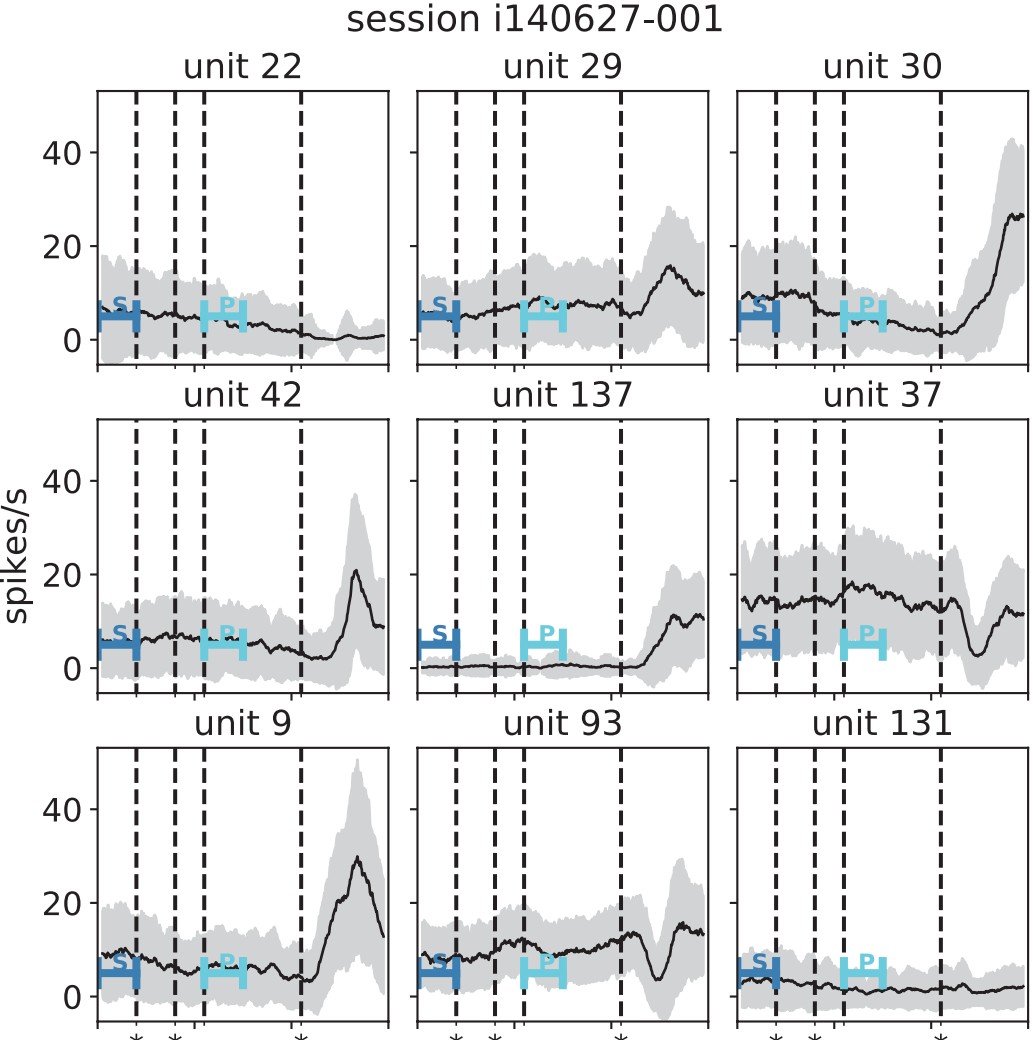

**Appendix 1—figure 5.** Stationarity of single-unit activity during a reach-to-grasp trial (monkey N, session i140627-001). Black lines indicate mean and gray areas +/- 1 standard deviation across trials of single unit activity in each panel (sliding window analysis with 5 ms step size and 100 ms window length). Blue bars indicate starting (S) and preparatory (P) periods used in the analysis (*Figure 6* in the main text). First, second and fourth dashed lines (marked with stars) indicate visual signals lighting up and the third dashed line indicates the removal of a visual cue and beginning of a waiting period.

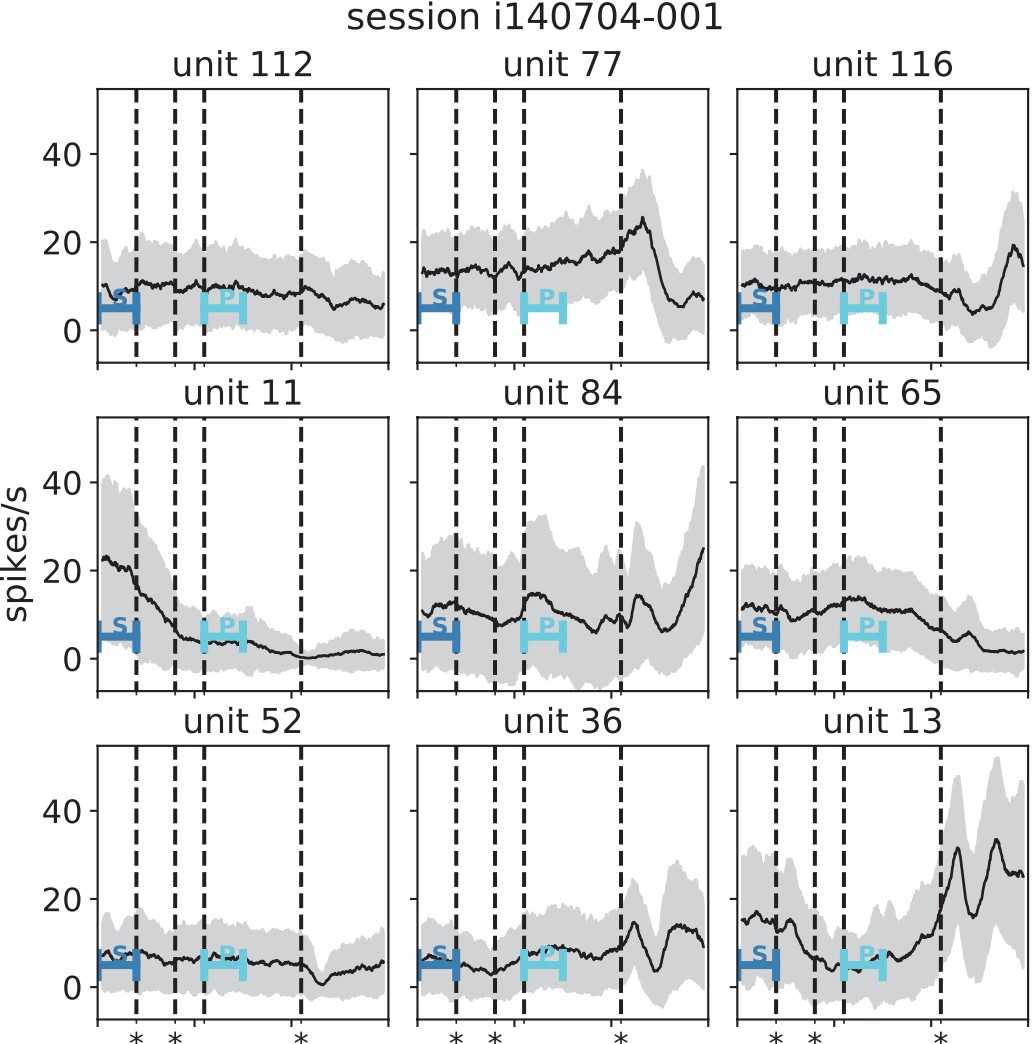

**Appendix 1—figure 6.** Stationarity of single-unit activity during a reach-to-grasp trial (monkey N, session i140704-001). Analogous to **Appendix 1—figure 5**.

Thus both the population-level and single-unit level analyses are in line with the second test for stationarity that we show in **Figure 6**. There we compare the firing rate and covariance changes between two 200 ms segments of the same epoch to the firing rate and covariance changes between two 200 ms segments of different epochs. If the neural activity was not stationary within an epoch then we would not obtain correlation coefficients of almost one between firing rates in **Figure 6E** and correlation coefficients up to 0.9 between covariance patterns within one epoch in **Figure 6F**. In summary, the analyses together make us confident that assuming stationarity within an epoch is a good approximation to show that there are significant behaviorally related changes in covariances across epochs of the reach-to-grasp experiment.

**Appendix 1—table 2.** Numbers of trials and single units per reach-to-grasp recording session. Session names starting with "e" correspond to monkey E and session names starting with "i" to monkey N.

| Session | $N_{trials}$ | $N_{single\ units}$ |
| --- | --- | --- |
| e161212-002 | 108 | 129 |
| e61214-001 | 99 | 118 |

*Appendix 1—table 2 Continued on next page*

Appendix 1—table 2 Continued

| Session | N$_{trials}$ | N$_{single units}$ |
|---|---|---|
| e161222-002 | 102 | 118 |
| e170105-002 | 101 | 116 |
| e170106-001 | 100 | 113 |
| i140613-001 | 93 | 137 |
| i140617-001 | 129 | 155 |
| i140627-001 | 138 | 145 |
| i140702-001 | 157 | 134 |
| i140703-001 | 142 | 142 |
| i140704-001 | 141 | 124 |
| i140721-002 | 160 | 96 |
| i140725-002 | 151 | 106 |

## 4 Network model

We are considering neuronal network models with isotropic and distance-dependent connection profiles. Ultimately, we are interested in describing cortical networks with two-dimensional sheet-like structure. But, for developing the theory, we first consider the simpler case of a one-dimensional ring and subsequently develop the theory on a two-dimensional torus, ensuring periodic boundary conditions in both cases. $N$ equidistantly distributed neurons form a grid on these manifolds. The position of neuron $i \in \{1, ..., N\}$ is described by the vector $\boldsymbol{r}_i \in \mathbb{R}^D$, with $D \in \{1, 2\}$. The connections $W_{ij}$ from neuron $j$ to neuron $i$ are drawn randomly with a connection probability that decays with distance between neurons $|\boldsymbol{r}_i - \boldsymbol{r}_j|$, described by the normalized connectivity profile $p(\boldsymbol{r})$, $\int p(\boldsymbol{r}) \, \mathrm{d}^D r = 1$, which we assume to obey radial symmetry. The connection probability decays on a characteristic length scale $d$. As we are working on discrete lattices, we introduce the probability of two neurons being connected $p_{ij}$, which is defined by the relation $p(\boldsymbol{r}_i - \boldsymbol{r}_j) = \lim_{a \to 0} p_{ij}/a$, with lattice spacing $a$. We set the synaptic weights for connections of a single type to a fixed value $w$, but allow for multiple connections between neurons, that is $W_{ij} \in \{0, w, 2w, ...\} = n_{ij} \cdot w$ for all sending neurons $j$ of a given type, where $n_{ij}$ is binomially distributed. Such multapses are required to simultaneously meet biological constraints on neuronal indegrees, neuron densities, and spatial ranges of connections. If instead one assumed Bernoulli connectivity, an analysis analogous to Eq. 7 of *Senk et al., 2018* would yield a connection probability exceeding unity.

We introduce two populations of neurons, excitatory (E) and inhibitory (I) neurons. The number of neurons of a given population $a \in \{E, I\}$ is $N_a$, and their ratio is $q = N_E/N_I$, which, for convenience, we assume to be an even number (see permutation symmetry below). The connection from population $b$ to population $a$ has the synaptic weight $w_{ab}$ and characteristic decay length of the connectivity profile $d_{ab}$. The average number of inputs drawn per neuron is fixed to $K_{ab}$. In order to preserve translation symmetry, $q$ excitatory neurons and one inhibitory neuron are put onto the same lattice point, as shown in *Figure 3A* in the main text.

Linear-response theory has been shown to faithfully capture the statistics of fluctuations in asynchronous irregular network states (*Lindner et al., 2005*). Here we follow *Grytskyy et al., 2013*, who show that different types of neuronal network models can be mapped to an Ornstein-Uhlenbeck process and that the low-frequency limit of this simple rate model describes spike count covariances of spiking models well (*Tetzlaff et al., 2012*). In particular, *Dahmen et al., 2019* showed quantitative agreement of linear-response predictions for the statistics of spike-count covariances in leaky integrate-and-fire networks for the full range of spectral bounds $R \in [0, 1)$. Therefore, we consider a network of linear rate neurons, whose activity $\boldsymbol{x} \in \mathbb{R}^N$ is described by

$$\tau \frac{\mathrm{d}}{\mathrm{d}t} \boldsymbol{x} = -\boldsymbol{x} + \boldsymbol{W}\boldsymbol{x} + \boldsymbol{\xi} \quad ,$$

with uncorrelated Gaussian white noise $\boldsymbol{\xi}$, $\langle \xi_i(t) \rangle = 0$, $\langle \xi_i(s)\xi_j(t) \rangle = D_i \delta_{ij} \delta(s - t)$. The solution to this differential equation can be found by multiplying the whole equation with the left eigenvectors $\boldsymbol{u}_\alpha$ of $\boldsymbol{W}$

$$\tau \frac{\mathrm{d}}{\mathrm{d}t} y_\alpha = -y_\alpha + \lambda_\alpha y_\alpha + \xi_\alpha \quad , \tag{8}$$

where $y_\alpha = \boldsymbol{u}_\alpha \cdot \boldsymbol{x}$, $\xi_\alpha = \boldsymbol{u}_\alpha \cdot \boldsymbol{\xi}$, and $\lambda_\alpha$ is denoting the corresponding eigenvalue of $\boldsymbol{W}$. Neglecting the noise term, the solutions are given by

$$y_\alpha(t) \propto \Theta(t) \exp[-\frac{t}{\tau}(1 - \lambda_\alpha)] \quad , \tag{9}$$

with Heaviside function $\Theta(t)$. These are the eigenmodes of the linear system and they are linear combinations of the individual neuronal rates

$$y_\alpha = \sum_{i=1}^N (u_\alpha)_i x_i \quad .$$

Note that the weights $(u_\alpha)_i$ of these linear combinations depend on the details of the effective connectivity matrix $\boldsymbol{W}$. The stability of an eigenmode is determined by the corresponding eigenvalue $\lambda_\alpha$. If $\mathrm{Re}\,(\lambda_\alpha) < 1$, the eigenmode is stable and decays exponentially. If $\mathrm{Re}\,(\lambda_\alpha) > 1$, the eigenmode is unstable and grows exponentially. If $\mathrm{Im}\,(\lambda_\alpha) \neq 0$, the eigenmode is oscillatory with an exponential envelope. $\mathrm{Re}\,(\lambda_\alpha) = 1$ is here referred to as the critical point. This type of stability is also called linear stability to stress that these considerations are only valid in the linear approximation. Realistic neurons have a saturation at high rates, which prevents activity from diverging indefinitely. A network is called linearly stable if all modes are stable. This is determined by the real part of the largest eigenvalue of $\boldsymbol{W}$, called spectral bound $R$. In inhibition-dominated networks, the spectral bound is determined by the heterogeneity in connections and $R \lessgtr 1$ defines the dynamically balanced critical state (*Dahmen et al., 2019*).

The different noise components $\xi_\alpha$ excite the corresponding eigenmodes of the system and act as a driving force. A noise vector $\boldsymbol{\xi}$ that is not parallel to a single eigenvector $\boldsymbol{u}_\alpha$ excites several eigenmodes, each with the corresponding strength $\xi_\alpha$.

Note that the different eigenmodes do not interact, which is why the total activity $\boldsymbol{x}$ is given by a linear combination, or superposition, of the eigenmodes

$$\boldsymbol{x} = \sum_{\alpha=1}^N y_\alpha \boldsymbol{v}_\alpha \quad ,$$

where $\boldsymbol{v}_\alpha$ denotes the α-th right eigenvector of the connectivity matrix $\boldsymbol{W}$.

## 5 Covariances

Time-lag integrated covariances $c_{ij} = \int d\tau \, \langle x_i(t)x_j(t + \tau) \rangle - \langle x_i(t) \rangle \langle x_j(t + \tau) \rangle$ can be computed analytically for the linear dynamics (*Gallego et al., 2020*). They follow from the connectivity $\boldsymbol{W}$ and the noise strength $D$ as (*Pernice et al., 2011*; *Trousdale et al., 2012*; *Grytskyy et al., 2013*; *Lindner et al., 2005*)

$$\boldsymbol{c} = [\mathbf{1} - \boldsymbol{W}]^{-1} \boldsymbol{D} [\mathbf{1} - \boldsymbol{W}]^{-\mathrm{T}} \quad , \tag{10}$$

with identity matrix $\mathbf{1}$. These covariances are equivalent to covariances of spike counts in large time windows, given by the zero-frequency component of the Fourier transform of $\boldsymbol{x}$ (sometimes referred to as Wiener-Khinchin theorem *Gardiner, 1985*; even though the theorem proper applies in cases where the Fourier transforms of the signals $\boldsymbol{x}$ do not exist). Spike count covariances (*Figure 1B* in the main text) can be computed from trial-resolved spiking data (*Dahmen et al., 2019*). This equivalence allow us to directly relate theoretical predictions for covariances to the experimentally observed ones.

While *Equation (10)* provides the full information on covariances between any two neurons in the network, this information is not available in the experimental data. Only a small subset of neuronal activities can be recorded such that inference of connectivity parameters from *Equation 10* is unfeasible. We recently proposed in *Dahmen et al., 2019* to instead consider the statistics of covariances as the basis for comparison between models and data. Using *Equation 8* and *Equation 10* as a starting point, field theoretical techniques allow the derivation of equations for the mean $\bar{c}$ and variance $\overline{\delta c^2}$ of cross-covariances in relation to the mean $\boldsymbol{M}$ and variance $\boldsymbol{S}$ of the connectivity matrix $\boldsymbol{W}$ (*Dahmen et al., 2019*):

$$\bar{c} = [\mathbf{1} - \mathbf{M}]^{-1} \mathbf{D}_{\mathrm{r}} [\mathbf{1} - \mathbf{M}]^{-\mathrm{T}} \quad, \tag{11}$$

$$\overline{\delta c^2} = [\mathbf{1} - \mathbf{S}]^{-1} \mathbf{D}_{\mathrm{r}}^2 [\mathbf{1} - \mathbf{S}]^{-\mathrm{T}} \quad. \tag{12}$$

$\mathbf{M}$ and $\mathbf{S}$ are defined in the subsequent section. The renormalized input noise strength is given by

$$\mathbf{D}_{\mathrm{r}} = \mathbf{D} + \mathrm{diag}\left[\mathbf{D}\left(\mathbf{1} - \mathbf{S}\right)^{-1}\mathbf{S}\cdot\mathbf{I}\right] \quad, \tag{13}$$

with input noise covariance $\mathbf{D}$, and the all-ones vector $\mathbf{I} = (1, \dots, 1)^{\mathrm{T}} \in \mathbb{R}^N$. Note that *Equation (12)* only holds for cross-covariances ($i \neq j$). The diagonal terms $\left[\overline{\delta c^2}\right]_{ii}$, that is the variance of auto-covariances, do get a second contribution, which is negligible for the cross-covariances considered here.

## 6 Cumulant generating function of connectivity matrix

For calculating the mean and variance of the covariances of the network activity ((11) and (12)) we need mean $\mathbf{M}$ and variance $\mathbf{S}$ of connectivity $\mathbf{W}$. In the following, we derive the cumulant generating function (*Gardiner, 1985*) of $W_{ij}$.

The number of connections $n$ from neuron $j$ to neuron $i$ is a binomial random variable with $K$ trials with the probability of success given by $p_{ij}$ (in the following, for brevity, we ignore the index $i$, $p_{ij} \equiv p_j$)

$$\mathrm{Prob}_j(n) = \binom{K}{n} p_j^n \left(1 - p_j\right)^{K-n} \quad.$$

The average number of connections from neuron $j$ to neuron $i$ is $K_j = p_j K$, which assures the correct average total indegree

$$\sum_j K_j = K \sum_j p_j = K \quad.$$

The moment generating function of a connectivity matrix element $W_j \equiv W_{ij} \in \{0, w, 2w, \dots\}$ is given by

$$Z_{W_j}(k) = \sum_{n=0}^{K} \binom{K}{n} p_j^n \left(1 - p_j\right)^{K-n} e^{nwk} \quad.$$

In a realistic network, $K$ is very large. In the limit $K \to \infty$, while keeping $Kp = \mathrm{const.}$, the binomial distribution converges to a Poisson distribution and we can write

$$\begin{aligned}
Z_{W_j}(k) &\approx \sum_{n=0}^{K} \frac{K_j^n}{n!} e^{-K_j} e^{nwk} \\
&= \sum_{n=0}^{K} \frac{\left(K_j e^{wk}\right)^n}{n!} e^{-K_j} \\
&\xrightarrow{K \to \infty} \exp\left[K_j \left(e^{wk} - 1\right)\right] \quad.
\end{aligned}$$

Taking the logarithm leads to the cumulant generating function

$$G_{W_j}(k) \approx p_j K \left(e^{wk} - 1\right) \quad,$$

and the first two cumulants

$$\begin{aligned}
M_{ij} &= \left.\frac{\partial}{\partial k} G_{W_j}(k)\right|_{k=0} = p_j K w = p\left(\left|x_i - x_j\right|\right) K w \quad, \\
S_{ij} &= \left.\frac{\partial^2}{\partial k^2} G_{W_j}(k)\right|_{k=0} = p_j K w^2 = p\left(\left|x_i - x_j\right|\right) K w^2 \quad.
\end{aligned}$$

## 7 Note on derivation of variance of covariances

Note that $\mathbf{M}$ and $\mathbf{S}$ have an identical structure determined by the connectivity profile and the structure of the covariance equation is identical for the mean *Equation (11)* and variance *Equation (12)* as well. This is why in the following we only derive the results for the mean of covariances.

The results for the variance of covariances is obtained by substituting $w$ by $w^2$ and $D_r$ by $D_r^2$. As we show, divergences in expressions related to the mean covariances arise if the population eigenvalue $\lambda_0$ of the effective connectivity matrix approaches one. In expressions related to the variance of covariances, the divergences are caused by the squared spectral bound $R^2$ being close to one. In general expressions, we sometimes write $\zeta$ in order to denote either the population eigenvalue or the spectral bound, corresponding to the context of mean or variance of covariances.

## 8 Utilizing symmetries to reduce dimensionality

For real neuronal networks, the anatomical connectivity is never known completely, let alone the effective connectivity. This is why we are considering disorder-averaged systems. They are described by the mean $M$ and variance $S$ of the connectivity. The latter inherit the underlying symmetries of the network, like for example the same radially symmetric connectivity profile for all neurons of one type. As neuronal networks are high dimensional systems, calculating covariances from *Equation (11)* and *Equation (12)* first seems like a daunting task. But, leveraging the aforementioned symmetries similarly as in *Kriener et al., 2013* allows for an effective reduction of the dimensionality of the system, thereby rendering the problem manageable.

As a demonstrative example of how this is done, consider a random network of $N$ neurons on a one-dimensional ring, in which a neuron can form a connection with weight $w$ to any other neuron with probability $p_0$. In that case, $M$ is a homogeneous matrix, with all entries given by the same average connectivity weight

$$
M = \begin{pmatrix} p_0 w & p_0 w & \dots & p_0 w \\ p_0 w & p_0 w & \dots & p_0 w \\ \vdots & \vdots & \ddots & \vdots \\ p_0 w & p_0 w & \dots & p_0 w \end{pmatrix} \quad .
$$

This corresponds to an all-to-all connected ring network. Due to the symmetry of the system, moving all neurons by one lattice constant does not change the system. The translation operator $T$, representing this operation mathematically, is defined via its effect on the vector of neuron activity $x$

$$
Tx = T \begin{pmatrix} x_1 \\ x_2 \\ \vdots \\ x_N \end{pmatrix} = T \begin{pmatrix} x_N \\ x_1 \\ \vdots \\ x_{N-1} \end{pmatrix} \quad .
$$

Applying $T$ $N$-times yields the identity operation

$$
T^N = \mathbf{1} \quad .
$$

Hence, its eigenvalues are given by complex roots of one

$$
e^{-i2\pi l/N} = e^{-i2\pi la/L} = e^{-ik_l a} \quad , \quad l \in \{0, 1, ..., N-1\} \quad ,
$$

with $L = Na$ denoting the circumference of the ring. This shows that $T$ has $N$ one-dimensional eigenspaces. Since the system is invariant under translation, $M$ is invariant under the transformation $TMT^{-1} = M$, and thus $M$ and $T$ commute. As $M$ leaves eigenspaces of $T$ invariant (if $v$ is an eigenvector of $T$, $Mv$ is an eigenvector with the same eigenvalue, so they need to be multiples of each other), all eigenvectors of $T$ must be eigenvectors of $M$. Accordingly, knowing the eigenvectors of $T$ allows diagonalizing $M$. The normalized (left and right) eigenvectors of $T$ are given by

$$v_{k_l} = \frac{1}{\sqrt{N}} \begin{pmatrix} 1 \\ e^{ik_l a} \\ e^{i2k_l a} \\ \vdots \\ e^{i(N-1)k_l a} \end{pmatrix} .$$

We get the eigenvalues of $M$ by multiplying it with the eigenvectors of $T$

$$\begin{aligned} m_{k_l} &= v_{k_l}^\dagger M v_{k_l} \\ &= \frac{1}{N} p_0 w v_{k_l}^\dagger \cdot \begin{pmatrix} \sum_{j=0}^{N-1} e^{ik_l aj} \\ \sum_{j=0}^{N-1} e^{ik_l aj} \\ \vdots \\ \sum_{j=0}^{N-1} e^{ik_l aj} \end{pmatrix} , \end{aligned}$$

which is always zero, except for $l = 0$, which corresponds to the population eigenvalue $\lambda_0 := m_{k_0} = N p_0 w$ of $W$ (**Figure 3C** in the main text). Now, we can simply write down the diagonalized form of $M$

$$\begin{pmatrix} \lambda_0 & 0 & \dots & 0 \\ 0 & 0 & \dots & 0 \\ \vdots & \vdots & \ddots & \vdots \\ 0 & 0 & \dots & 0 \end{pmatrix} ,$$

and we effectively reduced the $N$-dimensional to a one dimensional problem. Inverting $A := 1 - M$ in **Equation (11)** is straightforward now, since it is diagonal in the new basis. Its eigenvalues can be written as $a_k = 1 - m_k$, where we suppressed the index $l$. Therefore its inverse is given by

$$\begin{aligned} A_{ij}^{-1} &= \sum_k a_k^{-1} \left( v_k \right)_i \left( v_k^\dagger \right)_j \\ &= \frac{1}{N} \sum_k \frac{1}{1-m_k} e^{ik(x_i - x_j)} \\ &= \frac{1}{N} \sum_k \left( 1 + \frac{m_k}{1-m_k} \right) e^{ik(x_i - x_j)} \\ &= \delta_{ij} + \frac{1}{N} \frac{\lambda_0}{1-\lambda_0} . \end{aligned}$$

The renormalized noise can be evaluated using that the all-ones vector occurring in equation **Equation (13)** is the eigenvector $v_0$ of $S$. After identifying the eigenvalue $s_0$ with the squared spectral bound $R^2$, we find

$$D_r = \mathrm{diag} \left( \frac{D}{1-R^2} \right) ,$$

which allows us to express the mean cross-covariances $\bar{c}$ (see **Equation (11)**) and the variance of cross-covariances $\overline{\delta c^2}$ (see **Equation (12)**) in terms of the eigenvectors of $M$ and $S$ respectively

$$\bar{c} = \frac{D}{1-R^2} \left\{ \frac{2\lambda_0}{N(1-\lambda_0)} + \left[ \frac{\lambda_0}{N(1-\lambda_0)} \right]^2 N \right\} ,$$

$$\overline{\delta c^2} = \left( \frac{D}{1-R^2} \right)^2 \left\{ \frac{2R^2}{N(1-R^2)} + \left[ \frac{R^2}{N(1-R^2)} \right]^2 N \right\} .$$

## 9 One-dimensional network with one population

The simplest network with spatial connectivity is a one-dimensional ring of neurons with one population of neurons. Following section Section 6, the mean connectivity matrix has the form

$$M = Kw \begin{pmatrix} p_{11} & p_{12} & \cdots & p_{1N} \\ p_{21} & p_{22} & \cdots & p_{2N} \\ \vdots & \vdots & \ddots & \vdots \\ p_{N1} & p_{N2} & \cdots & p_{NN} \end{pmatrix} .$$

As $p_{ij}$ only depends on the distance of two neurons, the rows in $M$ are identical, but shifted by one index.

## 9.1 Dimensionality reduction

We follow the procedure developed in Section 8, as the system is invariant under translation as well. Suppressing the subscripts of $k$, we get the eigenvalues of $M$

$$\begin{aligned} m_k &= \tfrac{1}{N} Kw \left(1, e^{-ika}, ..., e^{-i(N-1)ka}\right) \begin{pmatrix} \sum_{j=0}^{N-1} p_{1(j+1)} e^{ikaj} \\ \sum_{j=0}^{N-1} p_{2(j+1)} e^{ikaj} \\ \vdots \\ \sum_{j=0}^{N-1} p_{N(j+1)} e^{ikaj} \end{pmatrix} \\ &= Kw \sum_{j=0}^{N-1} p_{(j+1)} e^{ikaj} \\ &= Kwa \sum_x p(x) e^{-ikx} \quad, \end{aligned}$$

where the sum over $x$ denotes a sum over all lattice sites. We used the translational symmetry from the first to the second line. The change of sign in the exponential from line two to three is due to the fact that we are summing over the second index of $p_{ij}$. Thus, the eigenvalues are effectively given by the discrete Fourier transform of the connectivity profile. Expressing $A^{-1}$ using the eigenvectors $v_k$ of $M$ leads to

$$\begin{aligned} A_{ij}^{-1} &= \tfrac{1}{N} \sum_k \tfrac{1}{1-m_k} e^{ik(x_i - x_j)} \\ &= \tfrac{1}{N} \sum_k \left(1 + \tfrac{m_k}{1-m_k}\right) e^{ik(x_i - x_j)} \\ &= \delta_{ij} + \tfrac{1}{N} \sum_k \tfrac{m_k}{1-m_k} e^{ik(x_i - x_j)} \\ &\equiv \delta_{ij} + \mu_{ij} \quad, \end{aligned} \tag{14}$$

where we extracted an identity for later convenience, and we defined $\mu_{ij}$.

Next, we consider the renormalized noise, which is given by **Equation (13)**. Using that the all-ones vector $I$ in the second term is the eigenvector of $S$ corresponding to $k = 0$, we get

$$D \left(1 - S\right)^{-1} S \cdot v_0 = D \frac{s_0}{1 - s_0} \quad.$$

Again, we identify $s_0$ with the spectral bound $R^2$, and find

$$D_r = D + D \frac{R^2}{1 - R^2} = \frac{D}{1 - R^2} \quad. \tag{15}$$

Inserting **Equation (14)** and **Equation 15** into **Equation (11)** yields

$$\bar{c}_{ij} = \frac{D}{1 - R^2} \left( \delta_{ij} + 2\mu_{ij} + \sum_k \mu_{ik} \mu_{kj} \right) \quad.$$

## 9.2 Continuum limit

As we assume the lattice constant to be small, we know that the connectivity profile is sampled densely, and we are allowed to take the continuum limit. Therefore, we write

$$
\begin{aligned}
m_k &= Kw \sum_j p_{(j+1)} e^{ikaj} \\
&= Kw \sum_j a \frac{p_{(j+1)}}{a} e^{ikaj} \\
&\xrightarrow{a\to 0} Kw \int_{-L/2}^{L/2} dx\, p(-x) e^{ikx} \\
&= Kw \int_{-L/2}^{L/2} dx\, p(x) e^{-ikx} \quad .
\end{aligned}
$$

Note that $\lim_{a\to 0} \sum_j p_j/a = \lim_{a\to 0} \sum_j p(x_i - x_j)/a = \int dx\, p(-x)$, because we are summing over the second index $j$. If the decay constant $d$ of the connectivity profile is small compared to the size of the network $L$, we can take $L$ to infinity and finally end up with

$$
m(k) = Kw \int dx\, p(x) e^{-ikx} \quad . \tag{16}
$$

Analogously, we find

$$
A^{-1}(x) = \delta(x) + \frac{1}{2\pi} \int dk\, \frac{m(k)}{1 - m(k)} e^{ikx} \equiv \delta(x) + \mu(x) \quad , \tag{17}
$$

where we defined

$$
\mu(x) = \frac{1}{2\pi} \int dk\, \mu(k) e^{ikx} \quad , \tag{18}
$$

with

$$
\mu(k) = \frac{m(k)}{1 - m(k)} \quad . \tag{19}
$$

Finally, we get

$$
\bar{c}(x) = \frac{D}{1 - R^2} \left[ \delta(x) + 2\mu(x) + \left( \mu * \mu \right)(x) \right] \quad , \tag{20}
$$

where the asterisk denotes the convolution.

## 9.3 Prediction of exponential decay of covariance statistics

Note that the integral in equation *Equation 18* can be interpreted as an integral in the complex plane. According to the residue theorem, the solution to this integral is a weighted sum of exponentials, evaluated at the poles of $[1 - m(k)]^{-1}$. As $\mu(x)$ appears in the equation for the mean covariances, and the convolution of two exponentials is an exponential with the prefactor (const. $+ |x|$), we expect the dominant behavior to be an exponential decay in the long-range limit, with decay constants given by the inverse imaginary part of the poles. The poles which are closest to zero are the ones which lead to the most shallow and thereby dominant decay. A real part of the poles leads to oscillations in $\mu(x)$.

## 9.4 Long-range limit

We cannot expect to solve the integral in *Equation 17* for arbitrary connectivity profiles. To continue our analysis, we make use of the Padé method, which approximates arbitrary functions as rational functions (*Basdevant, 1972*). We approximate $\mu(k)$ around $k = 0$ using a Padé approximation of order (0,2)

$$
\mu(k) \approx \frac{m(0)}{1 - m(0) - \frac{m''(0)}{2m(0)} k^2} \quad ,
$$

with

$$
\begin{aligned}
m(0) &= Kw \int dx\, p(x) = Kw = \lambda_0 \quad , \\
m''(0) &= -Kw \int dx\, x^2 p(x) = -Kw <x^2> \quad .
\end{aligned} \tag{21}
$$

This allows us to calculate the approximate poles of $\mu(k)$

$$k_0 = \pm\sqrt{\frac{2m(0)}{m''(0)}\left[1 - m(0)\right]} \quad . \tag{22}$$

As $2m(0)/m''(0)$ will be negative, due to factor $i^2$ from the second derivative of the Fourier integral, we write

$$k_0 = \pm i\sqrt{-\frac{2m(0)}{m''(0)}\left[1 - m(0)\right]} \quad .$$

Closing the integral contour in *Equation 18* in the upper half plane for $x > 0$, and in the lower half plane for $x < 0$, we get

$$\mu(x) = -\frac{m(0)^2}{m''(0)}\sqrt{-\frac{m''(0)}{2m(0)}\frac{1}{1 - m(0)}}\exp\left(-\frac{|x|}{\sqrt{-\frac{m''(0)}{2m(0)}\frac{1}{1-m(0)}}}\right) \equiv -\frac{m(0)^2}{m''(0)}d_{\text{eff},\mu}\exp\left(-\frac{|x|}{d_{\text{eff},\mu}}\right) \quad ,$$

where we defined the effective decay constant for the mean covariances

$$\bar{d} = \sqrt{-\frac{m''(0)}{2m(0)}\frac{1}{1 - m(0)}} = \sqrt{\frac{\langle x^2 \rangle}{2}\frac{1}{1 - \lambda_0}} \quad ,$$

with $m(0) = \lambda_0$ and $m''(0) = \lambda_0\langle x^2 \rangle$, since $m(k)$ is the Fourier transform of the connectivity profile *Equation (16)*. Note that $\lambda_0 = Kw$ again is the population eigenvalue of the effective connectivity matrix $W$. For evaluating *Equation (11)* and *Equation (12)*, we need to calculate the convolution of $\mu$ with itself

$$\left(\mu * \mu\right)(x) = \int \mathrm{d}y\, \mu(x - y)\mu(y) = \frac{m(0)^4}{m''(0)^2}\bar{d}^2\left(\bar{d} + |x|\right)\exp\left(-\frac{|x|}{\bar{d}}\right) \quad .$$

The final expression for the mean covariances is

$$\bar{c}(x) = \frac{D}{1 - R^2}\left\{\delta(x) + \left[\left(\frac{m(0)^4}{m''(0)^2}\bar{d}^2 - 2\frac{m(0)^2}{m''(0)}\right)\bar{d} + \frac{m(0)^4}{m''(0)^2}\bar{d}^2\,|x|\right]\exp\left(-\frac{|x|}{\bar{d}}\right)\right\} \quad .$$

Equivalently, for the variance of covariances we obtain the final result

$$\overline{\delta c^2}(x) = \left(\frac{D}{1 - R^2}\right)^2\left\{\delta(x) + \left[\left(\frac{s(0)^4}{s''(0)^2}d_{\text{eff}}^2 - 2\frac{s(0)^2}{s''(0)}\right)d_{\text{eff}} + \frac{s(0)^4}{s''(0)^2}d_{\text{eff}}^2\,|x|\right]\exp\left(-\frac{|x|}{d_{\text{eff}}}\right)\right\} \quad ,$$

where

$$s(k) = Kw^2 \int \mathrm{d}x\, p(x)\mathrm{e}^{-ikx} \quad .$$

Note that the quality of the Padé approximation depends on the outlier eigenvalue and the spectral bound. For the variances, the approximation works best for spectral bounds $R$ close to 1. The reason for this is that we are approximating the position of the poles in the complex integral *Equation (18)*. We make an approximation around $k = 0$ and *Equation (22)* shows that the position of the complex poles moves closer to $k = 0$ as $s(0) \equiv R^2 \rightarrow 1$.

**General results:**

Using *Equation (21)*

$$m(0) = Kw = \lambda_0 \quad , \quad m''(0) = -Kw\left\langle x^2 \right\rangle \quad ,$$

we find

$$\bar{c}(x) = \frac{D}{1 - R^2} \left\{ \delta(x) + \left[ \frac{Kw\left(1 - 3Kw\right)}{2\left\langle x^2 \right\rangle \left(1 - Kw\right)} \bar{d} + \frac{\left(Kw\right)^2}{\left\langle x^2 \right\rangle^2} \bar{d}^2 \, |x| \right] \exp\left(-\frac{|x|}{\bar{d}}\right) \right\} \quad,$$

with

$$\bar{d} = \sqrt{\left| \frac{\left\langle x^2 \right\rangle}{2} \frac{1}{1 - \lambda_0} \right|}.$$

For the variance we use

$$s(0) = Kw^2 = R^2 \quad, \quad s''(0) = -Kw^2 \left\langle x^2 \right\rangle \quad,$$

to get

$$\overline{\delta c^2}(x) = \frac{D^2}{(1 - R^2)^2} \left\{ \delta(x) + \left[ \frac{Kw^2\left(1 - 3Kw^2\right)}{2\left\langle x^2 \right\rangle \left(1 - Kw^2\right)} d_{\text{eff}} + \frac{\left(Kw^2\right)^2}{\left\langle x^2 \right\rangle^2} d_{\text{eff}}^2 \, |x| \right] \exp\left(-\frac{|x|}{d_{\text{eff}}}\right) \right\} \quad,$$

with

$$d_{\text{eff}} = \sqrt{\left| \frac{\left\langle x^2 \right\rangle}{2} \frac{1}{1 - R^2} \right|} \quad.$$

**Exponential connectivity profile:**
Using an exponential connectivity profile given by

$$p(x) = \frac{1}{2d} e^{-|x|/d} \quad,$$

we find $\left\langle x^2 \right\rangle = 2d^2$ and

$$\bar{d} = \sqrt{\left| \frac{1}{1 - \lambda_0} \right|} d, \quad d_{\text{eff}} = \sqrt{\left| \frac{1}{1 - R^2} \right|} d \quad,$$

with $\lambda_0 = Kw$ for the mean, and $R^2 = Kw^2$ for the variance.
**Gaussian connectivity profile:**
Analogously, using a Gaussian connectivity profile given by

$$p(x) = \frac{1}{\sqrt{2\pi d^2}} e^{-x^2/(2d^2)} \quad,$$

we find $\left\langle x^2 \right\rangle = d^2$, and get

$$\bar{d} = \sqrt{\left| \frac{1}{2} \frac{1}{1 - \lambda_0} \right|} d, \quad d_{\text{eff}} = \sqrt{\left| \frac{1}{2} \frac{1}{1 - R^2} \right|} d \quad. \tag{23}$$

## 10 One-dimensional network with two populations

Realistic neuronal network consist of excitatory and inhibitory neurons. So we need to introduce a second population to our network. Typically, there are more excitatory than inhibitory neurons in the brain. Therefore, we introduce $q$ excitatory neurons for each inhibitory neuron. We place $q$ excitatory neurons and one inhibitory neuron together in one cell. The cells are distributed equally along the ring. For convenience, we define $N \equiv N_{\text{I}}$.

The structure of the connectivity matrix depends on the choice of the activity vector $x$. For later convenience we choose

$$x = \begin{pmatrix} x_1^{(E)} \\ x_1^{(I)} \\ x_2^{(E)} \\ x_2^{(I)} \\ \vdots \\ x_N^{(E)} \\ x_N^{(I)} \end{pmatrix},$$

where $x_i^{(E)}$ is a $q$-dimensional vector denoting the activity of the $q$ excitatory neurons in cell $i$. $M$ is a $(q+1)N \times (q+1)N$-matrix, which qualitatively has the structure

$$M = \begin{pmatrix} EE_{11} & EI_{11} & EE_{12} & EI_{12} & \cdots & EE_{1N} & EI_{1N} \\ IE_{11} & II_{11} & IE_{12} & II_{12} & \cdots & IE_{1N} & II_{1N} \\ EE_{21} & EI_{21} & EE_{22} & EI_{22} & \cdots & EE_{2N} & EI_{2N} \\ IE_{21} & II_{21} & IE_{22} & II_{22} & \cdots & IE_{2N} & II_{2N} \\ \vdots & \vdots & \vdots & \vdots & \ddots & \vdots & \vdots \\ EE_{N1} & EI_{N1} & EE_{N2} & EI_{N2} & \cdots & EE_{NN} & EI_{NN} \\ IE_{N1} & II_{N1} & IE_{N2} & II_{N2} & \cdots & IE_{NN} & II_{NN} \end{pmatrix}. \tag{24}$$

Note that $EE_{ij}$ are $q \times q$ matrices, $EI_{ij}$ are $q \times 1$ matrices, $IE_{ij}$ are $1 \times q$ matrices and $II_{ij}$ are $1 \times 1$ matrices. The entries $ab_{ij}$ describe the connectivities from population $b$ in cell $j$ to population $a$ in cell $i$. The entries are given by

$$ab_{ij} = \begin{cases} \frac{1}{q} w_{ab} K_{ab} \left( p_{ab} \right)_{ij} & \text{if } b = E \\ w_{ab} K_{ab} \left( p_{ab} \right)_{ij} & \text{if } b = I \end{cases}.$$

The difference stems from the fact that we have $q$ times as many excitatory neurons. As the total number of indegrees from excitatory neurons should be given by $K_{aE}$, we need to introduce a reducing factor of $1/q$, as the connection probability is normalized to one.

## 10.1 Dimensionality reduction
In the following, we will reduce the dimensionality of $M$ as done before in the case with one population. First, we make use of the symmetry within the cells. All entries in $M$ corresponding to connections coming from excitatory neurons of the same cell need to be the same. For that reason, we change the basis to

$$e_i^{(E)} = \frac{1}{\sqrt{q}} \begin{pmatrix} 0 \\ 0 \\ \vdots \\ I \\ \vdots \\ 0 \end{pmatrix} \quad, \quad e_i^{(I)} = \begin{pmatrix} 0 \\ 0 \\ \vdots \\ 1 \\ \vdots \\ 0 \end{pmatrix}, \tag{25}$$

where $I$ denotes a $q$-dimensional vector containing only ones. For a full basis, we need to include all the vectors with $I$ being replaced by a vector containing all possible permutations of equal numbers of ±1. In this basis $M$ is block diagonal

$$\begin{pmatrix} M' & 0 \\ 0 & 0 \end{pmatrix},$$

and $M'$ is an $2N \times 2N$ matrix, which has the same qualitative structure as shown in **Equation (24)**, but the submatrices $(\mathrm{ab})_{ij}$ are replaced by

$$ab_{ij} = \begin{cases} w_{\mathrm{EE}}K_{\mathrm{EE}}\left(p_{\mathrm{EE}}\right)_{ij} & \text{if } ab = \mathrm{EE} \\ \sqrt{q}w_{\mathrm{EI}}K_{\mathrm{EI}}\left(p_{\mathrm{EI}}\right)_{ij} & \text{if } ab = \mathrm{EI} \\ w_{\mathrm{IE}}K_{\mathrm{IE}}\left(p_{\mathrm{IE}}\right)_{ij}/\sqrt{q} & \text{if } ab = \mathrm{IE} \\ w_{\mathrm{II}}K_{\mathrm{II}}\left(p_{\mathrm{II}}\right)_{ij} & \text{if } ab = \mathrm{II} \end{cases} \quad .$$

Next, we use translational symmetry of the cells. The translation operator is defined by

$$Tx = T \begin{pmatrix} x_1^{(\mathrm{E})} \\ x_1^{(\mathrm{I})} \\ x_2^{(\mathrm{E})} \\ x_2^{(\mathrm{I})} \\ \vdots \\ x_N^{(\mathrm{E})} \\ x_N^{(\mathrm{I})} \end{pmatrix} = \begin{pmatrix} x_N^{(\mathrm{E})} \\ x_N^{(\mathrm{I})} \\ x_1^{(\mathrm{E})} \\ x_1^{(\mathrm{I})} \\ \vdots \\ x_{N-1}^{(\mathrm{E})} \\ x_{N-1}^{(\mathrm{I})} \end{pmatrix} \quad .$$

As the system is invariant under moving each cell to the next lattice site, $M'$ is invariant under the transformation

$$TM'T^{-1} = M' \quad .$$

Again, the eigenvalues of $T$ can be determined using $T^N = 1$ and they are the same as in the case of one population. But, note that here the eigenspaces corresponding to the single eigenvalues are two dimensional. The eigenvectors

$$v_k^{(\mathrm{E})} = \frac{1}{\sqrt{N}} \begin{pmatrix} 1 \\ 0 \\ e^{\mathrm{i}ka} \\ 0 \\ \vdots \\ e^{\mathrm{i}(N-1)ka} \\ 0 \end{pmatrix} \quad , \quad v_k^{(\mathrm{I})} = \frac{1}{\sqrt{N}} \begin{pmatrix} 0 \\ 1 \\ 0 \\ e^{\mathrm{i}ka} \\ \vdots \\ 0 \\ e^{\mathrm{i}(N-1)ka} \end{pmatrix} \quad ,$$

belong to the same eigenvalue. In this basis, $M'$ is block diagonal, with each block consisting of a 2×2 matrix, corresponding to one value of $k_l = \frac{2\pi l}{L}, l \in \{0, ..., N-1\}$

$$M' = \begin{pmatrix} M_{k_0} & 0 & \cdots & 0 \\ 0 & M_{k_1} & \cdots & 0 \\ \vdots & \vdots & \ddots & \vdots \\ 0 & 0 & \cdots & M_{k_{N-1}} \end{pmatrix} \quad .$$

Since all block matrices can be treated equally, we further reduced the problem to diagonalizing a 2×2 matrix. The submatrices take the form

$$M_k = \begin{pmatrix} m_{\mathrm{EE}}(k) & \sqrt{q}m_{\mathrm{EI}}(k) \\ m_{\mathrm{IE}}(k)/\sqrt{q} & m_{\mathrm{II}}(k) \end{pmatrix} \quad ,$$

with the discrete Fourier transform

$$m_{ab}(k) = K_{ab}w_{ab} \sum_{x=-Na/2}^{Na/2} p_{ab}(x)e^{-ikx} \quad .$$ (26)

Note that $x$ and $k$ are still discrete here, but we could take the continuum limit at this point. The eigenvalues of $M_k$ are given by

$$m_\pm(k) = \frac{1}{2}(m_{EE}(k) + m_{II}(k)) \pm \frac{1}{2}\sqrt{m_{EE}(k)^2 + m_{II}(k)^2 - 2m_{EE}(k)m_{II}(k) + 4m_{EI}(k)m_{IE}(k)} \quad .$$ (27)

The corresponding eigenvectors are

$$v_{1,2}(k) = \mathcal{N}_\pm \begin{pmatrix} \sqrt{q}m_{EI}(k) \\ m_\pm(k) - m_{EE}(k) \end{pmatrix} \quad ,$$ (28)

with normalization $\mathcal{N}_\pm$. The eigenvectors written in the Fourier basis are given by

$$v_\pm(k) = \mathcal{N}_\pm \left[ \sqrt{q}m_{EI}(k)v_k^{(E)} + \left( m_\pm(k) - m_{EE}(k) \right) v_k^{(I)} \right] \quad ,$$ (29)

and we can get the eigenvectors $\tilde{v}_\pm(k)$ in the basis we started with by extending $v_k^{(E)}$ and $v_k^{(I)}$ to vectors similar to *Equation (25)*, where the elements corresponding to excitatory neurons are repeated $q$-times. Note that the normalization of the original basis leads to an additional factor $1/\sqrt{q}$ in the first term of *Equation (29)*.

Analogously, we can find the left eigenvectors of $M$ by conducting the same steps with the transpose of $M$

$$u_\pm(k) = \mathcal{N}_\pm \left[ m_{IE}(k)v_k^{(E)\dagger}/\sqrt{q} + \left( m_\pm(k) - m_{EE}(k) \right) v_k^{(I)\dagger} \right] \quad ,$$ (30)

and the vectors in the original basis $\tilde{u}_\pm(k)$ are obtained similarly to the right eigenvectors. The normalization $\mathcal{N}_\pm$ is chosen such that

$$\tilde{u}_+(k) \cdot \tilde{v}_+(k) = 1 \quad ,$$
$$\tilde{u}_+(k) \cdot \tilde{v}_-(k) = 0 \quad ,$$
$$\tilde{u}_-(k) \cdot \tilde{v}_+(k) = 0 \quad ,$$
$$\tilde{u}_-(k) \cdot \tilde{v}_-(k) = 1 \quad ,$$

which leads to

$$\mathcal{N}_\pm = \sqrt{m_{EI}(k)m_{IE}(k) - \left( m_\pm(k) - m_{EE}(k) \right)^2} \quad .$$

Now, we can express $A^{-1}$ in terms of the eigenvalues and eigenvectors of $M$

$$A^{-1} = 1 + \sum_k \left( \frac{m_+(k)}{1 - m_+(k)}\tilde{v}_+(k) \cdot \tilde{u}_+(k) + \frac{m_-(k)}{1 - m_-(k)}\tilde{v}_-(k) \cdot \tilde{u}_-(k) \right) \quad ,$$ (31)

which leads to

$$A_{ij}^{-1} = \delta_{ij} + \frac{1}{N} \sum_k \mu_{ij}(k)e^{ik|x_i - x_j|} \quad ,$$ (32)

where we defined $\mu(k)$ similar to *Equation (19)*. Let $E$ and $I$ be the sets of indices referring to excitatory or inhibitory neurons respectively. We find

$$\mu_{ij}(k) \equiv \begin{cases} \mu_{EE}(k) & \text{for} \quad i,j \in E \\ \mu_{EI}(k) & \text{for} \quad i \in E, j \in I \\ \mu_{IE}(k) & \text{for} \quad i \in E, j \in I \\ \mu_{II}(k) & \text{for} \quad i,j \in I \end{cases} \quad ,$$

with

$$\begin{aligned} \mu_{EE}(k) &= \frac{1}{q}\frac{m_{EE}(k)+m_{IE}(k)m_{EI}(k)-m_{EE}(k)m_{II}(k)}{1-\zeta(k)} \quad , \\ \mu_{EI}(k) &= \frac{m_{EI}(k)}{1-\zeta(k)} \quad , \\ \mu_{IE}(k) &= \frac{1}{q}\frac{m_{IE}(k)}{1-\zeta(k)} \quad , \\ \mu_{II}(k) &= \frac{m_{II}(k)+m_{IE}(k)m_{EI}(k)-m_{EE}(k)m_{II}(k)}{1-\zeta(k)} \quad , \end{aligned} \tag{33}$$

and

$$\zeta(k) = m_{EE}(k) + m_{II}(k) + m_{EI}(k)m_{IE}(k) - m_{EE}(k)m_{II}(k) \quad .$$

## 10.2 General results

The renormalized noise is evaluated using the same trick as in the one population case. We express the all-ones vector using eigenvectors of the variance matrix $S$

$$I = a\tilde{v}_{+}(0) + b\tilde{v}_{-}(0) \quad .$$

Evaluating the coefficients $a$ and $b$ and inserting the corresponding solutions into *Equation (13)* yields

$$\boldsymbol{D}_{r} = \text{diag}\left(\underbrace{\underbrace{D_{r}^{(E)},\ldots,D_{r}^{(E)}}_{q-\text{times}},D_{r}^{(I)},\underbrace{D_{r}^{(E)},\ldots,D_{r}^{(E)}}_{q-\text{times}},D_{r}^{(I)},\ldots,\underbrace{D_{r}^{(E)},\ldots,D_{r}^{(E)}}_{q-\text{times}},D_{r}^{(I)}}_{N(q+1)-\text{entries}}\right) \quad , \tag{34}$$

with

$$\begin{aligned} D_{r}^{(E)} &= D\left[1 + \frac{s_{EE}(0)+s_{EI}(0)+s_{EI}(0)s_{IE}(0)-s_{EE}(0)s_{II}(0)}{1-R^2}\right] \quad , \\ D_{r}^{(I)} &= D\left[1 + \frac{s_{IE}(0)+s_{II}(0)+s_{EI}(0)s_{IE}(0)-s_{EE}(0)s_{II}(0)}{1-R^2}\right] \quad , \end{aligned}$$

with the eigenvalues $s_{ab}(k)$ of $S$. We again identified the spectral bound

$$R^2 = s_{EE}(0) + s_{II}(0) + s_{EI}(0)s_{IE}(0) - s_{EE}(0)s_{II}(0) \quad . \tag{35}$$

The mean covariances can be written as

$$\bar{c} = \boldsymbol{D}_{r} + \boldsymbol{\mu}\boldsymbol{D}_{r} + \boldsymbol{D}_{r}\boldsymbol{\mu}^{T} + \boldsymbol{\mu}\boldsymbol{D}_{r}\boldsymbol{\mu}^{T} \quad ,$$

where $\boldsymbol{\mu} = \boldsymbol{\mu}(x)$. We can distinguish three different kinds of covariances depending on the type of neurons involved

$$\overline{c_{ij}} \equiv \begin{cases} \overline{c_{EE}}(x) & \text{for} \quad i,j \in E \\ \overline{c_{EI}}(x) & \text{for} \quad i \in E, j \in I \quad \text{or} \quad i \in E, j \in I \\ \overline{c_{II}}(x) & \text{for} \quad i,j \in I \end{cases} \quad .$$

with

$$\overline{c_{\mathrm{EE}}}(x) = D_{\mathrm{r}}^{(\mathrm{E})}\delta(x) + 2D_{\mathrm{r}}^{(\mathrm{E})}\mu_{\mathrm{EE}}(x) + D_{\mathrm{r}}^{(\mathrm{E})}q\left(\mu_{\mathrm{EE}} * \mu_{\mathrm{EE}}\right)(x) + D_{\mathrm{r}}^{(\mathrm{I})}\left(\mu_{\mathrm{EI}} * \mu_{\mathrm{EI}}\right)(x) \quad,$$

$$\overline{c_{\mathrm{EI}}}(x) = D_{\mathrm{r}}^{(\mathrm{E})}\mu_{\mathrm{IE}}(x) + D_{\mathrm{r}}^{(\mathrm{I})}\mu_{\mathrm{EI}}(x) + D_{\mathrm{r}}^{(\mathrm{E})}q\left(\mu_{\mathrm{EE}} * \mu_{\mathrm{IE}}\right)(x) + D_{\mathrm{r}}^{(\mathrm{I})}\left(\mu_{\mathrm{II}} * \mu_{\mathrm{EI}}\right)(x) \quad,$$

$$\overline{c_{\mathrm{II}}}(x) = D_{\mathrm{r}}^{(\mathrm{I})}\delta(x) + 2D_{\mathrm{r}}^{(\mathrm{I})}\mu_{\mathrm{II}}(x) + D_{\mathrm{r}}^{(\mathrm{E})}q\left(\mu_{\mathrm{IE}} * \mu_{\mathrm{IE}}\right)(x) + D_{\mathrm{r}}^{(\mathrm{I})}\left(\mu_{\mathrm{II}} * \mu_{\mathrm{II}}\right)(x) \quad.$$

## 10.3 Long-range limit

From here on, we consider the special case in which the synaptic connections only depend on the type of the presynaptic neuron and not on the type of the postsynaptic neuron. This is in agreement with network parameters used in established cortical network models (**Potjans and Diesmann, 2014**; **Senk et al., 2018**), in which the connection probabilities to both types of target neurons in the same layer are usually of the same order of magnitude. In that case, all expressions become independent of the first population index $A_{ab} \equiv A_b$, and the only expressions we need to evaluate become

$$\mu_a(k) = \gamma_a \frac{m_a(k)}{1 - \zeta(k)} \quad,$$

with

$$\zeta(k) = m_{\mathrm{E}}(k) + m_{\mathrm{I}}(k) \quad,$$

and

$$\gamma_a = \begin{cases} 1 & \text{if } a = \mathrm{I} \\ 1/q & \text{if } a = \mathrm{E} \end{cases} \quad. \tag{36}$$

After taking the continuum limit, we can make a (0,2)-Padé approximation again

$$\mu_a(k) \approx \frac{\gamma_a m_a(0)}{1 - \zeta(0) - \left[\frac{\zeta''(0)}{2} + \left(1 - \zeta(0)\right)\frac{m_a''(0)}{2m_a(0)}\right]k^2} \quad,$$

which leads to the poles

$$k_0 = \pm\sqrt{\left[\frac{\zeta''(0)}{2\zeta(0)}\frac{\zeta(0)}{1 - \zeta(0)} + \frac{m_a''(0)}{2m_a(0)}\right]^{-1}} \quad,$$

or the effective decay constant of the mean covariances

$$\overline{d}_a = \mathrm{Im}(k_0)^{-1} = \sqrt{-\frac{\zeta''(0)}{2\zeta(0)}\frac{\zeta(0)}{1 - \zeta(0)} - \frac{m_a''(0)}{2m_a(0)}} \quad.$$

Using

$$\zeta \equiv \zeta(0) = w_{\mathrm{E}}K_{\mathrm{E}} + w_{\mathrm{I}}K_{\mathrm{I}} \quad,$$

$$\zeta'' \equiv \zeta''(0) = -w_{\mathrm{E}}K_{\mathrm{E}}\langle x^2\rangle_{\mathrm{E}} - w_{\mathrm{I}}K_{\mathrm{I}}\langle x^2\rangle_{\mathrm{I}} \quad,$$

$$m_a(0) = w_a K_a \quad,$$

$$m_a''(0) = -w_a K_a \langle x^2\rangle_a \quad,$$

we get

$$\overline{d}_a = \sqrt{\frac{w_{\mathrm{E}}K_{\mathrm{E}}\langle x^2\rangle_{\mathrm{E}} + w_{\mathrm{I}}K_{\mathrm{I}}\langle x^2\rangle_{\mathrm{I}}}{w_{\mathrm{E}}K_{\mathrm{E}} + w_{\mathrm{I}}K_{\mathrm{I}}}\frac{\zeta}{1 - \zeta} + \frac{\langle x^2\rangle_a}{2}} = \sqrt{\frac{(\omega\kappa\tilde{\eta}^2 + 1)}{\omega\kappa + 1}\frac{\zeta}{1 - \zeta}\frac{\langle x^2\rangle_{\mathrm{I}}}{2} + \frac{\langle x^2\rangle_a}{2}} \quad,$$

after introducing relative parameters

$$\omega = \frac{w_E}{w_I}, \quad \kappa = \frac{K_E}{K_I}, \quad \tilde{\eta}^2 = \frac{\langle x^2 \rangle_E}{\langle x^2 \rangle_I}, \quad \eta = \frac{\lambda_E}{\lambda_I} \quad .$$

The renormalized noise in *Equation (13)* reduces to

$$D_r = \frac{D}{1 - R^2} \quad . \tag{37}$$

The mean covariances are

$$\overline{c_{EE}}(x) = D_r \left[ \delta(x) + 2\mu_E(x) + q \left( \mu_E * \mu_E \right)(x) + \left( \mu_I * \mu_I \right)(x) \right] \quad ,$$

$$\overline{c_{EI}}(x) = D_r \left[ \mu_E(x) + \mu_I(x) + q \left( \mu_E * \mu_E \right)(x) + \left( \mu_I * \mu_I \right)(x) \right] \quad ,$$

$$\overline{c_{II}}(x) = D_r \left[ \delta(x) + 2\mu_I(x) + q \left( \mu_E * \mu_E \right)(x) + \left( \mu_I * \mu_I \right)(x) \right] \quad ,$$

with

$$\mu_a(x) = \gamma_a \frac{m_a(0)}{2(1 - \zeta)\overline{d}_a} \exp\left( -\frac{|x|}{\overline{d}_a} \right) \quad ,$$

and

$$\left( \mu_a * \mu_a \right)(x) = \left( \gamma_a \frac{m(0)}{2 \left( 1 - \zeta \right) \overline{d}_a} \right)^2 \left( \overline{d}_a + |x| \right) \exp\left( -\frac{|x|}{\overline{d}_a} \right) \quad .$$

Note that expressions coming from both populations contribute to each kind of covariance. Therefore, all mean covariances contain a part that decays with either of the decay constants we just determined. If, for example, the inhibitory decay constant is much larger than the excitatory one, $\overline{c_{EI}}(x)$ will decay with the largest decay constant in the long-range limit

**Exponential connectivity profile:**
Just as in Section 9.4 we get

$$\overline{d}_a = \sqrt{\frac{(\omega\kappa\eta^2 + 1)}{\omega\kappa + 1} \frac{\lambda_0}{1 - \lambda_0} d_I^2 + d_a^2} \quad , \quad d_{\text{eff},a} = \sqrt{\frac{(\omega^2\kappa\eta^2 + 1)}{\omega^2\kappa + 1} \frac{R^2}{1 - R^2} d_I^2 + d_a^2} \quad ,$$

with $\lambda_0 = w_E K_E + w_I K_I$ for the decay constant of the mean covariances, and $R^2 = w_E^2 K_E + w_I^2 K_I$ for the decay constant of the variances.

**Gaussian connectivity profile:**
And similar to Section 9.4 we get

$$\overline{d}_a = \sqrt{\frac{(\omega\kappa\eta^2 + 1)}{\omega\kappa + 1} \frac{\lambda_0}{1 - \lambda_0} \frac{d_I^2}{2} + \frac{d_a^2}{2}} \quad , \quad d_{\text{eff},a} = \sqrt{\frac{(\omega\kappa\eta^2 + 1)}{\omega\kappa + 1} \frac{\lambda_0}{1 - \lambda_0} \frac{d_I^2}{2} + \frac{d_a^2}{2}} \quad .$$

# 11 Two-dimensional network with one population

In the following, we are considering two-dimensional networks, which are supposed to mimic a single-layered cortical network. Neurons are positioned on a two-dimensional lattice ($N_x \times N_y$ grid) with periodic boundary conditions in both dimensions (a torus). We define the activity vector to be of the form

$$x = \begin{pmatrix} x_{1,1} \\ x_{1,2} \\ \vdots \\ x_{1,N_y} \\ x_{2,1} \\ \vdots \\ x_{2,N_y} \\ \vdots \\ x_{N_x,1} \\ \vdots \\ \vdots \\ x_{N_x,N_y} \end{pmatrix} .$$

The connectivity matrix is defined correspondingly.

## 11.1 Dimensionality reduction

In two dimensions we have to define two translation operators that move all neurons either one step in the $x$-direction, or the $y$-direction, respectively. They are defined via their action on $x$

$$\boldsymbol{T}_x \boldsymbol{x} = \begin{pmatrix} x_{N_x,1} \\ x_{N_x,2} \\ \vdots \\ x_{N_x,N_y} \\ x_{1,1} \\ \vdots \\ x_{1,N_y} \\ \vdots \\ x_{N_x-1,1} \\ \vdots \\ x_{N_x-1,N_y} \end{pmatrix} \quad , \quad \boldsymbol{T}_y \boldsymbol{x} = \begin{pmatrix} x_{1,N_y} \\ x_{1,1} \\ \vdots \\ x_{1,N_y-1} \\ x_{2,N_y} \\ \vdots \\ x_{2,N_y-1} \\ \vdots \\ x_{N_x,N_y} \\ \vdots \\ x_{N_x,N_y-1} \end{pmatrix} . \tag{38}$$

Similar reasoning as in one dimension leads to the eigenvalues

$$\mathrm{e}^{-\mathrm{i}k_l^{(x)}a} \quad , \quad k_l^{(x)} = \frac{2\pi}{L_x}l \quad , \quad l \in \{0, 1, ..., N_x - 1\} \quad ,$$

and similar for the $y$-direction. The eigenvectors can be inferred from the recursion relations

$$\boldsymbol{T}_x \boldsymbol{v} = \mathrm{e}^{-\mathrm{i}k_l^{(x)}a}\boldsymbol{v} \quad \Rightarrow \quad v_{(\alpha+1)\beta} = \mathrm{e}^{\mathrm{i}k_l^{(x)}a}v_{\alpha\beta} \quad ,$$
$$\boldsymbol{T}_y \boldsymbol{v} = \mathrm{e}^{-\mathrm{i}k_l^{(y)}a}\boldsymbol{v} \quad \Rightarrow \quad v_{\alpha(\beta+1)} = \mathrm{e}^{\mathrm{i}k_l^{(y)}a}v_{\alpha\beta} \quad ,$$

where entries $v_{\alpha\beta}$ of the vector $\boldsymbol{v}$ are defined analogously to *Equation (38)*. The eigenvectors are given by

$$v_{\boldsymbol{k}} = \frac{1}{\sqrt{N_x N_y}} \begin{pmatrix} \boldsymbol{v}^{(x)} \\ e^{ik^{(y)}a}\boldsymbol{v}^{(x)} \\ \vdots \\ e^{-ik^{(y)}a}\boldsymbol{v}^{(x)} \end{pmatrix} \quad , \quad \boldsymbol{v}^{(x)} = \begin{pmatrix} 1 \\ e^{ik^{(x)}a} \\ \vdots \\ e^{i\frac{N_x-1}{2}k^{(x)}a} \\ e^{-i\frac{N_x-1}{2}k^{(x)}a} \\ \vdots \\ e^{-ik^{(x)}a} \end{pmatrix} \quad ,$$

where we suppressed the subscripts of $k^{(x)}$ and $k^{(y)}$ again. Using that these eigenvectors are eigenvectors of $\boldsymbol{M}$ as well, yields the eigenvalues of $\boldsymbol{M}$

$$m_{\boldsymbol{k}} = v_{\boldsymbol{k}}^{\dagger} \boldsymbol{M} v_{\boldsymbol{k}} = Kw \sum_x \sum_y p(|\boldsymbol{x}|)e^{-i\boldsymbol{k}\cdot\boldsymbol{x}} \quad .$$

In the continuum limit, this becomes the two-dimensional Fourier transform

$$m(\boldsymbol{k}) = Kw \int \mathrm{d}^2 x\, p(\boldsymbol{x})e^{-i\boldsymbol{k}\cdot\boldsymbol{x}}. \tag{39}$$

The inverse of A is given by

$$A^{-1}(\boldsymbol{x}) = \delta(\boldsymbol{x}) + \mu(\boldsymbol{x}) \quad , \tag{40}$$

with the inverse two-dimensional Fourier transform

$$\mu(\boldsymbol{x}) = \frac{1}{(2\pi)^2} \int \mathrm{d}^2 k\, \frac{m(\boldsymbol{k})}{1 - m(\boldsymbol{k})} e^{i\boldsymbol{k}\cdot\boldsymbol{x}} \quad . \tag{41}$$

The expression for the renormalized noise is the same as in the one-dimensional case with one population. Hence, the mean covariances are given by

$$\bar{c}(\boldsymbol{x}) = \frac{D}{1 - R^2} \left[ \delta(\boldsymbol{x}) + 2\mu(\boldsymbol{x}) + \left( \mu ** \mu \right)(\boldsymbol{x}) \right] \quad , \tag{42}$$

which is the one-dimensional expression, except for the convolution, which is replaced by its two-dimensional analogon denoted here by the double asterisk.

## 11.2 Long-range limit
Employing the symmetry of the connectivity kernel, we rewrite the integral in $\mu(\boldsymbol{x})$ using polar coordinates

$$\mu(\boldsymbol{x}) = \frac{1}{(2\pi)^2} \int_0^\infty \mathrm{d}k \int_0^{2\pi} \mathrm{d}\varphi\, k \frac{m(k)}{1 - m(k)} e^{ikr\cos(\varphi)} \quad , \tag{43}$$

with $r = |\boldsymbol{x}|$, and make a Padé approximation of order (0,2) of the integration kernel

$$\mu(\boldsymbol{x}) = \frac{1}{(2\pi)^2} \int_0^\infty \mathrm{d}k \int_0^{2\pi} \mathrm{d}\varphi\, k \frac{m(0)}{1 - m(0) - \frac{m''(0)}{2m(0)}k^2} e^{ikr\cos(\varphi)} \quad . \tag{44}$$

Following (**Goldenfeld, 1992**, p.160f), we can interpret this as calculating the Green's function of the heat equation

$$\left[ 1 - m(0) + \frac{m''(0)}{2m(0)} \nabla^2 \right] \mu(\boldsymbol{x}) = m(0)\delta(r) \quad , \tag{45}$$

which can be solved, using the fact that $\mu(\boldsymbol{x})$ can only be a function of the radial distance $r$, due to the given symmetry of the kernel. Rewriting leads to

$$\left[ -\frac{1}{r}\frac{d}{dr}\left(r\frac{d}{dr}\right) + \bar{d}^{-2} \right]\mu(r) = \Gamma\delta(r) \quad,$$

with the effective decay constant

$$\bar{d} = \sqrt{-\frac{m''(0)}{2m(0)}\frac{1}{1-m(0)}} \quad, \tag{46}$$

and $\Gamma = -2m(0)^2/m''(0)$. Defining $\rho \equiv r/\bar{d}$, $\tilde{\mu}(\rho) \equiv \mu(r/\bar{d})$, and using $\delta(\rho\bar{d}) = \bar{d}^{-2}\delta(\rho)$, we get

$$\left[ -\frac{1}{\rho}\frac{d}{d\rho}\left(\rho\frac{d}{d\rho}\right) + 1 \right]\tilde{\mu}(\rho) = \Gamma\delta(\rho) \quad.$$

The solution to this equation is given by the modified Bessel function of second kind and zeroth order $K_0$

$$\tilde{\mu}(\rho) = \frac{\Gamma}{2\pi}K_0(\rho) \quad.$$

Reinserting the defined variables yields

$$\mu(r) = -\frac{m(0)^2}{\pi m''(0)}K_0\left(\frac{r}{\bar{d}}\right) \quad. \tag{47}$$

Note that the modified Bessel functions of second kind decay exponentially for long distances

$$K_i\left(\frac{r}{\bar{d}}\right) \xrightarrow{r\to\infty} \sqrt{\frac{\pi\bar{d}}{2r}}e^{-r/\bar{d}} \quad. \tag{48}$$

But, consider that the inverse square root of the distance appears in front of the exponential. Formally, this is the one-dimensional result. The only difference here is, that $m(k)$ is a two-dimensional Fourier transform instead of a one-dimensional one and $\mu(r)$ contains modified Bessel functions of second kind instead of exponentials.

In order to evaluate the expression for the mean covariances *Equation 42*, one needs to calculate the two-dimensional convolution of a modified Bessel function of second kind with itself, for which we use the following trick

$$\begin{aligned}
(K_0 ** K_0)\left(\frac{r}{\bar{d}}\right) &= \mathcal{F}^{-1}\left[\tilde{K}_0 \cdot \tilde{K}_0\right]\left(\frac{r}{\bar{d}}\right) \\
&= \frac{1}{2\pi}\mathcal{H}^{-1}\left[\frac{1}{(\beta+k^2)^2}\right]\left(\sqrt{\beta}r\right) \\
&= -\frac{1}{2\pi}\frac{d}{d\beta}\mathcal{H}^{-1}\left[\frac{1}{\beta+k^2}\right]\left(\sqrt{\beta}r\right) \\
&= -\frac{1}{2\pi}\frac{d}{d\beta}K_0\left(\sqrt{\beta}r\right) \\
&= \frac{d_{\text{eff},\mu}r}{4\pi}K_1\left(\frac{r}{\bar{d}}\right) \quad,
\end{aligned}$$

where $\mathcal{F}$ denotes the Fourier transform, $\mathcal{H}$ denotes the Hankel transform, and $\beta = \bar{d}^{-2}$. The last step can be found in *Abramowitz and Stegun, 1964*, 9.6.27.

The mean covariances are given by

$$\begin{aligned}
\bar{c}(r) &= \frac{D}{1-R^2}\left[\delta(r) - 2\frac{m(0)^2}{\pi m''(0)}K_0\left(\frac{r}{\bar{d}}\right) + \frac{m(0)^4}{m''(0)^2}\frac{\bar{d}r}{4\pi^3}K_1\left(\frac{r}{\bar{d}}\right)\right] \\
&\xrightarrow{r\to\infty} \frac{D}{1-R^2}\left[\delta(r) - \frac{m(0)^2}{m''(0)}\sqrt{\frac{2\bar{d}}{\pi r}}e^{-r/\bar{d}} + \frac{m(0)^4}{m''(0)^2}\sqrt{\frac{\bar{d}^3r}{32\pi^5}}e^{-r/\bar{d}}\right] \quad.
\end{aligned}$$

Using

$$m(0) = Kw \equiv \zeta \quad, \quad m''(0) = -Kw\left\langle r^2 \right\rangle \quad, \tag{49}$$

we get the effective decay constant

$$\bar{d} = \sqrt{\frac{1}{1-\zeta} \frac{\langle r^2 \rangle}{2}} d \quad . \tag{50}$$

**Exponential connectivity profile:**

Using a two-dimensional exponential connectivity profile

$$p(\boldsymbol{x}) = \frac{1}{2\pi d^2} \mathrm{e}^{-|\boldsymbol{x}|/d} \quad ,$$

leads to $\langle r^2 \rangle = 6d^2$, and we get

$$\bar{d} = \sqrt{\frac{3}{1-\lambda_0}} d \quad , \quad d_{\mathrm{eff}} = \sqrt{\frac{3}{1-R^2}} d \quad ,$$

with $\lambda_0 = Kw$, and $R^2 = Kw^2$.

**Gaussian connectivity profile:**

Using a two-dimensional Gaussian connectivity profile

$$p(\boldsymbol{x}) = \frac{1}{2\pi d^2} \mathrm{e}^{-\boldsymbol{x}^2/(2d^2)} \quad ,$$

leads to $\langle r^2 \rangle = 2d^2$, and we get

$$\bar{d} = \sqrt{\frac{1}{1-\lambda_0}} d \quad , \quad d_{\mathrm{eff}} = \sqrt{\frac{1}{1-R^2}} d \quad .$$

## 11.3 Note on higher order approximation

While the (0,2)-Padé approximation seems to yield good results for the one-dimensional cases, in two dimensions the results only coincide for large spectral radii (*Appendix 1—figure 7*). One can extract a higher order approximation of the poles of the integration kernel of $\mu(\boldsymbol{x})$ and thereby the effective decay constant $d_{\mathrm{eff}}$ using the DLog-Padé-method, for which one calculates an $(n, n+1)$-Padé approximation of the logarithmic derivative of the integration kernel around zero (*Pelizzola, 1994*). Using a (1,2)-Padé approximation leads to

$$\bar{d} = \sqrt{-\frac{3(2m(0)-1)m''(0)^2 + (1-m(0))m(0)m''''(0)}{6m''(0)m(0)(1-m(0))}} \quad ,$$

which coincides with our previous results in the limit $m(0) \to 1$, and thus for large spectral radii. Note that this expression contains the fourth moment of the connectivity kernel $m''''(0) = wK\langle x^4 \rangle$.

## 12 Two-dimensional network with two populations

Finally, we consider a two-dimensional network with two populations of neurons. As in the one dimensional case, the neurons are gathered in cells, which contain one inhibitory and $q$ excitatory neurons. Again, they are placed on a two-dimensional lattice with periodic boundary conditions. The activity vector takes the form

$$x = \begin{pmatrix} \boldsymbol{x}_{1,1}^{(E)} \\ x_{1,1}^{(I)} \\ \boldsymbol{x}_{1,2}^{(E)} \\ x_{1,2}^{(I)} \\ \vdots \\ \boldsymbol{x}_{1,N_y}^{(E)} \\ x_{1,N_y}^{(I)} \\ \boldsymbol{x}_{2,1}^{(E)} \\ x_{2,1}^{(I)} \\ \vdots \\ \boldsymbol{x}_{N_x,N_y}^{(E)} \\ x_{N_x,N_y}^{(I)} \end{pmatrix} \quad , \tag{51}$$

where $\boldsymbol{x}_{i,j}^{(E)}$ denotes a $q$-dimensional vector.

## 12.1 Dimensionality reduction

We apply the procedure developed so far, which leads to the results we found in the one-dimensional case with two populations, with Fourier transforms and convolutions replaced by their two-dimensional analogons and modified Bessel functions of second kind instead of exponentials. So, we end up with

$$\overline{c_{\text{EE}}}(x) = D_{\text{r}}^{(E)}\delta(x) + 2D_{\text{r}}^{(E)}\mu_{\text{EE}}(x) + D_{\text{r}}^{(E)}q\left(\mu_{\text{EE}} * *\mu_{\text{EE}}\right)(x) + D_{\text{r}}^{(I)}\left(\mu_{\text{EI}} * *\mu_{\text{EI}}\right)(x) \quad ,$$

$$\overline{c_{\text{EI}}}(x) = D_{\text{r}}^{(E)}\mu_{\text{IE}}(x) + D_{\text{r}}^{(I)}\mu_{\text{EI}}(x) + D_{\text{r}}^{(E)}q\left(\mu_{\text{EE}} * *\mu_{\text{IE}}\right)(x) + D_{\text{r}}^{(I)}\left(\mu_{\text{II}} * *\mu_{\text{EI}}\right)(x) \quad ,$$

$$\overline{c_{\text{II}}}(x) = D_{\text{r}}^{(I)}\delta(x) + 2D_{\text{r}}^{(I)}\mu_{\text{II}}(x) + D_{\text{r}}^{(E)}q\left(\mu_{\text{IE}} * *\mu_{\text{IE}}\right)(x) + D_{\text{r}}^{(I)}\left(\mu_{\text{II}} * *\mu_{\text{II}}\right)(x) \quad ,$$

and $\mu_{ab}(x)$ given by (33) and the two-dimensional Fourier transform

$$m_{ab}(\boldsymbol{k}) = K_{ab}w_{ab}\int \mathrm{d}^2 x\, p_{ab}(\boldsymbol{x})\mathrm{e}^{-\mathrm{i}\boldsymbol{k}\cdot\boldsymbol{x}} \quad .$$

The renormalized noise is given by **Equation 34** with spectral bound **Equation 35**, with the eigenvalues $s_{ab}(k)$ replaced by the two-dimensional Fourier transforms $s_{ab}(\boldsymbol{k})$.

## 12.2 Long-range limit

Again, considering the special case in which the synaptic connections only depend on the type of the presynaptic neuron and not on the type of the postsynaptic neuron, the expressions simplify to

$$\mu_a(k) = \frac{m_a(k)}{1 - \zeta(k)} \quad , \tag{52}$$

with

$$\zeta(k) = m_{\text{E}}(k) + m_{\text{I}}(k) \quad .$$

Padé approximation of the Fourier kernel, integration using (**Goldenfeld, 1992**, p.160f) and suppressing the zero arguments of $\zeta$ and $m_a$ leads to

$$\mu_a(r) = \quad -\frac{\gamma_a w_a K_a}{2\pi(1-\zeta)\overline{d}_a^2}\mathrm{K}_0\left(\frac{r}{\overline{d}_a}\right)$$

$$\xrightarrow{r\to\infty} \quad -\frac{\gamma_a w_a K_a}{(1-\zeta)}\sqrt{\frac{1}{8\pi r\overline{d}_a^3}}\mathrm{e}^{-r/\overline{d}_a} \tag{53}$$

with

$$\bar{d}_a = \sqrt{-\frac{\zeta''}{2\zeta}\frac{\zeta}{1-\zeta} - \frac{m_a''}{2m_a}} \quad .$$

After introducing the same relative parameters as in Section 10.3, we find

$$\bar{d}_a = \sqrt{\frac{(\omega\kappa\tilde{\eta}^2 + 1)}{\omega\kappa + 1}\frac{\zeta}{1-\zeta}\frac{\langle x^2\rangle_{\mathrm{I}}}{2} + \frac{\langle x^2\rangle_a}{2}} \quad . \tag{54}$$

The two-dimensional convolutions are given by

$$\begin{aligned}
(\mu_a ** \mu_a)(r) &= \left[\frac{\gamma_a w_a K_a}{4(1-\zeta)}\right]^2 \frac{1}{\pi^3 \bar{d}_a^3} K_1\left(\frac{r}{\bar{d}_a}\right)\\
&\xrightarrow{r\to\infty} \left[\frac{\gamma_a w_a K_a}{4(1-\zeta)}\right]^2 \sqrt{\frac{1}{2\pi^5 \bar{d}_a^5 r}} \mathrm{e}^{-r/\bar{d}_a} \quad .
\end{aligned} \tag{55}$$

The renormalized noise simplifies to *Equation 37*. The mean covariances are given by

$$\begin{aligned}
\overline{c_{\mathrm{EE}}}(x) &= D_{\mathrm{r}}\left[\delta(x) + 2\mu_{\mathrm{E}}(x) + q\left(\mu_{\mathrm{E}} ** \mu_{\mathrm{E}}\right)(x) + \left(\mu_{\mathrm{I}} ** \mu_{\mathrm{I}}\right)(x)\right] \quad ,\\
\overline{c_{\mathrm{EI}}}(x) &= D_{\mathrm{r}}\left[\mu_{\mathrm{E}}(x) + \mu_{\mathrm{I}}(x) + q\left(\mu_{\mathrm{E}} ** \mu_{\mathrm{E}}\right)(x) + \left(\mu_{\mathrm{I}} ** \mu_{\mathrm{I}}\right)(x)\right] \quad ,\\
\overline{c_{\mathrm{II}}}(x) &= D_{\mathrm{r}}\left[\delta(x) + 2\mu_{\mathrm{I}}(x) + q\left(\mu_{\mathrm{E}} ** \mu_{\mathrm{E}}\right)(x) + \left(\mu_{\mathrm{I}} ** \mu_{\mathrm{I}}\right)(x)\right] \quad .
\end{aligned} \tag{56}$$

Remember that the result for the variances of the covariances is obtained by substituting $D_{\mathrm{r}}$ by its square, and $w_a$, or $\omega$ respectively, by its square, and setting $\zeta = R^2$.

*Equation (2)* in the main text can be proven by inserting the result for

$$\bar{d}_{\mathrm{E}}^2 - \bar{d}_{\mathrm{I}}^2 = \frac{\langle x^2\rangle_{\mathrm{E}}}{2} - \frac{\langle x^2\rangle_{\mathrm{I}}}{2} = \text{const.}\cdot(d_{\mathrm{E}}^2 - d_{\mathrm{I}}^2) \quad .$$

Using an exponential connectivity profile yields const. = 3, a Gaussian connectivity profile yields const. = 1.

**Exponential connectivity profile:**
Using the results from 11.2, we find

$$\bar{d}_a = \sqrt{3\left[\frac{(\omega\kappa\eta^2 + 1)}{\omega\kappa + 1}\frac{\lambda_0}{1-\lambda_0}d_I + d_a\right]} \quad , \quad d_{\mathrm{eff},a} = \sqrt{3\left[\frac{(\omega^2\kappa\eta^2 + 1)}{\omega^2\kappa + 1}\frac{R^2}{1-R^2}d_I + d_a\right]} \quad ,$$

with $\lambda_0 = w_{\mathrm{E}}K_{\mathrm{E}} + w_{\mathrm{I}}K_{\mathrm{I}}$, and $R^2 = w_{\mathrm{E}}^2 K_{\mathrm{E}} + w_{\mathrm{I}}^2 K_{\mathrm{I}}$.
**Gaussian connectivity profile:**
Using the results from 11.2, we find

$$\bar{d}_a = \sqrt{\frac{(\omega\kappa\eta^2 + 1)}{\omega\kappa + 1}\frac{\lambda_0}{1-\lambda_0}d_I + d_a} \quad , \quad d_{\mathrm{eff},a} = \sqrt{\frac{(\omega^2\kappa\eta^2 + 1)}{\omega^2\kappa + 1}\frac{R^2}{1-R^2}d_I + d_a} \quad .$$

## 12.3 Higher order approximation
Using a (1,2)-DLog-Padé method as in Section 11.3 yields

$$\bar{d}_a = \sqrt{-\frac{(1-\zeta)^2\left(m_a m_a'''' - 3m_a''^2\right) + m_a^2\left[(1-\zeta)\zeta'''' + 3\zeta''^2\right]}{6m_a(1-\zeta)\left[(1-\zeta)m_a'' + m\zeta''\right]}} \quad , \tag{57}$$

which again contains the fourth moments of the connectivity kernels.

## 13 Validation of theory

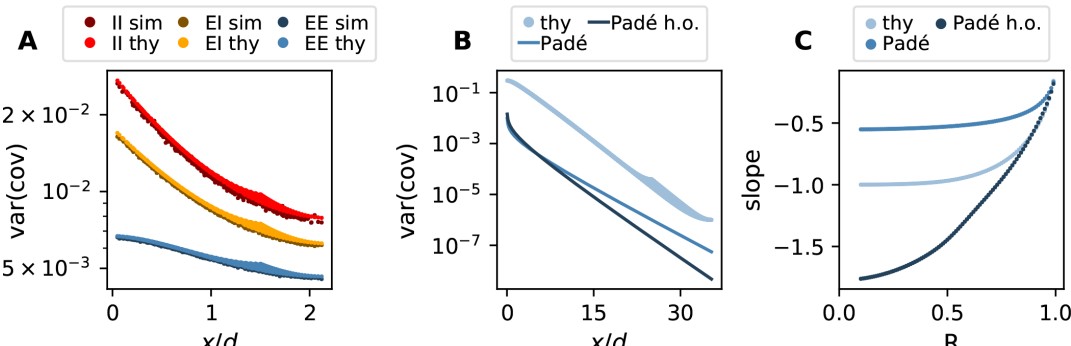

**Appendix 1—figure 7.** Comparison of simulation and theory. (**A**) Variance of EE, EI, and II covariances as a function of distance. Darker dots are the results of the simulation. Lighter ones are the prediction of the discrete theory. (**B**) Variance of EE covariances as a function of distance (**Equation (56)** for variances). The lightest blue dots are the predictions of the discrete theory ($\mu_a$ replaced by the discrete Fourier transform of **Equation (52)**, taking into account Section 7), the medium blue line is the (0,2)-Padé prediction ($\mu_a$ replaced by its Padé approximation **Equation (53)**, taking into account Section 7), and the dark blue line is the higher order (1,2)-DLog-Padé prediction ($\mu_a$ replaced by its Padé approximation **Equation (53)**, using **Equation (57)**, and taking into account Section 7). (**C**) Fitted slope of linear regions in panel B for different spectral bounds $R$ (light blue: discrete theory, medium blue: Padé approximation, dark blue: higher order Padé approximation).

In order to validate our results, we performed simulations, in which an effective connectivity matrix $W$ of a two-dimensional network was drawn randomly, and covariances were calculated using the result from **Pernice et al., 2011**, **Trousdale et al., 2012**, and **Lindner et al., 2005**

$$c(W) = \left(1 - W\right)^{-1} D \left(1 - W\right)^{-T} \quad .$$

The elements of the different components $W_{ab}$ of the effective connectivity matrix, similar to **Equation (24)**, were drawn from a binomial distribution with $K_b$ trials and a success probability of $\gamma_b p_b(|x|)$, with $\gamma_b$ given by **Equation (36)** and $|x|$ denoting the distance between the neurons.

We compared the results to the predictions by our discrete theory, continuum theory, and the long-range limit. We did this for all cases presented above: one dimension with one population, one dimension with two populations, two dimensions with one population, and two dimensions with two populations. In the cases of two populations we solely considered the special case of synaptic connections only depending on the type of the presynaptic neuron. The first three cases are not reported here. We simulated several sets of parameters, varying the number of neurons, the number of inputs, the decay constants and the spectral bound, of which we only report the one using the parameters listed in **Appendix 1—table 3**, because the results do not differ qualitatively. Using

$$R^2 = s(0) = K_E w_E^2 + K_I w_I^2 \quad ,$$

and choosing

$$\frac{w_I}{w_E} = -\frac{N_E}{N_I} = -q \quad ,$$

we calculated the synaptic weights

$$w_E = \frac{R}{\sqrt{K_E + q^2 K_I}} \quad , \quad w_I = -\frac{qR}{\sqrt{K_E + q^2 K_I}} \quad .$$

The comparison of simulation and discrete theory is shown in **Appendix 1—figure 7a**. Simulation and theory match almost perfectly. The continuum theory, which is shown in **Figure 3D,E** of the main text, matches as well as the discrete theory (not shown here). The slight shift in y-direction in **Appendix 1—figure 7a** is due to the fact that in the random realization of

the network the spectral bound is not exactly matching the desired value, but is slightly different for each realization and distributed around the chosen value. This jittering around the real spectral bound is more pronounced as $R \to 1$. Note that the simulated networks were small compared to the decay constant of the connectivity profile, in order to keep simulation times reasonable. This is why the variances do not fall off linearly in the semi-log plot. The kink and the related spreading starting around $x/d = 1.5$ is a finite size effect due to periodic boundary conditions: The maximal distance of two neurons along the axes in units of spatial decay constants is $(N/2)/d \approx 1.5$. Because of the periodic boundary conditions, the covariances between two neurons increases once the distance between them exceeds the maximal distance along an axis. This, together with the fact that the curve is the result of the discrete Fourier transform of *Equation 52*, implies a zero slope at the boundary. This holds for any direction in the two dimensional plane, but the maximal distances between two neurons is longer for directions not aligned with any axis and depends on the precise direction, which explains the observed spreading.

In order to validate the long-range limit, we compared our discrete theory with the result from the Padé approximation at large distances (*Appendix 1—figure 7b*). We do not expect the Padé approximation to hold at small distances. We are mainly interested in the slope of the variance of covariances, because the slope determines how fast typical pairwise covariances decay with increasing inter-neuronal distance. The slope at large distances for the (0,2)-Padé approximation is smaller than the prediction by our theory, but the higher order approximation matches our theory very well (*Appendix 1—figure 7C*). In the limit $R \to 1$ both Padé predictions yield similar results. The absolute value of the covariances in the Padé approximation can be obtained from a residue analysis. The (0,2)-Pade approximation yields absolute values with a small offset, analogous to the slope results. Calculating the residues for the (1,2)-DLog Padé approximation would lead to a better approximation. Note that for plotting the higher order prediction in *Appendix 1—figure 7b*, we just inserted *Equation (57)* into *Equation (53)* and *Equation (55)*.

**Appendix 1—table 3.** Parameters used to create theory figures. Decay constants in units of lattice constant $a$.

|  | Figure 3B, C | Figure 3D, E | App 1—fig 7A | App 1—fig 7B |  |
|---|---|---|---|---|---|
| $N_x$ | 61 | 201 | 61 | 1,001 | Number of neurons in x-direction |
| $N_y$ | 61 | 201 | 61 | 1,001 | Number of neurons in y-direction |
| $q$ | 4 | 4 | 4 | 4 | Ratio of excitatory to inhibitory neurons |
| $K_E$ | 100 | 100 | 100 | 100 | Number of excitatory inputs per neuron |
| $K_I$ | 50 | 50 | 50 | 50 | Number of inhibitory inputs per neuron |
| $d_E$ | 20 | 20 | 20 | 20 | Decay constant of excitatory connectivity profile |
| $d_I$ | 10 | 10 | 10 | 10 | Decay constant of inhibitory connectivity profile |
| $D$ | 1 | 1 | 1 | 1 | Squared noise amplitude |
| $R$ | 0.95 | 0.95 | 0.8 | 0.95 | Spectral bound |
|  | exponential | exponential | exponential | exponential | Connectivity kernel |

## 14 Parameters of NEST simulation

**Appendix 1—table 4.** Parameters used for NEST simulation and subsequent analysis.

**Network parameters**

| | | |
|---|---|---|
| $N$ | 2000 | Number of neurons |
| $p$ | 0.1 | Connection probability |
| $\tau$ | 1 ms | Time constant |
| $\sigma_\mu$ | 1 Hz | Standard deviation of external input |
| $\sigma_{\text{noise}}$ | 0.1 Hz | Standard deviation of noise |
| $R$ | $[0.1, 0.2, ..., 0.9]$ | Spectral bound |
| $\epsilon$ | 0.1 | Parameter controlling difference of two network simulations |

**Simulation Parameters**

| | | |
|---|---|---|
| $dt$ | 0.1 ms | Simulation step size |
| $t_{init}$ | 100 ms | Initialization time |
| $t_{sim}$ | 2000000 ms | Simulation time without initialization time |
| $t_{sample}$ | 1 ms | Sample resolution at which rates where recorded |

**Analysis Parameters**

| | | |
|---|---|---|
| $N_{\text{sample}}$ | 200 | Sample size |
| $T$ | 100 ms | Correlation time window |

## 15 Sources of heterogeneity

Sparseness of connections is a large source of heterogeneity in cortical networks. It contributes strongly to the variance of effective connection weights that determines the spectral bound, the quantity that controls stability of balanced networks (*Sompolinsky et al., 1988*; *Dahmen et al., 2019*): Consider the following simple model $W_{ij} = \mathcal{W}_{ij}\zeta_{ij}$ for the effective connection weights $W_{ij}$, where $\zeta_{ij} \in \{0, 1\}$ are independent Bernoulli numbers, which are 1 with probability $p$ and 0 with probability $1 - p$, and $\mathcal{W}_{ij}$ are independently distributed amplitudes. The $\zeta_{ij}$ encode the sparseness of connections and the $\mathcal{W}_{ij}$ encode the experimentally observed distributions of synaptic amplitudes and single neuron heterogeneities that lead to different neuronal gains. Since $\mathcal{W}_{ij}$ and $\zeta_{ij}$ are independent, the variance of $W_{ij}$ is

$$\text{Var}(W_{ij}) = p \cdot \text{Var}(\mathcal{W}_{ij}) + p(1 - p) \cdot \text{Mean}(\mathcal{W}_{ij})^2 .$$

For low connection probabilities as observed in cortex ($p(1 - p) \approx p$), assessing the different contributions to the variance thus amounts to comparing the mean and standard deviation of $\mathcal{W}_{ij}$. Even though synaptic amplitudes are broadly distributed in cortical networks, one typically finds that their mean and standard deviation are of the same magnitude (see e.g. *Sayer et al., 1990*, Tab 1; *Feldmeyer et al., 1999*, Tab 1; *Song et al., 2005*, Fig.1; *Lefort et al., 2009*, Tab 2; *Ikegaya et al., 2013*, Fig.1; *Loewenstein et al., 2011*, Fig. 2). Sparseness of connections (second term on the right hand side) is thus one of the dominant contributors to the variance of connections. For simplicity, the other sources, in particular the distribution of synaptic amplitudes, are left out in this study. They can, however, be straight-forwardly added in the model and the theoretical formalism, because it only depends on $\text{Var}(W_{ij})$.

