## [Editor Report]

This is a thorough study showing that long-range correlations in the brain can arise without common input drive or long-range anatomical connections. These long-range correlations are modulated by the animal's behavioral state, a surprising finding that suggests a computational role for control of this kind of correlation. The paper details some analytical methods for modeling this behavior in disordered systems. The work will be of broad interest to neuroscientists, computational biologists, and biophysicists.

---

## [Decision Letter]

**Decision letter after peer review:**

Thank you for submitting your article "Global organization of neuronal activity only requires unstructured local connectivity" for consideration by *eLife*. Your article has been reviewed by 2 peer reviewers, one of whom is a member of our Board of Reviewing Editors, and the evaluation has been overseen by Timothy Behrens as the Senior Editor. The reviewers have opted to remain anonymous.

Essential revisions:

1) The scenario put forward by the authors describe the data well, but a major drawback is that it needs fine tuning for the network to be close to the critical point. There are multiple other, and perhaps simpler, scenarios that could reproduce the data described by the authors, but they are not adequately discussed in this paper:

A: Fluctuating and broadly divergent external inputs could in principle generate the observed correlation structure. The authors quickly dismiss this scenario in the discussion: 'In such a scenario, covariances have been shown to be predominantly generated locally rather than from external inputs (4,15)'. It was not clear where this is shown in these papers. The authors should explain much more clearly why they believe the external input scenario is unlikely (see also issue #5).

B: Recurrent connectivity correlated with neuronal selectivity, as in visual cortex, but with salt-and-pepper organization. This would be a scenario similar to what has been proposed in multiple visual cortex models (see old work by Tsodyks, Sompolinsky, and more recently ref (14)). The authors also dismiss this scenario because of the lack of patchiness of the correlation structure in M1, but we would not expect any patchiness if selectivity is salt-and-pepper.

C: Long range connectivity within M1. The existence of horizontal long range connectivity has been demonstrated in multiple cortical areas, and in particular motor cortex (see DeFelipe et al., 1984). Such a long range connectivity could in principle give rise to the long-range correlations found by the authors.

D: Can this work be connected to the findings in Schwab et al., PRL 2014? If latent variables can give rise to signatures of criticality but do not require fine tuning of the brain's network state, might this not be a more biologically plausible origin for the long-range correlation observed here?

2) The main text is sorely lacking any details of the model analyzed and simulated by the authors. The reader has to go to p.47 of the Supplementary Material to finally understand the authors simulated a rate model with threshold-linear (ReLU) transfer functions. The model should be explained in the main text when it is first introduced, and the equations should be described in the Methods of the paper, and not confined in the Supplementary Material.

3) A related issue is that it remains unclear how general these theoretical results are. There are several places in the paper where the authors hint that their results are more general than the analyzed rate model, but it remains unclear whether this is the case, and in particular whether their results could also apply to networks of spiking neurons. This issue needs to be discussed more clearly.

4) Stationarity claims are not well substantiated. Figures 2 and 3 in the Appendix show that the population averaged activity is approximately stationary for different sessions. However, single neuron activity could be highly dynamic, but these fluctuations of activity could be washed out at the population level. It would be helpful to check whether trial-averaged single unit activity is indeed stationary during these epochs. If it is, then it should be shown in the Appendix. If the authors are not able to demonstrate stationarity at the single neuron level, then some of their conclusions should be toned down (in particular, the last paragraph of the Discussion). Note that this issue is strongly related to issue #1, scenario A.

5) There are claims of 'strong' spike count covariances at several points in the paper, but the covariances by themselves tell us nothing about the strength of correlations if we don't know the respective variances. In that respect, plotting correlations instead of covariances (as is done in many other studies of correlation structures in brain networks) would be much more informative about the strength of correlations between neurons. An example of why using covariances is problematic to assess strength of correlation is given by a comparison between Figures5 and 6: In Figure 5 covariances are of order 0.01, but they are of order 1 in Figure 6, likely because of a difference in mean rates of a factor 10 between the two.

6) The most interesting and compelling section of the paper is arguably the dependence of correlation on behavioral state. This result, though, seems a bit weak in magnitude. Are there other data to shore this up, or a null model to reveal how surprising this change is? A direct quantification of the covariance pattern change between epochs in Figure 6B, for example, would be good to see.

---

## [Author Response]

Essential revisions:1) The scenario put forward by the authors describe the data well, but a major drawback is that it needs fine tuning for the network to be close to the critical point. There are multiple other, and perhaps simpler, scenarios that could reproduce the data described by the authors, but they are not adequately discussed in this paper:

We thank the reviewers for agreeing with us that our derived mechanism describes the experimental data well. In the revised manuscript, we expand on the discussion of potential other mechanisms underlying the observed correlation structures and argue in more detail why we think the presented mechanism is the most plausible one.

A: Fluctuating and broadly divergent external inputs could in principle generate the observed correlation structure. The authors quickly dismiss this scenario in the discussion: 'In such a scenario, covariances have been shown to be predominantly generated locally rather than from external inputs (4,15)'. It was not clear where this is shown in these papers. The authors should explain much more clearly why they believe the external input scenario is unlikely (see also issue #5).

We agree with the reviewers that we cannot fully exclude the scenario in which the measured correlations in the local network below the Utah array are generated from particularly correlated fluctuating external inputs. However, this scenario would raise the question which mechanism causes these particular external input correlations. In sensory areas, neurons are strongly driven by input stimuli, which is often reflected in low-rank structures leading to sizable positive average covariances (Rosenbaum et al., 2017). The resting-state activity in motor cortex investigated here, however, lacks these direct stimulations as well as behaviorally related activity transients. We consequently observe only very weak average covariances that can be well explained by ongoing activity in balanced networks. Helias et al., (2014) (ref 4 in our previous version) investigates intrinsic and extrinsic sources of correlations in such networks and finds that intrinsic connectivity dominates correlations for recurrent network size N<10^7^(c_int_>>c_ext_, see their Figure 7a). Furthermore, for realistic local network sizes, local correlations only weakly depend on the amount of external correlations (their Figure 3a). Dahmen et al., (2019) (ref 19 in our previous version) shows the extreme case of driving a local network with correlated input from outside (see supplement section 3 “Effect of correlated external input”). In that setting, the correlated external population is of the same size as the local population, exhibiting the same intrinsic correlation structure as the local population. While the resulting correlated external input has some quantitative effect on the local statistics of covariances, their qualitative dependence on the spectral radius of the local network connectivity remains unchanged, as can be seen by all curves of normalized covariances in panel E collapsing on the same theoretical prediction that neglects correlated external inputs. Moreover, Ref 19 (Figure 4C) shows that the dependence of the correlations on the spectral bound is qualitatively the same in spatially organized networks. In the revised manuscript, we extended our discussion on external inputs to clarify why the local origin of the observed correlation structure seems most plausible. The new text in the Discussion reads:

“Spatially organized balanced network models have been investigated before in the limit of infinite network size, as well as under strong and potentially correlated external drive, as is the case, for example, in primary sensory areas of the brain (Rosenbaum et al., 2017; Baker et al., 2019). […] Despite sizable external input correlations projected onto the local circuit via potentially strong afferent connections, the dependence of the statistics of covariances on the spectral bound of the local recurrent connectivity is predicted well by the theory that neglects correlated external inputs (see supplement section 3 in Dahmen et al. (2019)).”

B: Recurrent connectivity correlated with neuronal selectivity, as in visual cortex, but with salt-and-pepper organization. This would be a scenario similar to what has been proposed in multiple visual cortex models (see old work by Tsodyks, Sompolinsky, and more recently ref (14)). The authors also dismiss this scenario because of the lack of patchiness of the correlation structure in M1, but we would not expect any patchiness if selectivity is salt-and-pepper.

We thank the reviewers for this remark. We agree with the reviewers that this point was not sufficiently discussed in the previous version of our manuscript. In the revised manuscript, we expanded the text in the discussion in the following way:

“Long-range connectivity, for example arising from a salt-and-pepper organization of neuronal selectivity with connections preferentially targeting neurons with equal selectivity (Ben-Yishai et al., 1995; Hansel and Sompolinsky, 1998; Roxin et al., 2005; Blumenfeld et al., 2006), would produce salt-and-pepper covariance patterns even in networks with small spectral bounds where interactions are only mediated via direct connections. […] In contrast, correlations due to imprinted network structures are static, so that a change in gain of the neurons will either strengthen or weaken the specific activity propagation, but it will not lead to a change of the sign of covariances that we see in our data.”

Furthermore, in order to emphasize more clearly that covariance patterns are highly dynamic as opposed to hard-wired via long-range direct connectivity, we expanded the explanation of Figure 6 to clarify that covariance patterns change across the two different epochs (S and P) of the reach-to-grasp experiment in a manner that makes a static origin less plausible:

“Despite similar overall distributions of covariances in S and P (Figure 6*D1*), covariances between individual neuron pairs are clearly different between S and P: Figure 6*B* shows the covariance pattern for one representative reference neuron in one example recording session of monkey N. […] In particular, the theory provides a mechanistic explanation for the different coordination patterns between neurons on the mesoscopic scale (range of a Utah array), which are observed in the two states S and P (Figure 6*B*). The coordination between neurons is thus considerably reshaped by the behavioral condition.”

We finally note that ref 14 mentioned by the reviewers employs a different mechanism than long-range direct connectivity. There the authors find in simulations that long-range coordination emerges close to linear instability by tuning anisotropies of local connection kernels in a neural field model. Through communication with the authors we identified that their observation can presumably be explained with our theoretical findings. The smoother heterogeneity (anisotropies) in ref 14, however, leads to stronger patchiness of covariance patterns than observed here for sparse networks. We contacted the authors of ref 14 and made sure that they agree with the discussion of their results in our manuscript and the way we contrast our work.

C: Long range connectivity within M1. The existence of horizontal long range connectivity has been demonstrated in multiple cortical areas, and in particular motor cortex (see DeFelipe et al., 1984). Such a long range connectivity could in principle give rise to the long-range correlations found by the authors.

We agree with the reviewers that long-range connections could play an important role in long-range coordination. However, to our knowledge, long-range connections are mostly patchy and predominantly excitatory, which could explain positive covariances at long distances, but would not provide an explanation for the large negative covariances between excitatory neurons at long distances (see e.g. Figure 1D). Furthermore, as already discussed above in response to point B, long-range connectivity would be static and therefore not change as much as we observe during different epochs of the reach-to-grasp experiment. Therefore, we believe that our mechanism via indirect interactions is more plausible. We added the reference to DeFelipe et al., in the Discussion of the revised manuscript:

“Likewise, embedded excitatory feed-forward motifs and cell assemblies via excitatory long-range patchy connections (DeFelipe et al., 1986) can create positive covariances at long distances (Diesmann et al., 1999; Litwin-Kumar and Doiron, 2012). Yet these connections cannot provide an explanation for the large negative covariances between excitatory neurons at long distances (see e.g. Figure 1*D*).”

D: Can this work be connected to the findings in Schwab et al., PRL 2014? If latent variables can give rise to signatures of criticality but do not require fine tuning of the brain's network state, might this not be a more biologically plausible origin for the long-range correlation observed here?

The work by Schwab et al., PRL 2014 studies how a power law shape of the ranked network state probabilities not only occurs at criticality but more generally in case of unobserved external inputs. In fact, the authors study ranked network state probabilities, whereas we here study the variance of covariances and their distance dependence which we find to be exponential in the long-range limit. As such, these two measures are clearly different. This is why it is unclear why the explanation of Zipf distributions based on latent variables could be an explanation for the long-range coordination studied here. Yet, the reviewers are right that any critical state also produces long-range correlations and that latent external inputs can lead to spurious signatures of criticality. As discussed above, we cannot fully exclude that an individual experimental finding is produced by a more specific mechanism than ours. However, we want to emphasize that we derive a single general mechanism that at the same time explains multiple experimental findings: broad covariances, a shallow decay of correlations and a sensitive state dependence. If, instead, external inputs and long-range connections were the underlying cause of these observations, it would require tuning of different sets of parameters to explain the observed correlation structures. Furthermore, we show that the here proposed mechanism is general in that it only relies on thelocal network connectivity. We agree that it requires sufficiently high variability of the connectivity for the network to reside in the dynamically balanced regime. Yet, this is not really a fine-tuning, since the dynamically balanced regime is not just a single point in parameter space. Also, since the effective connectivity depends on a large number of network and neuron parameters, there is a plethora of options for the network to a create and maintain a large value of the spectral bound. In the revised manuscript, we extended our discussion on this aspect in the following way:

“Note that both mechanisms of long-range coordination, the one studied in (Smith et al., 2018) and the one presented here, rely on the effective connectivity for the network to reside in the dynamically balanced critical regime. The latter regime is, however, not just one single point in parameter space, but an extended region that can be reached via a multitude of control mechanisms for the effective connectivity, for example by changing neuronal gains (Salinas and Sejnowski, 2001a,b), synaptic strengths (Sompolinsky et al., 1988), and network microcircuitry (Dahmen et al., 2020).”

2) The main text is sorely lacking any details of the model analyzed and simulated by the authors. The reader has to go to p.47 of the Supplementary Material to finally understand the authors simulated a rate model with threshold-linear (ReLU) transfer functions. The model should be explained in the main text when it is first introduced, and the equations should be described in the Methods of the paper, and not confined in the Supplementary Material.

We thank the reviewers for this comment and moved the model details from the supplement to the main text and Methods section. We would like to point out that the specific rate model has only been chosen for Figure 5 to make the point of the state-dependence intuitive. The main results are not derived for this specific model, but are based on model-independent properties of linear-response theory: In a previous work we have explicitly shown (see Grytskyy et al., (2013) and Figure 4 in Dahmen et al., (2019)) that theoretical predictions are in agreement with simulations of networks of spiking leaky integrate-and-fire models. We discuss this in the new initial text of the methods section “Network model simulation*”.*

3) A related issue is that it remains unclear how general these theoretical results are. There are several places in the paper where the authors hint that their results are more general than the analyzed rate model, but it remains unclear whether this is the case, and in particular whether their results could also apply to networks of spiking neurons. This issue needs to be discussed more clearly.

We thank the reviewers for pointing out that the previous version of our manuscript was not stressing enough the generality of our findings and applicability to networks of spiking neurons. As mentioned above, our study builds on top of a series of works (Tetzlaff et al., 2012; Trousdale et al., 2012; Pernice et al., 2012; Grytskyy et al., 2013; Helias et al., 2013; Dahmen et al., 2019) that showed quantitative agreement of predicted correlations by the employed linear-response theory with simulations of spiking neurons. We added the following explanation to the discussion of the revised manuscript:

“Our analysis of covariances on the single-neuron level goes beyond the balance condition and requires the use of field-theoretical techniques to capture the heterogeneity in the network (Dahmen et al., 2019; Helias and Dahmen, 2020). It relies on a linear-response theory, which has previously been shown to faithfully describe correlations in balanced networks of nonlinear (spiking) units (Tetzlaff et al., 2012; Trousdale et al., 2012; Pernice et al., 2012; Grytskyy et al., 2013; Helias et al., 2013; Dahmen et al., 2019). These studies mainly investigated population-averaged correlations with small spectral bounds of the effective connectivity. Subsequently, (Dahmen et al., 2019) showed the quantitative agreement of this linear-response theory for covariances between individual neurons in networks of spiking neurons for the whole range of spectral bounds, including the dynamically balanced critical regime. The long-range coordination studied in the current manuscript requires the inclusion of spatially non-homogeneous coupling to analyze excitatory-inhibitory random networks on a two-dimensional sheet with spatially decaying connection probabilities. This new theory allows us to derive expressions for the spatial decay of the variance of covariances.”

4) Stationarity claims are not well substantiated. Figures 2 and 3 in the Appendix show that the population averaged activity is approximately stationary for different sessions. However, single neuron activity could be highly dynamic, but these fluctuations of activity could be washed out at the population level. It would be helpful to check whether trial-averaged single unit activity is indeed stationary during these epochs. If it is, then it should be shown in the Appendix. If the authors are not able to demonstrate stationarity at the single neuron level, then some of their conclusions should be toned down (in particular, the last paragraph of the Discussion). Note that this issue is strongly related to issue #1, scenario A.

We agree with the reviewers that the population averaged statistics shown in Figures 2 and 3 in the Appendix of the previous version were not sufficient to claim that the activity was stationary within a given epoch (S or P) of the reach-to-grasp experiment. Therefore, we performed the analysis suggested by the reviewers and provide trial-averaged single unit activity in the new Figures 5 and 6 of the revised Appendix. These figures show that the majority of neurons has stationary activity statistics within the relevant epochs S and P, especially when comparing to their whole dynamic range that is explored during movement transients towards the end of the task. The statistics changes notably between the two epochs S and P. This is in line with the second test for stationarity that we show in Figure 6E,F. There we compare the firing rate and covariance changes between two 200ms segments of the same epoch to the firing rate and covariance changes between two 200ms segments of different epochs. If the neural activity was not stationary within an epoch then we would not obtain correlation coefficients of almost 1 between firing rates in Figure 6E and correlation coefficients up to 0.9 between covariance patterns within one epoch in Figure 6F. In summary, the analyses together make us confident that assuming stationarity within an epoch is a good approximation to show that there are significant behaviorally related changes in covariances across epochs of the reach-to-grasp experiment.

5) There are claims of 'strong' spike count covariances at several points in the paper, but the covariances by themselves tell us nothing about the strength of correlations if we don't know the respective variances. In that respect, plotting correlations instead of covariances (as is done in many other studies of correlation structures in brain networks) would be much more informative about the strength of correlations between neurons. An example of why using covariances is problematic to assess strength of correlation is given by a comparison between Figures5 and 6: In Figure 5 covariances are of order 0.01, but they are of order 1 in Figure 6, likely because of a difference in mean rates of a factor 10 between the two.

We agree with the reviewers that correlation coefficients are easier to interpret than covariances. In the revised manuscript we therefore begin with discussing correlation coefficients of spike counts in the experimental data. We replaced the covariances in the previous version of Figure 1 by correlation coefficients, showing that spike counts between individual neuron pairs exhibit strong positive and negative correlations. To show that the differences in correlations across neuron pairs stem from largely differing covariances, we added a new figure to the revised Appendix (Appendix Figure 1) showing correlation coefficients, covariances, and variances side-by-side. Note that the spread of covariances is a more robust measure than the spread of correlation coefficients, because the latter depend on single neuron properties, such as the individual neurons’ firing rates. For that reason, a theory for covariances depends on fewer parameters. For the remainder of the revised manuscript we therefore prefer to show covariances rather than correlations and added explanations to relate covariance magnitudes in Figures 5 and 6, as the reviewers correctly suggested.

“The mechanistic model in the previous section shows a qualitatively similar scenario (Figure 5*C,E*). By construction it produces different firing rate patterns in the two simulations. While the model is simplistic and in particular not adapted to quantitatively reproduce the experimentally observed activity statistics, its simulations and our underlying theory make a general prediction: Differences in firing rates impact the effective connectivity between neurons and thereby evoke even larger differences in their coordination if the network is operating in the dynamically balanced critical regime (Figure 5*D*).”

6) The most interesting and compelling section of the paper is arguably the dependence of correlation on behavioral state. This result, though, seems a bit weak in magnitude. Are there other data to shore this up, or a null model to reveal how surprising this change is? A direct quantification of the covariance pattern change between epochs in Figure 6B, for example, would be good to see.

We thank the reviewers for sharing our interest in the state dependence of covariances that we showed during the reach-to-grasp task. And we agree with the reviewers that some reference case is needed in order to judge how surprising the change of covariances between epochs is. There has already been a reference in the previous version of the manuscript, which we now see was not explained well enough. Since measurements of covariances based on a finite number of data points are noisy, we compare the changes in covariances within two 200ms segments of the same epoch to the changes in covariances between 200ms segments of two different epochs (Figure 6E,F). This statistics also includes a quantitative comparison between the two covariance patterns shown in Figure 6B as one data point (session 8). We clarify this in the revised caption of Figure 6. Furthermore, we expanded the explanation and discussion of Figure 6B-F in the text explaining how we gauge the observed changes between behaviorally different epochs. The revised text reads:

“Within each epoch, S or P, the neuronal firing rates are mostly stationary, likely due to the absence of arm movements which create relatively large transient activities in later epochs of the task which are not analyzed here (see Appendix 1 Section 3). […] The coordination between neurons is thus considerably reshaped by the behavioral condition.”